# A Scalable Distributed Framework for Multimodal GigaVoxel Image Registration

Rohit Jena[*,†]    Vedant Zope[†]    Pratik Chaudhari    James C. Gee

University of Pennsylvania
Philadelphia, PA 19104, USA
{rjena, zope, pratikac}@seas.upenn.edu, gee@upenn.edu

## Abstract

In this work, we propose **FFDP**, a set of IO-aware non-GEMM fused kernels supplemented with a distributed framework for image registration at unprecedented scales. Image registration is an inverse problem fundamental to biomedical and life sciences, but algorithms have not scaled in tandem with image acquisition capabilities. Our framework complements existing model parallelism techniques proposed for large-scale transformer training by optimizing non-GEMM bottlenecks and enabling convolution-aware tensor sharding. We demonstrate unprecedented capabilities by performing multimodal registration of a $100\mu m$ *ex-vivo* human brain MRI volume at native resolution – an inverse problem more than $570\times$ larger than a standard clinical datum in about a minute using only 8 A6000 GPUs. FFDP accelerates existing state-of-the-art optimization and deep learning registration pipelines by upto $6 - 7\times$ while reducing peak memory consumption by $20 - 59\%$. Comparative analysis on a $250\mu m$ dataset shows that FFDP can fit upto $64\times$ larger problems than existing SOTA on a single GPU, and highlights both the performance and efficiency gains of FFDP compared to SOTA image registration methods.

## 1 Introduction

Image registration (also called 'image alignment' or 'image matching') is a non-linear inverse problem ubiquitous in biomedical and life sciences. Given $d$-dimensional images $F : \Omega \to \mathbb{R}^d$ and $M : \Omega \to \mathbb{R}^d$ defined on domain $\Omega$ (usually a compact subset of $\mathbb{R}^d$), image registration seeks to find a coordinate transform $\varphi : \Omega \to \Omega$ that deforms the moving image $M$ to look similar to the fixed image $F$. Mathematically, we minimize the following objective (Fig. 1):

$$\varphi^* = \arg\min_{\varphi \in G} L(\varphi) \doteq C(F, M \circ \varphi) + R(\varphi) \tag{1}$$

where $C$ is a cost or dissimilarity function, and $\circ$ is the interpolation operator, i.e. $(I \circ g)(x) = I(g(x))$ for all $x \in \Omega$. Popular choices of $\varphi$ are affine and deformable transforms, i.e. $\varphi(x) = Ax + t$, and $\varphi(x) = x + u(x)$. Modern registration pipelines (Hoffmann et al., 2021; Jena et al., 2024a) consider an affine matching followed by a deformable matching step, resulting in a composite transform $\varphi(x) = Ax + t + u(x)$. $u$ is called the displacement field, modeled as a grid of per-voxel vectors $u(x) \in \mathbb{R}^d$. For an image of size $N$, the displacement field is a tensor of size $dN$. We use $[\mathbf{x}]_\Omega$, $A[\mathbf{x}]_\Omega + t$, and $[\mathbf{u}]_\Omega$ to denote the identity grid, grid of affine transformed coordinates, and deformation grid defined on $\Omega$ respectively. Common choices of $C$ are mean squared error, Localized Normalized Cross Correlation (Avants et al., 2008a), and Mattes Mutual Information (Mattes et al., 2001). Common choices of $R$ include Sobolev norm of the gradient or warp fields (Beg et al., 2005; Mang et al., 2019; Avants et al., 2008b), total variation, and inverse-consistency (Christensen & Johnson, 2001). To optimize Eq. (1), iterative methods optimize $\varphi^*$ directly using gradient descent, and deep learning methods learn a deep neural network $\varphi = f_\theta(F, M)$. Image registration establishes a common coordinate system, aligning scans across individuals and atlases (Hering et al., 2022;

---

[*]Corresponding author.
[†]These authors contributed equally.

**Figure 1: Image Registration Problem. (a)**: The task is to find a coordinate transform that warps the moving image $M$ to the fixed image $F$. Individual corresponding points are shown as violet arrows; the per-pixel coordinate transform is shown as a warp field $\varphi$, and the transformed image $M \circ \varphi$. **(b)**: A typical registration pipeline - the grid sampler warps the moving image, that is then compared to the fixed image using a loss function. **Green** denotes the optimizable warp, **red** denotes the primary bottlenecks that we optimize in this paper.

Marcus et al., 2007; Murphy et al., 2011). This alignment is a prerequisite for multimodal data fusion, cross-subject comparison, morphometric analysis (Das et al., 2009), and construction of large-scale atlases (Wang et al., 2020b). Establishing such voxelwise correspondence is fundamental for studying anatomical variability, detecting pathological signatures (Ravikumar et al., 2021), and advancing precision medicine (Börner et al., 2022; Jonsson et al., 2022). The saliency and centrality of the task across various biomedical and life science applications has spurred numerous methodological advances in the field, spanning more than three decades of research (Gee et al., 1993; Tian et al., 2024).

Over the past decade, advances in MRI, CT, PET, STPT, and microscopy have enabled ultra-high-resolution imaging, often more than three orders of magnitude larger than macroscopic biomedical domains (Balchandani & Naidich, 2015; Esquivel et al., 2022; Badawi et al., 2019; Gambarotto et al., 2019; Wassie et al., 2019; Kleven et al., 2023; Wang et al., 2020b; Mansour et al., 2025; Kleinfeld et al., 2011). While a typical clinical registration problem involves ∼20M parameters, high-resolution ex-vivo human brain scans can require solving up to 11B parameters, far beyond the ∼50M-parameter scale at which current registration methods remain reliable. As a result, state-of-the-art deformable image alignment struggles to scale to the resolutions demanded in modern neuroimaging, computational pathology, developmental biology, and connectomics, creating a substantial performance gap. In parallel, innovations in large-scale transformer training such as IO-aware fused operations (Dao et al., 2022; Dao, 2023; Spector et al., 2025) and 5D parallelism for distributing larger-than-memory workloads (Shoeybi et al., 2019; Li et al., 2023; Jacobs et al., 2024; Li et al., 2024; Zhao et al., 2023; Ansel et al., 2024) optimize GEMM-like workflows. However, the fundamental concepts utilized by these methods (IO-awareness, recomputing and aggregating intermediates on shared memory to minimize high bandwidth memory (HBM) storage, identifying partial aggregates across hosts to minimize communication overheads for distributed optimization) are broadly applicable to a wide class of problems of the non-GEMM nature.

In this paper, we apply these concepts to scale image registration algorithms to match parity with the developments in both increasing resolution of image acquisition *and* compute capabilities. To that end, our contributions are twofold. First, we identify key compute and memory bottlenecks in image registration algorithms, and propose novel components that fit problems upto $64\times$ larger than existing algorithms on a single GPU. Second, we propose **F**lash **F**used **D**istributed **P**rimitives (FFDP), a distributed framework to scale registration to an arbitrary number of GPUs, thereby scaling to ultra high-resolution problems. We present a first-of-its-kind demonstration: aligning a $250\mu m$ in-vivo MRI (Lüsebrink et al., 2017) to a $100\mu m$ *ex-vivo* human brain FLASH volume (Edlow et al., 2019) – a multimodal registration problem more than **570**$\times$ larger than a standard clinical datum (Marcus et al., 2007), with over 11.8B transform parameters – completed in *one minute* using only 8 A6000 GPUs. FFDP accelerates existing traditional registration pipelines by upto $7.48\times$ while reducing memory consumption by upto $59\%$, and deep learning pipelines by upto $6.14\times$ while consuming upto $24\%$ less memory. We highlight the necessity of performing high-resolution registration by comparing our method with various SOTA optimization and deep learning baselines on a $250\mu m$ T1-weighted MRI dataset, showing unprecedented performance and gains in efficiency.

## 2 RELATED WORK

### 2.1 MEMORY EFFICIENT AND LARGE SCALE OPTIMIZATION

Recent years have also witnessed tremendous innovations in large-scale transformer model training. IO-aware implementations typically include individual fused kernels (Dao et al., 2022; Dao, 2023) and domain-specific languages (Spector et al., 2025; PyTorch, 2025) to minimize launch latency and large memory overheads. To distribute larger-than-memory model training workloads across multiple GPUs, 5D parallelism techniques (Shoeybi et al., 2019; Li et al., 2023; Jacobs et al., 2024; Li et al., 2024; Zhao et al., 2023; Ansel et al., 2024) have been proposed. Many of these techniques leverage a divide-and-conquer approach to break down a larger GEMM-like operation like matrix multiplication or attention into smaller sub-problems that can be executed on multiple GPUs and synchronized to compute the final result. To our knowledge, most of these techniques are tailored to transformer-specific architectures and GEMM-like operations (self attention, FeedForward, LayerNorm, etc.) only, and a Model Parallel variant for convolution-aware tensor sharding and synchronization is not available.

### 2.2 LARGE SCALE REGISTRATION IN LIFE SCIENCES AND BIOMEDICAL IMAGING

***Ex-vivo* neuroimaging.** Neuroanatomical studies often integrate high-resolution ex-vivo MRI, blockface imaging, and histology to bridge the gap between in-vivo imaging and microscopic "gold standard" pathology (Casamitjana et al., 2025; Ravikumar et al., 2024). While large-scale consortia like SEA-AD and HMBA, along with submillimeter whole-brain datasets (Edlow et al., 2019; Lüsebrink et al., 2017), aim to map cellular and molecular organization across species, computational costs often limit analysis to local effects. Specifically, existing tools cannot register these datasets at native resolution due to excessive memory requirements. We overcome this limitation, demonstrating native-resolution registration of whole-brain datasets in one minute using eight A6000 GPUs (see Section 5.2), thereby preserving fine anatomical details typically lost to downsampling.

**Large-scale registration in model organisms.** Over the past decade, imaging across the life sciences and biomedical domains has progressed from mesoscale surveys to organ- and organism-wide acquisitions at cellular or even subcellular resolution. These span transparent organisms and small animal models (e.g., *C. elegans*, zebrafish, adult *Drosophila*) (Varol et al., 2020; Venkatachalam et al., 2016; Marquart et al., 2017; Gupta et al., 2018; Peng et al., 2011; Brezovec et al., 2024), adult mouse and rat brains imaged at sub-micron resolutions (Gong et al., 2016; Wang et al., 2020a; Kleven et al., 2023) using Light Sheet Fluorescence Microscopy (LSFM) and Serial Two-Photon Microscopy (STPT) imaging. Such modalities routinely generate gigavoxel to teravoxel volumes (Kutten et al., 2016; Nazib et al., 2018). Their scientific utility, however, hinges on the ability to perform registration at the native resolution of acquisition, i.e. aligning specimens (or modalities) in a common coordinate system without sacrificing the fine-scale morphologies including cell bodies, layers, axon bundles, synaptic neighborhoods, etc. that motivate high-resolution acquisition in the first place (Nazib et al., 2018; Goubran et al., 2013).

Across these diverse domains, the unifying requirement demands access to scalable multimodal registration algorithms - a challenge we address in this work. We provide an extended discussion of more related work and the necessity of our approach in Section A.

## 3 FUSED KERNELS FOR MEMORY EFFICIENT REGISTRATION ON A SINGLE GPU

**Bottlenecks of a deformable image registration pipeline** Our primary objective is to identify compute and memory bottlenecks in *large-scale* image matching tasks. In identifying these bottlenecks, training-free optimization methods are better suited than deep networks since the latter has a much larger activation memory footprint, which forms the primary memory bottleneck (Tazi et al., 2024). For instance, for a $250\mu m$ image pair, a standard deep learning method (Hoffmann et al., 2021) generates an activation map of size 27GB only after the first layer. Extrapolating memory usage for clinical data, existing deep networks will require upto 1.2TB of GPU memory at inference to process these image volumes at native resolution. In contrast, a training-free optimizer can fit this problem in less than 45GB of GPU memory. We use FireANTs (Jena et al., 2024a) as our base framework to identify compute and memory bottlenecks in a typical image registration problem. We analyze

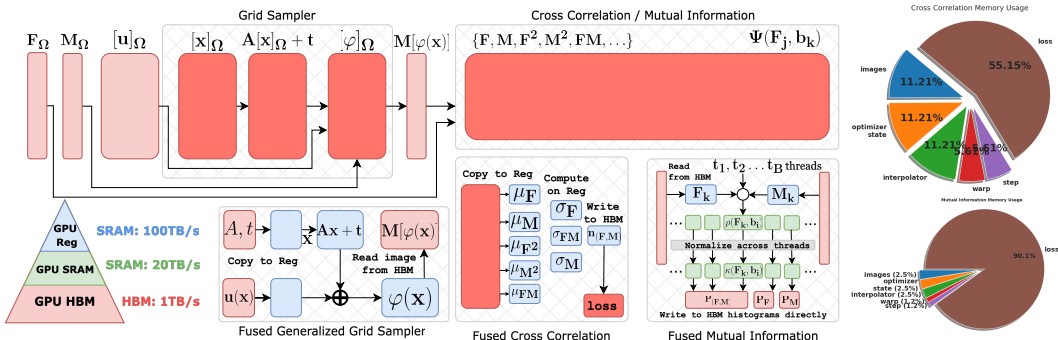

**Figure 2: Left**: FFDP uses fused kernels to eliminate intermediate HBM memory usage (in **dark red**) for memory-bound workhorse operations (grid_sampler, LNCC, MI) for large-scale image registration. For grid_sampler and LNCC, additional intermediate per-pixel variables (warp coordinates, patchwise statistics) are computed per-pixel in registers (blue). For MI, the Parzen Windowing and histogram aggregation is performed using shared memory (green), avoiding large HBM overheads. **Right**: Pie charts show the breakdown of memory overheads for storing the image, grid, optimizer state, and intermediate variables for MI and LNCC losses.

the flamegraph of a typical clinical MRI registration task from the *OASIS* brain dataset (Marcus et al., 2007) in Fig. 20. We identify three key memory bottlenecks in image matching pipelines (1) deformable interpolation and warp composition (2) cross-correlation loss, and (3) mutual information loss (see Fig. 2(right)). [1] We first propose efficient designs to fit larger problems on a single GPU, and then extend the framework to distributed registration.

### 3.1 Composite Implicit Grid Sampler

A fundamental operation used in image registration is the *grid sampler*. This operator allows us to warp an image $M$ using a deformation field $\varphi : \Omega \to \Omega$ and computes the image $M' : M'(x) = M(\varphi(x))$. Virtually every image registration pipeline uses this operation to warp the moving image using an affine, deformable, or composite transform. For affine and composite transforms, the operator initializes a regular grid $[\mathbf{x}]_\Omega$, a grid of size $3N$. The affine grid $A[\mathbf{x}]_\Omega + t$ is another grid of size $3N$. If a deformable grid $[\mathbf{u}]_\Omega$ is optimized, then a third grid $A[\mathbf{x}]_\Omega + t + [\mathbf{u}]_\Omega$ is materialized, costing a total of $9N$ overhead for an image of size $N$. To consolidate these memory overheads, we propose a composite implicit grid sampler. This is a fused CUDA kernel that performs the following operation:

$$\texttt{fused\_grid\_sampler}(I; A, t, [\mathbf{u}], S, x_{\text{bounds}})(x) = I(Ax + t + Su(x))$$

where $A, S \in GL(d, \mathbb{R})$ are affine matrices, $t$ is a translation vector, $[\mathbf{u}]$ is the deformation grid, and $x_{\text{bounds}}$ are the bounds of the (implicit) identity grid $[\mathbf{x}]_\Omega$. There are three benefits of this approach. First, the kernel avoids materializing any additional grids in HBM, reducing the memory overhead of the kernel from $O(n)$ to $O(1)$ with no loss in runtime or accuracy. Second, when the warp $[\mathbf{u}]_\Omega$ is sharded across hosts in a distributed setting, the identity grid $[\mathbf{x}]_\Omega$ needs to be sharded correctly too. Since the identity grid is implicitly defined by its bounds $x_{\text{bounds}} = (x_{\min}, x_{\max}) \in \mathbb{R}^{2d}$, our implementation can be easily used in a distributed optimization setting without instantiating partial shards $[\mathbf{x}]_{\Omega_h}$. Finally, the matrix $S$ is used to rescale the deformation field to sample from the coordinates of the sharded images $I_h$ which lie on the grid $\Omega_h$ instead of $\Omega$ (see Section I.2) without initializing additional memory. The backward pass is very similar to the existing PyTorch implementation, with the exception of the gradient of the affine matrix. We discuss the derivation and pseudocode of the forward and backward pass in the Section H.

### 3.2 Implicit Parzen Windowing for Mutual Information

Mattes Mutual Information (MI) is one of the most commonly used loss functions for *multimodal* image matching (Chen et al., 2022; Avants et al., 2009; Mattes et al., 2001). For random variables $X$ and $Y$, MI is the KL divergence between the joint distribution $P(X, Y)$ and product of marginal

---

[1] A GPU's memory hierarchy spans multiple tiers: registers (per-thread, single-cycle), shared memory/L1 cache (on-chip, tens of KB, low latency within a block), L2 cache (MBs, shared across SMs, moderate latency), and global memory (**HBM**). Our work focuses on reducing HBM usage for key non-GEMM operations used in image registration, by maximizing register and shared memory usage while minimizing global memory traffic.

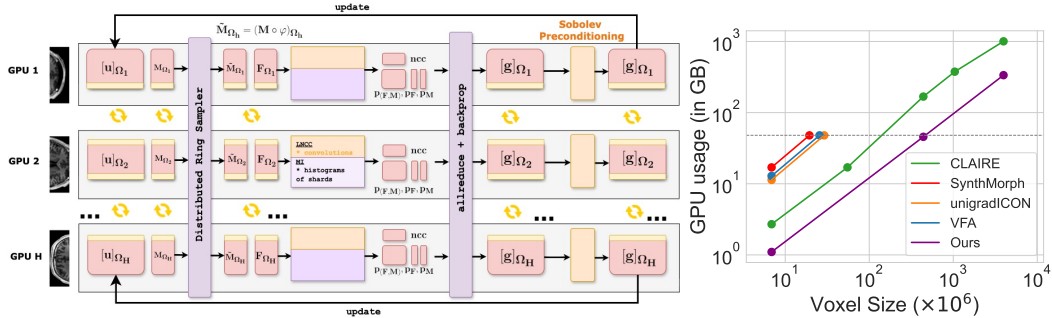

**Figure 3: Left**: Overview of our distributed framework. GridParallel (GP) shards the fixed and moving images $(F, M)$ and the warp field $[\mathbf{u}]$ across multiple GPUs. Yellow blocks and arrows denote synchronized halo boundaries between GPUs, enabling smoothing on images and warp fields without an allgather. The ring sampler (violet) computes interpolated image shards on the fly, avoiding materialization of the full moving image. We then compute losses (MSE, LNCC, MI), compute gradients w.r.t. each warp shard, apply Sobolev regularization with GP, and update shards by gradient descent. **Right**: Scaling efficiency compared to deep methods and CLAIRE (Mang et al., 2019), a distributed registration method. Most SOTA deep learning baselines require orders-of-magnitude more memory for the same problem size and scalability is limited to a single GPU (dotted line). Our framework scales to arbitrarily large problem sizes while using about $5\times$ less memory than CLAIRE.

distributions $P(X)P(Y)$ of the intensities of the two images. For image matching, $X$ and $Y$ are the pixel intensities for the images $I, J$. The distributions are estimated using a kernel density estimator:

$$P_I(v) = \frac{1}{N} \sum_k \kappa(v - I_k), \quad P_{(I,J)}(v, w) = \frac{1}{N} \sum_k \kappa(v - I_k)\kappa(w - J_k) \tag{2}$$

where $\kappa$ is a kernel function of choice. Common choices of $\kappa$ are the Gaussian (Guo, 2019) and 3rd order B-Spline kernels (Thévenaz & Unser, 2000). To empirically compute the KL divergence, the distributions Eq. (2) are discretized over $B$ equally spaced bins on the domain of $u \in I, v \in J$. However, to compute the joint histogram of size $B^2$, this method requires materializing the entire Parzen Block $\Psi_I(j, k) = \kappa(b_j - I_k)$ of size $2k_P BN$, where $k_P$ is a kernel-dependent constant. Since $N >> B$ ($B$ is typically chosen to be 32), this operation becomes a significant memory bottleneck for large $N$. For instance, a typical clinical image volume ($N \approx 30\text{MB}$) with 32 bins will consume **7.5GB** of HBM - a significantly huge cost that grows much faster for larger problems.

Our efficient implementation leverages the fact that $B$ is small to avoid materializing the tensors $\Psi_I, \Psi_J \in \mathbb{R}^{B \times N}$ altogether and use high-throughput shared memory to compute and accumulate the histogram entries and partial gradients for each image pixel. We provide the detailed derivation in Section G. This leads to an efficient implementation that consumes $O(1)$ additional HBM instead of $O(N)$ (holding $B$ constant). This leads to upto **98%** lesser HBM usage for images considered in our experiments, and an asymptotic $100\%$ reduction in HBM usage for large images (Fig. 7(top-right)).

### 3.3 Efficient Implicit Fused Cross-Correlation

Local Normalized Cross-Correlation (LNCC) is used ubiquitously in signal and image processing as a similarity metric. In deformable image registration, it is used as a robust similarity function to compare anatomical similarities (Chen et al., 2022; Hoffmann et al., 2021; Avants et al., 2008b; Wu et al., 2024). Most LNCC implementations are memory-bound due to the large number of intermediate variables. Our analysis in Section F shows that the computational graph adds $16\times$ HBM overhead, and upto another $16\times$ HBM overhead for computing gradients with respect to all intermediates.

To avoid these huge memory overheads, we fuse all the intermediate computation in a fused kernel. Our fused forward pass requires only $5\times$ memory for storing all intermediates ($I, J, I^2, J^2, IJ$ convolved with matrix $w$). In Section F we analytically derive the gradient and show that the input gradients can be computed by modifying the saved intermediates *in-place*. This leads upto a **76.5% reduction** in memory (see Table 3) and outperforms even `torch.compile` implementations.

### 4 Extending image registration to multiple GPUs

Our composite implicit grid sampler and improved loss functions allows optimizing problems with image sizes that are upto two magnitudes larger than other baselines on a single A6000 GPU (Fig. 5a).

**Figure 4:** **(a)** Neighboring coordinates in the warp field may refer to pixel locations on arbitrary image shards due to the deformable nature of the warp field, making distributed interpolation non-trivial. **(b)** Ring Sampler interleaves fetching of image shards and aggregating the partial sums of interpolated values, avoiding a memory-expensive allgather. **(c)** Bilinear Interpolation is decomposed into partial sums over image shards, which are accumulated with a ring topology communication, similar to Liu et al. (2024b).

However, many applications using mesoscopic and microscopic data require registration of images that do not fit on a single GPU. Inspired by distributed frameworks for LLM training (Shoeybi et al., 2019; Rajbhandari et al., 2020) and initial work on distributed image registration (Mang et al., 2019), we propose a distributed framework that allows sharding large images across multiple GPUs to efficiently scale to arbitrarily large problem sizes with any similarity loss function.

**Distributed Setting.** For distributed registration with $H$ hosts or GPUs, we partition the domain $P(\Omega) = \{\Omega_1, \Omega_2, \dots \Omega_H\}$ such that $|\Omega_i| = N/H$, $\Omega_i \cap \Omega_j = \phi \quad \forall i \neq j$ and $\cup_i \Omega_i = \Omega$. We use $[\mathbf{x}]_{\Omega_h}$, $A[\mathbf{x}]_{\Omega_h} + t$, and $[\mathbf{u}]_{\Omega_h}$ to denote the sharded tensors defined on domain $\Omega_h$.

### 4.1 Grid Parallel for Boundary-Synchronized Image Sharding

Techniques like Tensor/Sequence/Expert/Context Parallel have been tremendously successful in distributed optimization by sharding large models and sequences across multiple GPUs (Shoeybi et al., 2019; Li et al., 2023; Liu et al., 2024b;a). However, these techniques work for transformer-like architectures and input sequences where the model parameters and activations do not require boundary synchronization. In contrast, image registration contains operations that require boundary synchronization between image and grid shards to perform mathematically correct convolutions. Examples of such operations include convolutions for calculating LNCC, total variation loss, Sobolev norm of the gradient and warp fields (Mang et al., 2019; Avants et al., 2008b; Beg et al., 2005).

To enable these functionalities and complement existing parallelism techniques, we propose '*Grid Parallel*' (GP) as an abstraction on a tensor. GP shards a tensor across hosts, stores the sharded dimension and bounds as metadata, and provides synchronization operations to augment the tensor with sufficient boundary padding from neighboring shards prior to performing a convolution operation. GP allows us to partition the fixed images, $[\mathbf{u}]$, and the optimizer state $[\mathbf{m_1}], [\mathbf{m_2}]$ – essentially sharding the entire problem across $H$ hosts while allowing the user to apply convolutional operations seamlessly. We compare the performance of GP with naive DTensor sharding in Section D.

### 4.2 Distributed Ring Sampler

Despite the sharding in GP, the moving image $M$ cannot be sharded across GPUs due to the random-access nature of the `grid_sample` operation applied on $M$. In general, the warp vector $\varphi(x)$ residing on GPU $i$ can point to coordinates that reside on the sharded image on GPU $j$ for any $j \neq i$. Even for neighboring coordinates $x_s, x_u \in [\mathbf{x}]_i$, the coordinates $\varphi(x_s)$ and $\varphi(x_u)$ can point to different shards $j_1 \neq j_2 \neq i$. This is illustrated in Fig. 4(a). Keeping the entire moving image in memory limits the maximum problem size to $N \leq V$, where $V$ is the memory per GPU, regardless of the number of hosts $H$. However, we want the maximum problem size to scale with $H$. Therefore, we propose a distributed `grid_sampler` that allows us to *correctly* interpolate the moving image with sharded images scattered across multiple hosts without performing an `allgather` operation on the moving image.

Our approach leverages the key observation that (bi/tri)linear interpolation can be decomposed as an aggregate of partial sums of interpolated values on individual image shards. Fig. 4(b) illustrates this example. These individual image shards are sent across hosts in a ring topology, similar to Liu et al. (2024b), and the partial sum is aggregated inplace. This operation only incurs an additional $N/H$ HBM overhead for fetching the sharded image from other hosts, scaling efficiently to arbitrary large

**(a)** Performance comparison across methods and resolutions.

| Resolution | Method | AvgDice Score ↑ | InvDice Score ↑ | AvgHD90$^{\mathrm{cum}}$ (mm) ↓ |
|---|---|---|---|---|
| 1 mm | Baseline | $0.579 \pm 0.055$ | $0.141 \pm 0.142$ | $1.587 \pm 0.908$ |
| | Anatomix | $0.796 \pm 0.035$ | $0.386 \pm 0.138$ | $0.468 \pm 0.137$ |
| | CLAIRE | $0.776 \pm 0.044$ | $0.344 \pm 0.120$ | $0.554 \pm 0.150$ |
| | FireANTs | $0.822 \pm 0.032$ | $0.435 \pm 0.147$ | $0.393 \pm 0.126$ |
| | ITK-dreg | $0.662 \pm 0.055$ | $0.199 \pm 0.125$ | $1.002 \pm 0.277$ |
| | SynthMorph | $0.801 \pm 0.022$ | $0.378 \pm 0.133$ | $0.455 \pm 0.098$ |
| | TransMorph | $0.851 \pm 0.016$ | $0.468 \pm 0.161$ | $0.310 \pm 0.064$ |
| | UniGradICON (IO) | $0.826 \pm 0.022$ | $0.391 \pm 0.155$ | $0.384 \pm 0.095$ |
| | UniGradICON | $0.815 \pm 0.026$ | $0.393 \pm 0.156$ | $0.419 \pm 0.113$ |
| | VFA | $0.851 \pm 0.023$ | $0.494 \pm 0.169$ | $0.323 \pm 0.096$ |
| | Ours | $0.838 \pm 0.028$ | $0.436 \pm 0.148$ | $0.341 \pm 0.109$ |
| 500 µm | Baseline | $0.580 \pm 0.055$ | $0.138 \pm 0.143$ | $1.357 \pm 0.326$ |
| | Anatomix$^\dagger$ | $0.758 \pm 0.040$ | $0.325 \pm 0.159$ | $0.619 \pm 0.169$ |
| | CLAIRE | $0.779 \pm 0.051$ | $0.275 \pm 0.210$ | $0.570 \pm 0.211$ |
| | FireANTs | $0.841 \pm 0.033$ | $0.489 \pm 0.163$ | $0.340 \pm 0.127$ |
| | ITK-dreg | $0.699 \pm 0.056$ | $0.240 \pm 0.130$ | $0.834 \pm 0.254$ |
| | SynthMorph$^\dagger$ | $0.771 \pm 0.035$ | $0.337 \pm 0.133$ | $0.557 \pm 0.144$ |
| | TransMorph$^\dagger$ | $0.759 \pm 0.028$ | $0.300 \pm 0.175$ | $0.624 \pm 0.127$ |
| | UniGradICON$^\dagger$ | $0.610 \pm 0.044$ | $0.133 \pm 0.122$ | $1.231 \pm 0.262$ |
| | UniGradICON (IO)$^\dagger$ | $0.615 \pm 0.047$ | $0.149 \pm 0.136$ | $1.527 \pm 1.495$ |
| | VFA$^\dagger$ | $0.805 \pm 0.044$ | $0.419 \pm 0.181$ | $0.462 \pm 0.163$ |
| | Ours | $0.872 \pm 0.028$ | $0.528 \pm 0.180$ | $0.258 \pm 0.099$ |
| 250 µm | Baseline | $0.580 \pm 0.055$ | $0.136 \pm 0.141$ | $1.409 \pm 0.322$ |
| | Anatomix$^\dagger$ | $0.620 \pm 0.031$ | $0.161 \pm 0.115$ | $1.179 \pm 0.190$ |
| | CLAIRE | $0.809 \pm 0.054$ | $0.378 \pm 0.133$ | $0.570 \pm 0.211$ |
| | FireANTs$^\dagger$ | $0.777 \pm 0.064$ | $0.341 \pm 0.199$ | $0.629 \pm 0.295$ |
| | ITK-dreg | $0.758 \pm 0.048$ | $0.299 \pm 0.125$ | $0.613 \pm 0.191$ |
| | SynthMorph$^\dagger$ | $0.690 \pm 0.052$ | $0.243 \pm 0.164$ | $0.882 \pm 0.239$ |
| | TransMorph$^\dagger$ | $0.689 \pm 0.044$ | $0.191 \pm 0.132$ | $0.973 \pm 0.245$ |
| | UniGradICON (IO)$^\dagger$ | $0.398 \pm 0.062$ | $0.063 \pm 0.071$ | $3.491 \pm 3.198$ |
| | UniGradICON$^\dagger$ | $0.359 \pm 0.044$ | $0.045 \pm 0.056$ | $2.992 \pm 0.670$ |
| | VFA$^\dagger$ | $0.714 \pm 0.066$ | $0.281 \pm 0.216$ | $0.821 \pm 0.300$ |
| | Ours | $0.895 \pm 0.029$ | $0.597 \pm 0.204$ | $0.216 \pm 0.098$ |

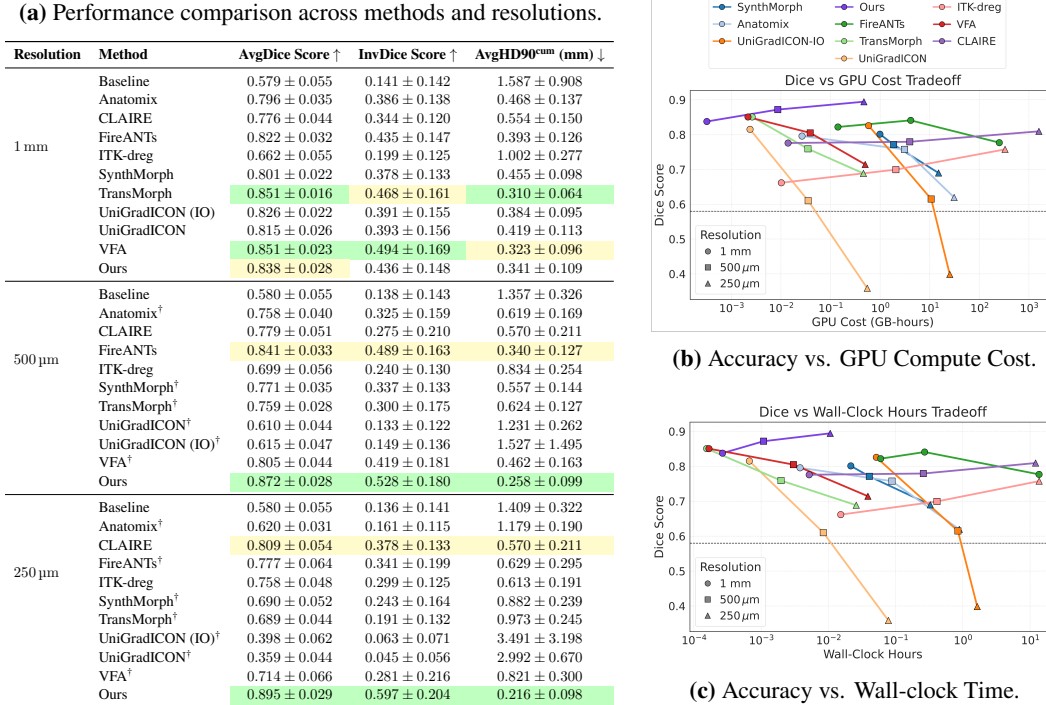

**(b)** Accuracy vs. GPU Compute Cost.

**(c)** Accuracy vs. Wall-clock Time.

**Figure 5:** Registration performance on Faux-OASIS dataset at $1\,\mathrm{mm}$, $500\,\mathrm{µm}$, and $250\,\mathrm{µm}$ (native $250\,\mathrm{µm}$); mean $\pm$ std over pairs. ↑ higher is better; ↓ lower is better. HD90 values are reported using our cumulative definition (see Sec. K.2).  (Green)/  (Yellow) = best/second; $^\dagger$ = patch-based

problem sizes for sufficiently large $H$. The detailed derivation and correctness of this operation is shown in Section I.

### 4.3 DISTRIBUTED LOSS FUNCTIONS

Since the moved image and fixed image are sharded cross $H$ hosts, the loss function must take this into account to compute the loss function correctly.

**Mean Squared Error (MSE).** Since MSE is a per-pixel loss, we compute the individual MSE on host $h$ and perform an `allreduce` operation.

**Localized Normalized Cross Correlation (LNCC).** The LNCC computes per-pixel patch similarities for each pixel, using a convolution over its neighbors. For sharded images, the patch statistics at the boundary requires a boundary synchronization with its neighboring shards which is provided by our GP implementation. After computing the LNCC for all pixels in each shard, we perform another `allreduce` to compute the LNCC over the entire image.

**Mutual Information (MI).** The MI loss computes the joint histograms $p_{(I,J)}(x, y)$ and marginals $p_I(x), p_J(y)$. However, these distributions are partial aggregates from the sharded images on each GPU. Eq. (2) can be rewritten as $p_I(v) = \sum_h \frac{N_h}{N} \left( \frac{1}{N_h} \sum_{k \in \Omega_h} \kappa(v - I_k) \right), p_{IJ}(v, w) = \sum_h \frac{N_h}{N} \left( \frac{1}{N_h} \sum_{k \in \Omega_h} \kappa(v - I_k)\kappa(w - J_k) \right)$, where the red terms correspond to the per-host histogram computation. Performing an `allreduce` to compute the weighted average of these histograms (with weights $N_h/N$) results in a valid and correct joint and marginal distributions over all hosts. This also leads to only a $B^2 + 2B$ communication overhead regardless of $N$, making a distributed implementation highly practical.

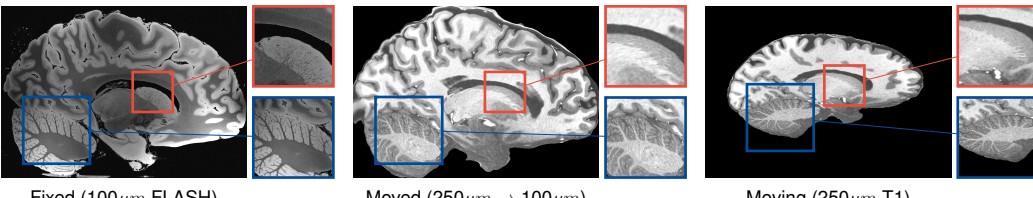

Fixed (100$\mu m$ FLASH)     Moved (250$\mu m \rightarrow$ 100$\mu m$)     Moving (250$\mu m$ T1)

**Figure 6:** Qualitative comparison on registration of 100$\mu m$ ex-vivo brain MRI (T1 $\rightarrow$ FLASH) image. Fine details like cerebellar white matter are not visible at macroscopic scales, but are aligned at 100$\mu m$. Fixed image is of size 1760 $\times$ 1760 $\times$ 1278. Best viewed zoomed in. More results in Fig. 11.

## 5 EXPERIMENTS

Our primary goals are to (a) accelerate both optimization and neural network based registration workflows, and (b) solve significantly larger image registration problems. We show the efficacy of our method by accelerating existing registration workflows on standard clinical data. This is followed by optimizing a multimodal registration task with more than 11.8B optimizable parameters, an unprecedented result in large-scale registration. We compare the performance and computational efficiency of our method with various state-of-the-art baselines on a simulated 250$\mu m$ ex-vivo brain MRI dataset, followed by ablations on various components of our framework.

**Baselines.** To accelerate existing registration workflows, we compare against TransMorph (Chen et al., 2022) and FireANTs (Jena et al., 2024a), which are state-of-the-art deep learning and optimization based registration frameworks respectively. In addition, we perform comparative evaluation with two methods explicitly designed for large-scale registration: ITK-DReg (itk) (CPU-based) and CLAIRE (Mang et al., 2019) (multi-GPU), and several SOTA learning-based approaches for clinical data - SynthMorph (Hoffmann et al., 2021), Vector-Field Attention (Liu et al., 2024c), unigradICON (Tian et al., 2024) (with/without instance optimization), anatomix+ConvexAdam (Dey et al., 2025).

### 5.1 ACCELERATING EXISTING REGISTRATION WORKFLOWS AND ABLATIONS

For deep networks, we train `TransMorph-large` under three loss configurations: (a) LNCC+Dice, (b) MI+Dice, and (c) LNCC+scaling-and-squaring (Ashburner, 2007) +Dice. For each configuration shown in Table 1, we either use the vanilla PyTorch implementation (Baseline) or our kernels (Ours). For classical optimization, we benchmark runtime and memory against multiple LNCC backends (FireANTs, VoxelMorph/TransMorph, Fast LNCC, `torch.compile`, and Ours) and MI backends (PyTorch and Ours with and without `torch.compile`). Tables 1 and 4 and Fig. 12 show that during network training our kernels converge 6.1$\times$ faster with LNCC while using 16.5% less memory, and reduce MI memory usage by 24.7%. Despite being designed for very large images, the runtime and memory benefits are significant for clinical-scale data (i.e., 30MB for OASIS). Optimization frameworks see larger gains: FireANTs achieves up to 95.2% memory savings and 2.6$\times$ speedup with MI, and a 7.5$\times$ speedup over FastLNCC (Jia et al., 2025) (and 2.9$\times$ over FireANTs' LNCC backend which applies separable convolutions on FastLNCC), with 44-59% lower memory usage overall.

### 5.2 REGISTRATION TO A 100 MICRON EX-VIVO BRAIN MRI VOLUME

To showcase the efficacy of our method on real large scale images, we register a 250$\mu m$ in-vivo MRI image (Lüsebrink et al., 2017) to a 100 $\mu m$ *ex-vivo* FLASH human brain volume (Edlow et al., 2019). This represents an inverse problem with more than 11.2B optimizable parameters (compared to ~20M for clinical datasets), or 44.8GB of GPU memory. The entire problem does not fit on most GPUs, necessitating distributed multimodal registration. We optimize a composite transform - affine followed by a diffeomorphic mapping; details can be found in Section E.1. Multimodal deformable registration took ~58 seconds on 8 NVIDIA A6000 GPUs, which is unprecedented at this resolution. Fig. 6 shows qualitative results, highlighting the ability to register highly detailed structures such as cerebellar white matter; these structures are not visible at macroscopic scales. The resultant advantages of performing registration at this scale can allow researchers to characterize the

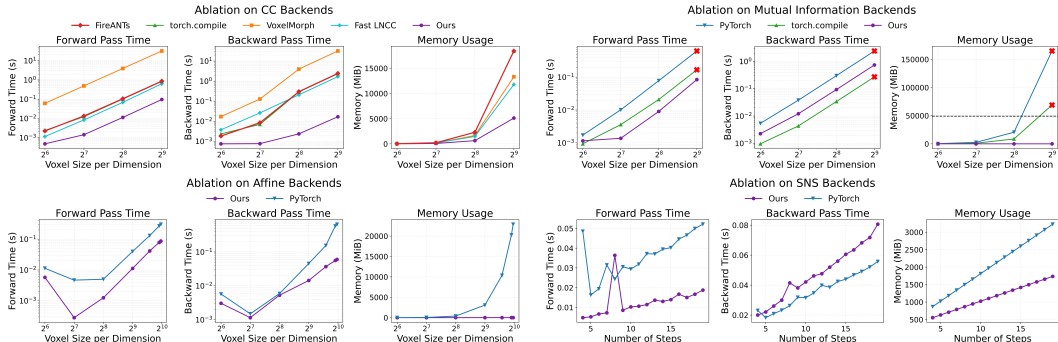

**Figure 7:** Ablations on key workhorse operations: LNCC, MI, `grid_sampler`, and scaling-and-squaring operations. Our fused kernels consume significantly less HBM and runtime.

neuroanatomy at microscopic resolutions and allow morphometric analysis of cortical layers and subcortical nuclei among other structures.

Registration accuracy in these studies is measured using privately annotated fiducial markers, hindering reproducibility and comparability of methodological advances. Due to lack of scalable frameworks, most high-resolution studies simply run ANTs at a significantly downsampled resolution (Kleven et al., 2023; Mansour et al., 2025; Wang et al., 2020b; Kronman et al., 2024; Bogovic et al., 2020; Edlow et al., 2019) and upsample the warp field to the native resolution.

### 5.3 Comparative Analysis on a Simulated ex-vivo Brain MRI Dataset

**The faux-OASIS dataset** To compare registration performance at high resolutions and leverage existing methods as baselines, we synthesize the *faux-OASIS* dataset, which mimics the anatomical distribution of an MRI dataset at $250\mu m$ isotropic resolution (more details in Section K). At $250\mu m$, the deformation field has 1.32B degrees of freedom per image pair, compared to $\sim$20M for OASIS.

**Baselines and evaluation.** All methods (including CLAIRE and FireANTs without FFDP) run out of memory at $250\mu m$ resolution. We proposed two modifications

**Table 1:** Accelerating TransMorph (**Top**) and FireANTs (**Bottom**) training with various computation backends.

| Variant | Loss | Diffeomorphic | Training Time (h) | GPU Mem (GB) | Val DSC |
|---|---|---|---|---|---|
| Baseline | LNCC | ✗ | 171.20 | 20.01 | 86.74 |
| Ours | LNCC | ✗ | 27.84 | 16.95 | 87.23 |
| Baseline | LNCC | ✓ | 171.42 | 21.28 | 86.55 |
| Ours | LNCC | ✓ | 27.93 | 17.34 | 87.09 |
| Baseline | MI | ✗ | 26.09 | 22.34 | 86.74 |
| Ours | MI | ✗ | 24.94 | 16.80 | 86.80 |

| Loss | Backend | Dice Score ↑ | Runtime (s) ↓ | Memory (MB) ↓ |
|---|---|---|---|---|
| LNCC | FireANTs | 78.81 ± 3.87 | 1.44 ± 0.08 | 1044.5 ± 0.0 |
| LNCC | FastLNCC | 76.96 ± 3.60 | 3.76 ± 0.16 | 1026.3 ± 0.0 |
| LNCC | VXM/TM | 76.96 ± 3.60 | 57.08 ± 2.45 | 1418.5 ± 0.0 |
| LNCC | `torch.compile` | 69.35 ± 4.09 | 0.82 ± 0.04 | 860.7 ± 0.0 |
| LNCC | Ours | 78.67 ± 3.04 | 0.50 ± 0.01 | 577.5 ± 0.0 |
| MI | PyTorch | 75.88 ± 3.45 | 7.51 ± 0.37 | 12206.3 ± 0.0 |
| MI | `torch.compile` | 75.88 ± 3.45 | 1.05 ± 0.05 | 3865.5 ± 0.0 |
| MI | Ours | 75.88 ± 3.44 | 2.90 ± 0.16 | 577.5 ± 0.0 |
| MI | `torch.compile`+Ours | 75.93 ± 3.47 | 2.95 ± 0.16 | 657.3 ± 0.0 |

to deep learning based methods to enable them to work on this dataset: (a) inspired by several high-resolution studies (Wang et al., 2020b; Mansour et al., 2025; Edlow et al., 2019), we register the images at a downsampled resolution, and then upsample the deformation field (b) inspired by several histology registration methods (Wodzinski et al., 2024; Lotz et al., 2015; Liang et al., 2021), we perform patchwise registration and mosaicing of the final deformation. We compare the methods at three resolutions: 1mm, $500\mu m$, and $250\mu m$. At 1mm, the full image fits within a patch, providing a baseline reference comparable to reported OASIS performance. At higher resolutions, patches are defined by each method's default input size with stride equal to 50% of the patch size. FireANTs augmented with FFDP is denoted as *Ours*. We report Dice, inverse-weighted Dice (InvDice; Mang et al. (2019)), and average Haussdorf distance capped at 90 percentile (AvgHD90). To compare efficiency, we measure both wall-clock time and GPU-hours.

**Results.** Fig. 5a summarizes performance metrics. At 1mm, most methods achieve performance consistent with their reported performance on OASIS, including VFA and TransMorph which were trained on the OASIS dataset with label supervision. At higher resolutions, nearly all methods degrade, especially for InvDice and HD90, which emphasize alignment of fine structures. In contrast, our method improves in accuracy: at $250\mu m$, we improve Dice by 18.1 points, InvDice by 31.6 points, and reduce AvgHD90 by 62.1%. The correlation between resolution and performance is also

observed in (Mang et al., 2019; Mang & Ruthotto, 2017; Nazib et al., 2018); in addition we verify that patch-based methods *degrade* in performance at higher resolutions.

**Figure 8:** Scaling and GP ablations.

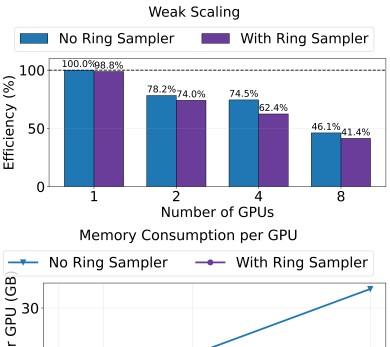

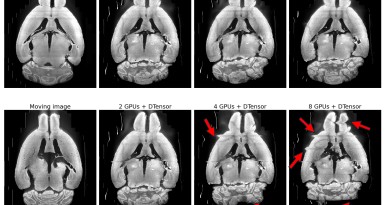

**(a)** Weak scaling and Per-GPU memory consumption of FFDP.

**(b)** Qualitative ablation of GP synchronization in FFDP on the fMOST mouse brain dataset (Tustison et al., 2024). Red arrows highlight regions affected by incorrect boundary effects due to no GP. See Fig. 10 for more examples.

This degradation among patchwise methods is expected; histology-style pipelines typically register consecutive slides with small deformations after affine alignment. At high resolution, patching reduces anatomical context and the patches become progressively more out-of-distribution (see Fig. 19). Patchwise or downsampling strategies are therefore insufficient for ultra-high resolution large-scale registration, and existing deep methods cannot be repurposed to work at higher resolutions efficiently. Accuracy-efficiency tradeoffs in Figs. 5b and 5c show that our method is Pareto-efficient compared to all other methods (CPU, deep learning, and distributed GPU methods), requiring up to $500\times$ fewer GPU-hours compared to alternatives at $250\mu m$.

### 5.4 ABLATION STUDIES

We ablate on the efficiency of various workhorse operations used in image registration in Fig. 7 and Table 3. We compare our implementations to community-standard PyTorch implementation (Jia et al., 2025; Chen et al., 2022) and `torch.compile` versions. For grid sampler and MI kernels, our kernels have $O(1)$ extra HBM overhead instead of $O(N)$ in the PyTorch implementation. For LNCC, our implementation achieves an average speedup in the forward pass by $5.22\times$ and $56.98\times$ in the backward pass. Our `grid_sampler` also leads to an efficient scaling-and-squaring operation, commonly used in deep learning registration pipelines (Chen et al., 2022), with a memory reduction of $50\%$ compared to the baseline implementation. **Scalability Analysis.** We test the weak scaling of our distributed framework by registering synthetic images with increasing voxel sizes. For $H$ GPUs, we instantiate an image pair of size $700 \times 700 \times 700H$ and shard the images, warp, and optimizer state across $H$ GPUs. Fig. 8a shows weak scaling of FFDP with and without ring sampler. Without the ring sampler, the `grid_sample` operation requires storing the moving image of size $700 \times 700 \times 700H$ on each GPU, leading to peak HBM memory increasing linearly with $H$. This implies the framework would not scale to arbitrarily large problem sizes, regardless of cluster size $H$. Peak Memory consumption is independent of $H$ with the Ring Sampler, and scaling efficiency is only minimally affected. **Ablation on GP.** We ablate the effect of GP by replacing it with DTensor sharding (no boundary sync). Figs. 8b, 9 and 10 show that incorrect boundary synchronization leads to undesirable artifacts in the moved images, and reduces labelmap overlap.

## 6 CONCLUSION

We propose a novel distributed framework for arbitrarily large image registration problems. Our work identifies and proposes IO-aware and distributed-friendly implementations of workhorse operations in image registration algorithms, enabling registration of images at arbitrarily large resolutions on a single GPU. Our fused primitives demonstrate compelling results in both improving existing registration pipelines and scaling to arbitrarily large, multimodal problems pertinent in modern life science applications, that were previously infeasible without approximations. FFDP shows unprecedented registration capabilities that will enable researchers to leverage and effectively work with large-scale image volumes and unearth new insights leveraging the large resolution images.

ACKNOWLEDGMENTS

This work was supported by the National Institutes of Health (NIH) under grants EB031722 and NS135568.

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
