## A    APPENDIX

## B    RELATED WORKS

### B.1    MEMORY EFFICIENT AND LARGE SCALE OPTIMIZATION

Recent advances in large scale transformer-based model training has amassed significant attention and efforts to alleviate key bottlenecks in both memory and compute efficiency. Activation memory forms a key bottleneck in many deep learning training pipelines, and recent advances propose fused operations (Dao et al., 2022; Dao, 2023; Shah et al., 2024; PyTorch, 2023; Bikshandi & Shah, 2023; Dong et al., 2024) to significantly reduce HBM usage without approximations. Other techniques propose sub-quadratic approximations to the quadratic complexity of the attention operation and propose highly efficient and IO-aware fused kernels (Yuan et al., 2025; Dong et al., 2024; Wang et al., 2024). However, as these models and their inputs get increasingly larger in size, they do not fit on a single GPU. Various distributed techniques like Tensor Parallel (Shoeybi et al., 2019), Sequence Parallel (Li et al., 2023; Jacobs et al., 2024; Li et al., 2024), pipeline parallel (Qi et al., 2023; Lamy-Poirier, 2023; Liu et al., 2024a), fully-sharded data parallel (FSDP2) (Ansel et al., 2024; Zhao et al., 2023; Rajbhandari et al., 2020) have been proposed that distribute (shard) the model and its inputs across multiple GPUs for transformer-like models. Another research area approaches the problem of scaling large models by building compilers and intermediate representations to enable writing optimized kernels at runtime OpenAI (2021); Ansel et al. (2024); Spector et al. (2025); Chen et al. (2018); Abadi et al. (2016). To our knowledge, most of these techniques are tailored to transformer-specific architectures and GEMM-like operations (self attention, feedforward, batchnorm, etc.) only, and a Tensor/Model Parallel variant for convolution-aware sharding is not available. However, other disciplines including biomedical and clinical imaging, life sciences, climate modeling, drug discovery, genomics, geosciences, robotics leverage other key components that do not fit in the transformer-specific framework, or are GEMM-like in nature. We focus on the compute and memory bottlenecks in the image registration problem, that is a key component in a variety of biomedical and life science applications.

### B.2    LARGE SCALE REGISTRATION IN LIFE SCIENCES AND BIOMEDICAL IMAGING

#### B.2.1    *EX-VIVO* NEUROIMAGING AND HISTOLOGY FOR NEUROANATOMICAL AND PATHOLOGICAL STUDIES.

A large body of neuroanatomical studies are performed in conjunction with ex-vivo and blockface imaging and histology to create detailed, multi-scale anatomical references by integrating structural, molecular, and cytoarchitectural information across imaging modalities (Casamitjana et al., 2025; Ravikumar et al., 2024). In-vivo MRI scans are typically limited by resolution due to constraints on scan time and motion artifacts associated with longer scan times. This makes in-vivo MRI scans unsuitable for studying the microstructural changes associated with neurodegenerative disease progression. Registration of fine anatomical details like cortical layers, axonal projections, or individual nuclei are useful to understand neuropathology, and such analyses are not possible at macroscopic clinical scales. Therefore, high-resolution ex-vivo scans and blockface imaging are used as a bridge between in-vivo and histology, with the latter used as a gold standard for ground-truth microscopic tissue characterization and pathology. Many complementary stains are used to visualize neuropathological features, including protein aggregates, neuronal loss, gliosis, and myelin integrity. Accurate registration of these structures is important to improve our understanding of morphological effects of pathology. For example, Alzheimer's Disease (AD) is characterized by cortical atrophy in the medial temporal lobes, particularly hippocampus, entorhinal cortex, and parahippocampal gyrus (Ravikumar et al., 2024; Echávarri et al., 2011). Accurate atrophy quantification of these structures can only be reliably performed at ~0.5 mm or better resolution MRI or ex vivo imaging, necessitating high resolution registration. Parkinson's Disease (PD) is characterized by degeneration of DA neurons in the substantia nigra (Triarhou, 2013) and subthalamic nucleus that are small (~5-10 mm), requiring <0.7 mm isotropic or ex vivo imaging for volumetry or susceptibility mapping for accurate delineation (Welton et al., 2023). Multiple Schelosis is characterized by cortical lesions (Madsen et al., 2021; Beck et al., 2018) that cannot be delineated at the in-vivo resolution and typically requires high resolution ex-vivo imaging and histopathology integration. Except in-vivo imaging, all other modalities are very high resolution typically ranging from $500\mu m$ up to $100\mu m$ (Ravikumar et al., 2024; Echávarri et al., 2011; Welton et al., 2023; Madsen et al., 2021) for ex vivo imaging

and $\sim 10 \mu m$ for histology sections. High-resolution imaging and registration are essential in these contexts because they enable accurate cross-modal alignment and preservation of fine anatomical detail that would otherwise be lost through downsampling. Most of these studies, however, limit their analyses to localized effects due to the significant computational cost of registering the entire brain at high resolution. Recently, projects like Allen Brain Atlas (all) and multiple large scale consortia including Seattle Alzheimer's Disease Brain Cell Atlas (SEA-AD) consortium and the Human Mouse Brain Atlas (HMBA) consortium are aimed at creating detailed, multimodal brain atlases linking cellular, molecular, and anatomical organization across species and disease states - combining individual efforts from multiple institutions together into a unified resource. Achieving this multimodal organization at the whole brain level requires high-resolution registration tools to accurately align diverse imaging modalities while preserving fine-scale cytoarchitectural detail.

Most in-vivo to histology registration workflows require registration of an in-vivo image to its ex-vivo counterpart. The $100 \mu m$ ex-vivo and $250 \mu m$ in-vivo images released in Edlow et al. (2019); Lüsebrink et al. (2017) are intended to be used as high-resolution templates to enable accurate studies, but lack of computationally efficient methods restricts their broad usage in the neuroimaging community. Our paper performs a native-scale registration on a modest GPU server with 8 A6000 GPUs to showcase the distributed capabilities, democratizing the use of such high resolution data for advancing the state of neuropathology studies.

### B.2.2 High resolution pulmonary imaging enables subvoxel landmark localization.

Pulmonary CT mapping is a key component of lung disease diagnosis and treatment, and accurate landmark tracking requires registration at high resolution. Lung CT images can be acquired at submillimeter resolution (Murphy et al., 2011), but deep learning methods often require downsampling to accommodate their memory requirements (Falta et al., 2023; Hering et al., 2020). In the LungCT Learn2Reg challenge, the Lung CT images have a resolution of 1.25-1.75mm and the top performing methods achieve an average landmark error of 1.83mm. However, in the EMPIRE10 challenge, the average physical resolution of the images is 0.7mm and the average landmark error of most top methods (FireANTs, DISCO) is around 0.649mm, reaching subvoxel landmark localization. This demonstrates that with an appropriately high resolution, top methods can achieve subvoxel accuracy in landmark errors without learning. Moreover, due to the large voxel sizes in the EMPIRE10 dataset (with average voxel size of $412.8 \times 317.2 \times 364.9$, about $5\times$ larger than OASIS brain MRI on average), most top performing methods are iterative methods, sometimes used in conjunction with patch based feature extractors. This retrospective analysis shows the direct impact of using higher resolution to improve landmark accuracy in pulmonary imaging, and the benefits of using native-scale registration.

### B.2.3 Large scale registration in model organisms.

Over the past decade, imaging across the life sciences and biomedical domains has progressed from mesoscale surveys to organ- and organism-wide acquisitions at cellular or even subcellular resolution. These span transparent organisms and small animal models (e.g., *C. elegans*, zebrafish, adult *Drosophila*) (Varol et al., 2020; Venkatachalam et al., 2016; Marquart et al., 2017; Gupta et al., 2018; Peng et al., 2011; Brezovec et al., 2024), whole-rodent brains imaged at micron or submicron sampling (Gong et al., 2016; Wang et al., 2020a), and non-human primate (NHP) and human ex vivo MRI at hundreds of microns (Skibbe et al., 2023; Milham et al., 2018; Edlow et al., 2019; Lüsebrink et al., 2017). Such modalities routinely generate giga- to teravoxel volumes (Kutten et al., 2016; Nazib et al., 2018). Their scientific utility, however, hinges on the ability to perform registration at the native resolution of acquisition, i.e. aligning specimens (or modalities) in a common coordinate system without sacrificing the fine-scale morphologies-cell bodies, layers, axon bundles, synaptic neighborhoods– that motivate high-resolution acquisition in the first place (Nazib et al., 2018; Goubran et al., 2013).

**Cellular-resolution atlases in model organisms.** In *C. elegans*, statistical atlases of neuron positions require aligning whole-animal volumes to preserve the fidelity of closely apposed cells (Varol et al., 2020; Venkatachalam et al., 2016). In zebrafish, deformable registration with cellular-level precision and minimal perturbation of tissue morphology enables pooling of gene expression, single-neuron morphologies, and brain-wide activity (Marquart et al., 2017; Gupta et al., 2018). In adult *Drosophila*, whole-brain registration underpins large-scale databases and enables structure–function integration

(for example, aligning two-photon functional volumes to EM-derived connectomes) (Peng et al., 2011; Brezovec et al., 2024).

**Whole-brain rodent imaging**   Large scale efforts like NIH's Brain Research through Advancing Innovative Neurotechnologies (BRAIN) Initiative - Cell Census Network (BICCN) aims to provide researchers and the public with a comprehensive reference of the diverse cell types in human, mouse, and non-human primate brain, and researchers collect a wide range of multimodal data including MRI, sectioning tomography, microscopy, antibody stains (e.g. calbindin), and spatial transcriptomics. In rodents, fMOST pipelines yield whole-brain images at micron sampling (e.g., $0.32\mu m$ voxels generating >10TB datasets) for tracing long-range axons and quantifying cytoarchitecture (Gong et al., 2016). Constructing stereotaxic spaces such as the Allen CCFv3 and Waxholm rat atlas requires deformable registration that preserves layers and boundaries (Wang et al., 2020a; Kleven et al., 2023; Kronman et al., 2024). Currently, there is a huge gap between the resolution at which data is acquired and the resolution at which templates are created. For example, STPT images can be collected at less than $1\mu m$ resolution (Liwang et al., 2023), but the Allen CCFv3 template is generated at $10\mu m$ by upsampling the registrations from $25\mu m$ due to compute constraints. Extrapolating the runtime reported in the method used to generate the CCFv3 template (Wang et al., 2020a), registration will require about 19 hours for a single pair or about 7.26 CPU-years for a single iteration of template matching - and is therefore impossible to curate without access to huge HPC clusters. This is contrasted to our method that can perform registration in about a minute or two on a modest server rack with 8 GPUs (or 5.48 GPU days for a single iteration of template building) - saving a significant amount of time and resources. Certain phenomena of interest like cellular organization and brain-wide connectomes are emergent only at very high resolutions, necessitating computational tools that can scale with the data.

The lack of computational tools for large-scale registration has a trickle-down effect on follow up studies as well. For instance, ANTsX pipelines for mouse brain registration to the CCFv3 atlas is performed at $50\mu m$ instead of $10\mu m$ for compute reasons (Tustison et al., 2024). Developmental atlases also register the CCFv3 at resolutions significantly downsampled from the original $10\mu m$ template (Kronman et al., 2024; Liwang et al., 2025) citing lack of computational resources as one of the primary reasons.

**Zebrafish**   Initially adopted as a developmental biology model because of its ease of domestication, high fecundity, and transparent early life stages, the zebrafish has gained broader prominence with advances in brain imaging, molecular genetic tools, and behavioral assays (Kenney et al., 2021). For the AZBA template (Kenney et al., 2021), the raw images are collected at $4\mu m$ but was resampled to $8\mu m$ ($8\times$ downsampling) due to system constraints. The tools used for the registration (Friedel et al., 2014) do not recommend running locally and only on a distributed cluster. Brain-wide cellular resolution imaging of transgenic zebrafish lines (Tabor et al., 2019) is performed on large clusters like Biowulf Linux cluster at the National Institutes of Health, significantly reducing accessibility of these imaging resources to researchers, signifying an unmet need for efficient and distributed multimodal registration frameworks.

Across these diverse domains, the unifying requirement demands access to scalable multimodal registration algorithms - a challenge we address in this work.

### B.3   DEFORMABLE IMAGE REGISTRATION

Given two images $F : \Omega \to \mathbb{R}^d$ and $M : \Omega \to \mathbb{R}^d$ defined on domain $\Omega$ (usually a compact subset of $\mathbb{R}^d$), Deformable Image Registration (DIR) refers to an inverse problem to find a transformation $\varphi : \Omega \to \Omega$ that warps the moving image $M$ to the fixed image $F$. Prior to deep learning, the inverse problem was solved using iterative solvers (Klein et al., 2009; Tustison & Avants, 2013; Andersson et al., 2007; Ashburner, 2012; Avants et al., 2006), and has been made significantly more scalable by recent advances in GPU-based libraries (Mang et al., 2019; Mang & Ruthotto, 2017; Jena et al., 2024a). Meanwhile, earliest deep learning for image registation (DLIR) methods (Cao et al., 2017; Krebs et al., 2017; Rohé et al., 2017; Sokooti et al., 2017) used supervised learning to predict a transformation field using pseudo ground truth transformations. However, since the inverse problem is generally ill-posed, unsupervised and weakly supervised learning methods (Balakrishnan et al., 2019; Zhao et al., 2019b;a; Joshi & Hong; De Vos et al., 2019; Mok & Chung, 2020; Zhang et al.,

2021; Qiu et al., 2021; Lebrat et al., 2021; Jia et al., 2022; Mok & Chung, 2022) became dominant. However, these methods perform virtually identically to iterative solvers in the unsupervised setting, and show relatively brittle performance under domain shift(Jena et al., 2024b; Jian et al., 2024; Jena et al., 2025). Other recent work has shown increased reliability under domain shift (Liu et al., 2024c; Chen et al., 2022). Another line of work combines neural priors with iterative solvers to leverage learnable features with strong convergence properties and robustness of solvers (Wu et al., 2024; Hu et al., 2024; Wu et al., 2022; Zhao et al., 2019a;b). However, almost all deep learning-based methods typically work reliably only at a macroscopic resolution, with most methods working only at a standard resolution of 1mm or $192 \times 160 \times 224$ voxels, and running out of memory on larger images, even on other macroscopic problems like lung or full body CT unless they are significantly downsampled (Falta et al., 2023). This is a significant limitation given modern real life applications including the ultra-high-resolution image acquisition techniques used for *ex-vivo* neuroanatomical and developmental biology studies, spanning subcellular structures and connectomes in species like C.elegans (Varol et al., 2020; Venkatachalam et al., 2016), zebrafish (Marquart et al., 2017; Gupta et al., 2018), adult Drosophila (Bogovic et al., 2020; Brezovec et al., 2024; Peng et al., 2011), rodents (Wang et al., 2020b; Mansour et al., 2025; Kleven et al., 2023; Kronman et al., 2024), and non-human primates (Skibbe et al., 2023; Davis & Maga, 2018; Frye et al., 2022). The scale of these problems is often two to three orders of magnitude larger than the scale of existing deep learning methods. A simple extrapolation shows that existing deep learning methods will require $\approx 1.87$ TB of GPU memory to train a model on a $250\mu m$ ex-vivo brain dataset, making them impractical for training on larger problems. (Mang et al., 2019; Mang & Ruthotto, 2017) propose a distributed framework for registering arbitrarily large images, but is limited to MSE loss function and a one-parameter subgroup of diffeomorphic transforms (stationary velocity field), which is less flexible than the entire space of diffeomorphic transforms (Mang et al., 2019; Jena et al., 2024a). Moreover, they show results on upto 256 GPUs which indicates room for improvement in terms of scaling efficiency. In our work, we propose a distributed framework that is upto an order of magnitude more efficient than (Mang et al., 2019) on large problems.

## C  LIMITATIONS AND FUTURE WORK

One of the limitations of the proposed framework is the relatively poor weak scaling of the method in the distributed setting (41% on 8 GPUs without NVLink or Infiniband). Even so, for most life science applications feasibility is the first step towards scalable, distributed, multimodal registration, and future work will focus on improving the weak scaling of the method. Another active avenue for future work is to enable Virtual GridParallel (VGP) to use fewer GPUs by sequentially offloading and onloading consecutive shards from CPU onto a single GPU. Deformable Registration of the 100 $\mu m$ volume in Appendix 5.2 took only *one minute* on 8 A6000 GPUs, but equivalently it would take around 15-20 minutes to register this pair on a single A6000 GPU with VGP, accounting for repeated CPU offloading. This is an acceptable timeframe for large-scale studies, allowing researchers to prototype and iterate on large-scale image volumes as well with a single GPU. Other avenues for future work include collecting labeled data at high-resolution for various real-world life science applications and performing comparative studies on these datasets.

## D  LLM USAGE

We use an LLM (minimally) to polish the manuscript and improve clarity of ideas and organization. All LLM-generated text is thoroughly reviewed, proofread, and revised by the first author of the paper.

## E  CORRECTNESS OF GRID PARALLEL IMPLEMENTATION

The GridParallel framework aims to add additional synchronization primitives for performing mathematically correct convolutions across image or grid shards. These convolutions are required to compute the LNCC loss, and applying Sobolev preconditioning of the warp field and its gradient. Without GP synchronization, the implementation is equivalent to a DTensor sharding (Ansel et al., 2024). To our knowledge, existing Model Parallel and FSDP techniques are exclusively built for model weights and activations for linear and self.-attention layers and do not support this functionality. The pseudocode for convolution with GP synchronization is provided in Algorithm 1.

To ablate the effect of GridParallel synchronization, we register images at $500\mu m$ resolution from the faux-OASIS dataset with and without the GridParallel synchronization to measure the effect on performance. Results in Fig. 9 show only a minimal drop in performance with DTensor sharding. We posit that this is because the faux OASIS dataset does not contain real-world noise and other artifacts that can degrade performance with incorrect boundary synchronization. To study the effect of GP synchronization on a more challenging dataset, we register images at $10\mu m$ resolution from the fluorescence micro-optical sectioning tomography (fMOST) mouse brain dataset (Tustison et al., 2024) with and without the GridParallel synchronization to measure the effect on performance. This dataset contains image volumes of size $1202 \times 1078 \times 627$ voxels, or a displacement field of 9.74GB. The data contains a myriad of complex artifacts, namely stripe artifacts, boundary halo effects, and speckle noise from image stitching and reconstruction. We run FireANTs with multi-scale optimization at scales 16, 8, 4, 2, $1\times$ downsampling for 200, 200, 200, 100, 50 iterations. We use our Fused LNCC implementation with a window size of 7, and a learning rate of 0.5. Smoothing regularizations are set to $\sigma_{\text{grad}} = 1.0$ and $\sigma_{\text{warp}} = 0.5$. We ablate on 2, 4 and 8 GPUs.

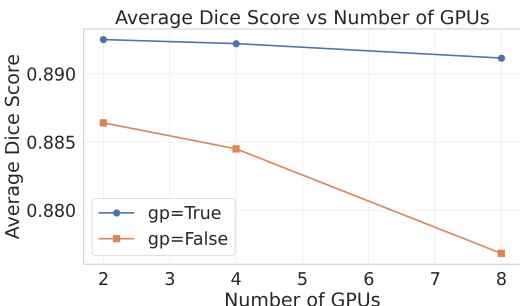

**Figure 9:** Quantitative ablation of GP synchronization on the faux-OASIS dataset.

Since we do not have ground truth annotations for this dataset, we only make qualitative observations. Unlike the faux-OASIS dataset, the fMOST dataset is more challenging with high levels of image heterogeneity and complex anatomical structures. Fig. 10 shows that the performance without GP synchronization is significantly affected as a function of GPUs. Specifically, the boundaries introduce undesirable artifacts due to mathematically incorrect smoothing and LNCC losses computed across shard. GP synchronization produces qualitatively better results regardless of the number of GPUs used to shard the problem.

---

**Algorithm 1** Convolution with GP synchronization

---

**Require:** $T$ (tensor), $r$ (rank), $k$ kernel size, $W$ kernel filter, sharding index $sh$, GP size $gp\_size$
1: pad $\leftarrow (k-1)/2$
2: bl $\leftarrow$ None
3: br $\leftarrow$ None
4: **if** $r > 0$ **then**
5:     bl $\leftarrow$ get_boundary$(r-1, \text{pad})$
6: **end if**
7: **if** $r <$ gp_size **then**
8:     br $\leftarrow$ get_boundary$(r+1, \text{pad})$
9: **end if**
10: $T_{\text{pad}} \leftarrow$ concat$([\text{bl}, T, \text{br}], \text{dim} = \text{sh})$
11: out $\leftarrow$ conv$(T_{\text{pad}}, W)$
12: crop_from_left $\leftarrow 0$
13: crop_from_right $\leftarrow 0$
14: **if** $r > 0$ **then**
15:     crop_from_left $\leftarrow$ pad
16: **end if**
17: **if** $r <$ gp_size **then**
18:     crop_from_right $\leftarrow$ pad
19: **end if**
20: out $\leftarrow$ crop$(\text{out}, (\text{crop\_from\_left}, \text{crop\_from\_right}), \text{dim} = \text{sh})$
21: **return** out

---

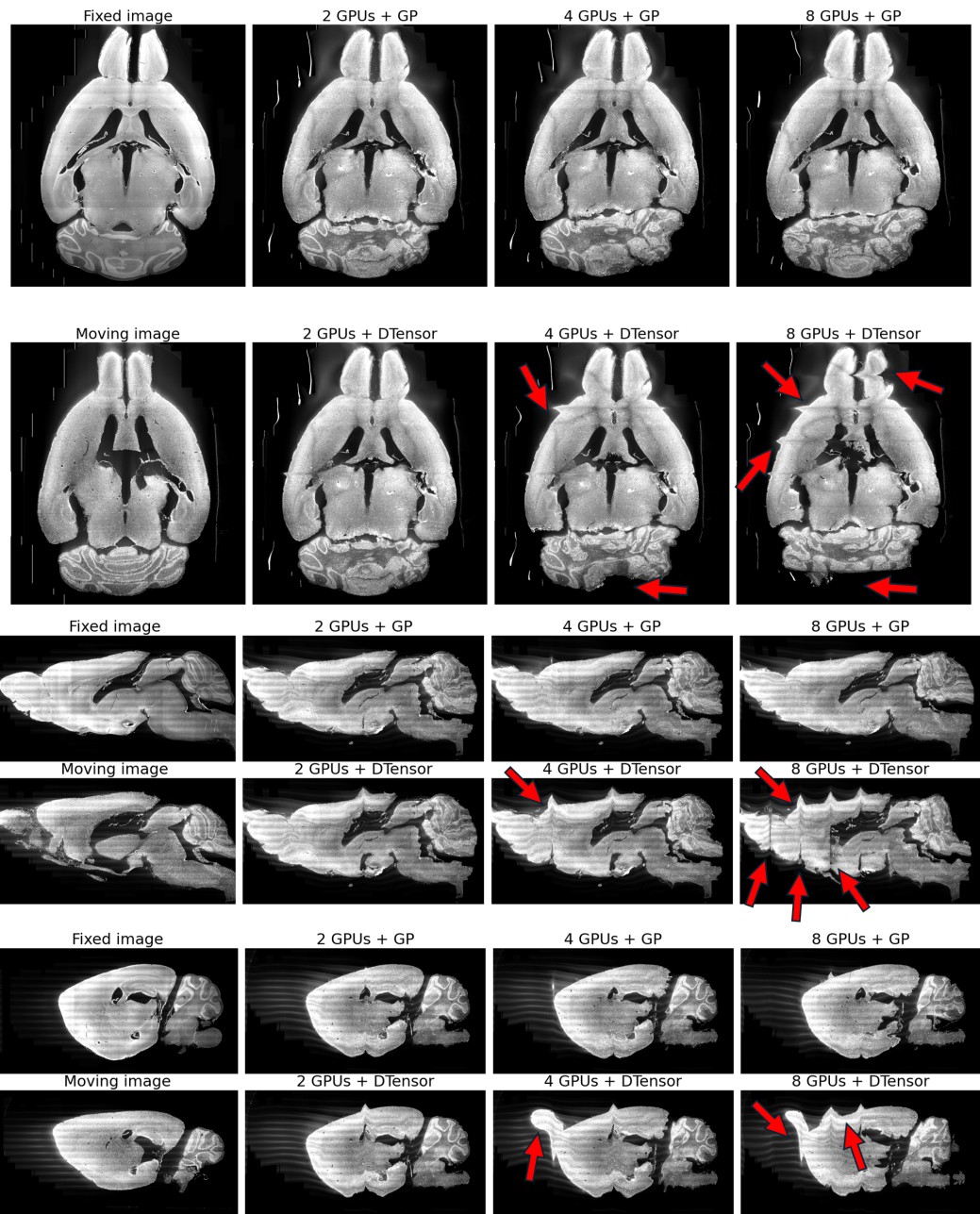

**Figure 10:** Qualitative ablation of GP synchronization in FFDP on the fMOST mouse brain dataset. Red arrows highlight regions affected by incorrect boundary effects due to no synchronization.

## F ACCELERATING TRANSMORPH TRAINING

In this section, we plot the performance of TransMorph training with and without our fused operations. Table 1 summarizes the performance of TransMorph training with and without our fused operations for three commonly used configurations. We further plot the validation performance across these settings with respect to Wall clock time in Fig. 12. Our fused operations demonstrate efficiency with fast convergence while reducing memory usage.

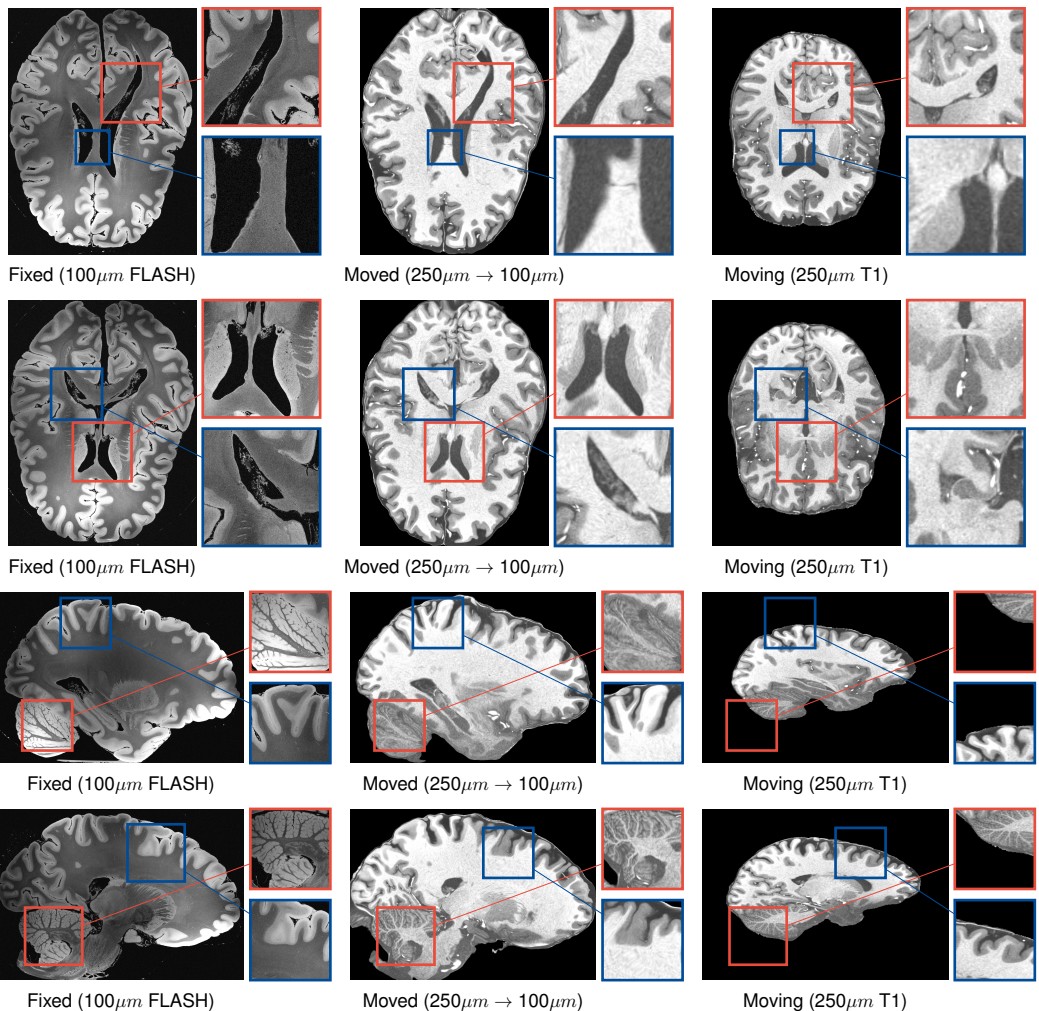

**Figure 11:** Qualitative comparison of registration results from $250\mu m$ T1 (Lüsebrink et al., 2017) to $100\mu m$ ex-vivo FLASH (Edlow et al., 2019). Intricate structures like cerebellar white matter and GM-WM interfaces are not very discernable at 1mm, but can be aligned at $100\mu m$ with our method.

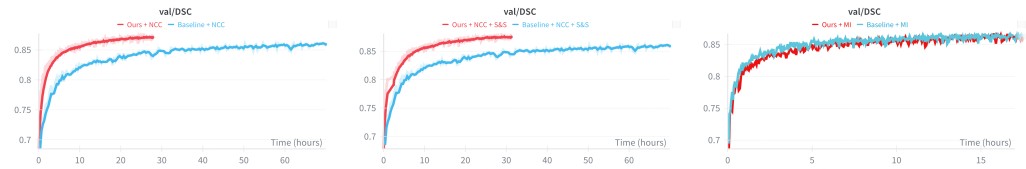

**Figure 12:** Ablation on TransMorph training runtime with and without our fused operations. For LNCC, our method converges in about 30 hours, while the baseline converges in about a week.

## F.1 REGISTRATION TO A 100 MICRON EX-VIVO BRAIN MRI VOLUME

In this section, we describe the parameters used for the registration of a $250\mu m$ in-vivo T1-weighted MRI volume described in Lüsebrink et al. (2017) to the $100\mu m$ ex-vivo brain FLASH volume described in Edlow et al. (2019). First, we perform an multi-scale affine registration at 3mm, 2mm, 1mm, $500\mu m$ resolutions for 500, 250, 100, 100 iterations respectively, using the Fused Mutual Information loss. This step takes about 12 seconds to run on a single NVIDIA A6000 GPU. The second step was to run multi-scale deformable optimization with scales 3.2mm, 1.6mm, 0.8mm, 0.4mm, 0.2mm, 0.1mm (scale factors of 32, 16,8, 4, 2, 1) for 250, 100, 100, 100, 50, 20 iterations

**Table 2: Qualitative Comparison of Methods**. We compare the methods on qualitative features such as GPU support, multimodal capabilities, ability to run for unequal sizes of fixed and moving images, non-standard image sizes, whether the model can work with full context for larger images, whether the model supports multi-GPU training, and whether the model supports arbitrary loss functions. Deep learning methods support multimodal registration only if they are trained on multiple modalities. CLAIRE requires the image sizes to be divisible by the number of GPUs, and does not support arbitrary loss functions. Our method supports all of the above features, leading to a seamless experience for users with minimal data preprocessing overhead.

| Method | GPU support | Multimodal | $N \neq M$ | Non-std sizes | Full Context | Multi-GPU | Supported Similarty functions |
|---|---|---|---|---|---|---|---|
| Deep learning | ✓ | ✓? | ✗ | ✗ | ✗ | ✗ | Fixed at training |
| ITK-DReg | ✗ | ✓ | ✓ | ✓ | ✗ | ✗ | ITK-filters |
| CLAIRE | ✓ | ✗ | ✗ | ✓? | ✓ | ✓ | MSE only |
| Ours | ✓ | ✓ | ✓ | ✓ | ✓ | ✓ | Any |

respectively, using the fused LNCC loss. This step took about 58 seconds on 8 NVIDIA A6000 GPUs. Qualitative results are shown in Fig. 11.

# G   AN EFFICIENT FUSED LOCAL NORMALIZED CROSS CORRELATION LOSS

Local Normalized Cross Correlation (LNCC) loss is ubiquitously used throughout the image registration literature(Liu et al., 2024c; Avants et al., 2008b; 2009; ANTsX; Jena et al., 2024a; Wu et al., 2024; Hu et al., 2024; Wu et al., 2022; Zhao et al., 2019a;b), owing to its robust behavior to unimodal and multimodal images alike. This operation is a key memory-bound bottleneck in image registration pipelines. Few approaches have been proposed to provide improved implementations (Jia et al., 2025; Chen et al., 2022), but we note that these implementations are still memory intensive and thus not scalable. We address this bottleneck by analytically deriving a fused implementation that is memory efficient and scalable.

**Definition of LNCC loss.**  Given two images $F$ and $M$, and a radially symmetric averaging convolution filter $W$ such that $\sum_k w_k = 1$ , we define the Local Normalized Cross Correlation (LNCC) loss as:

$$\mathcal{L} = \frac{1}{N} \sum_i n_i \quad , \quad n_i = \frac{A_i^2}{B_i C_i + \epsilon} \tag{3}$$

where

$$\mu_i^F, \mu_i^M = \sum_k w_{ik} F_k, \sum_k w_{ik} M_k \tag{4}$$

$$A_i = \sum_k w_{ik}(F_k - \mu_i^F)(M_k - \mu_i^M) \tag{5}$$

$$B_i = \sum_k w_{ik}(F_k - \mu_i^F)^2 \tag{6}$$

$$C_i = \sum_k w_{ik}(M_k - \mu_i^M)^2 \tag{7}$$

Here, we use overloaded notation $w_{ik} = w_{(i-k)} = w_{(k-i)} = w_{ki}$ due to radial symmetry of $w$. We can expand Eqs. (5) to (7) as follows:

$$A_i = \left( \sum_k w_{ik} F_k M_k \right) - \mu_i^F \mu_i^M = \mu_i^{FM} - \mu_i^F \mu_i^M \tag{8}$$

$$B_i = \left( \sum_k w_{ik} F_k^2 \right) - (\mu_i^F)^2 = \mu_i^{F^2} - (\mu_i^F)^2 \tag{9}$$

$$C_i = \left( \sum_k w_{ik} M_k^2 \right) - (\mu_i^M)^2 = \mu_i^{M^2} - (\mu_i^M)^2 \tag{10}$$

---

**Algorithm 2** Vanilla PyTorch LNCC implementation

---

**Require:** $F$ (input image), $M$ (reference image), $w$ (window size), $reduction$ (reduction type)
1: Define a radially symmetric convolution filter $W$ of size $w \times w \times w$ with $\sum W[i] = 1$
2: Define $\texttt{state} = (F, M, F^2, M^2, FM)$ ▷ Elementwise operations
3: **Compute** $\texttt{state} = W * \texttt{state}$ ▷ Convolution
4: Get $\mu_F = \texttt{state}[0]$ ▷ Local mean of $F$
5: Get $\mu_M = \texttt{state}[1]$ ▷ Local mean of $M$
6: Compute $\sigma_F^2 = \texttt{state}[2] - \mu_F^2$ ▷ Local variance of $F$
7: Compute $\sigma_M^2 = \texttt{state}[3] - \mu_M^2$ ▷ Local variance of $M$
8: Compute $\sigma_{FM} = \texttt{state}[4] - \mu_F \cdot \mu_M$ ▷ Local covariance of $F$ and $M$
9: Compute LNCC $= \frac{\sigma_{FM}^2}{\sigma_F^2 \sigma_M^2 + \epsilon}$ ▷ Add small $\epsilon$ to avoid division by zero
10: **if** $reduction$ == NONE **then**
11:     **return** LNCC
12: **else**
13:     Compute loss: Loss $= 1 - \text{mean}(\text{LNCC})$
14:     **return** Loss
15: **end if**

---

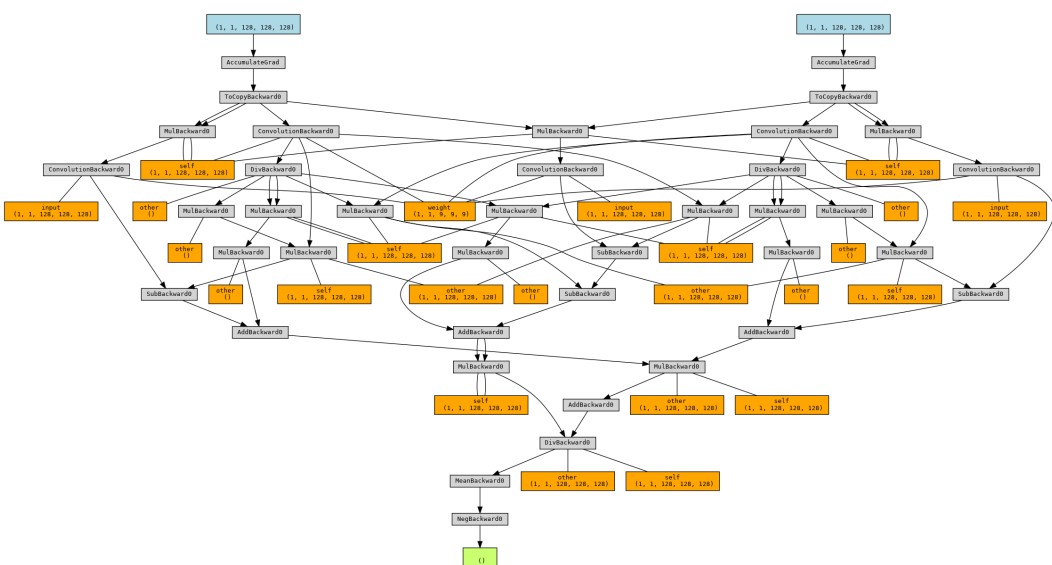

**Figure 13:** Computational graph of the vanilla PyTorch implementation of the LNCC loss function. Blue nodes denote the input images, Orange nodes denote intermediate tensors that are stored in HBM, Gray nodes denote operations on the computational graph, and Green node denotes the final loss. Orange nodes are the primary memory bottleneck.

Algorithm 2 outlines a vanilla PyTorch implementation of the LNCC loss function. The computational overhead of the algorithm arises due to many intermediates stored in high-bandwidth memory (HBM). Specifically, the quantities $W * \texttt{state}, I^2, J^2, IJ, \sigma_I^2, \sigma_J^2, \sigma_{IJ}, \mu_I^2, \mu_J^2, \mu_I \mu_J, \sigma_I^2 \sigma_J^2, (\sigma_I^2 \sigma_J^2 + \epsilon), \sigma_{IJ}^2, \sigma_{IJ}^2 / (\sigma_I^2 \sigma_J^2 + \epsilon)$ are all stored as intermediate tensors, each of size $N$, totalling a $16N$ memory overhead in addition to storing $\texttt{state}$. The computational graph of the vanilla PyTorch implementation is shown in Fig. 13. During the backward pass, the backprop algorithm computes the gradient with respect to each of these variables costing an additional $16N$ memory overhead. A $\texttt{torch.compile}$ implementation fuses some of the arithmetic, but leaves a lot of room for improvement (see Fig. 7). We present an algorithm that only requires an additional intermediate variable $\texttt{state}$ of size $5N$, saving upto $27N$ memory.

## G.1 AN EFFICIENT FUSED LNCC IMPLEMENTATION

During the forward pass, we initialize a $\texttt{state}$ variable of size $5N$. To minimize HBM reads from $F$ and $M$, we write a fused kernel to initialize the $\texttt{state}$ variable using only one HBM read from $F$

and $M$. The code in Line 4-9 are elementwise operations, and can be fused into another kernel. The forward pass therefore consumes only $5N$ additional memory. The pseudocode for the efficient fused LNCC implementation is shown in Algorithm 3.

**Efficient Backward Pass**  In a vanilla PyTorch implementation, the gradients are computed for each intermediate variables in the reverse order in the computational DAG shown in Fig. 13. Typically, our implementation would also require defining the backward pass by computing the gradients with respect to the intermediate variables, and then propagating them to the input images. However, we derive the backpropagation with respect to $I$ and $J$, given the gradient $g_i = \frac{\partial L}{\partial n_i}$ to avoid calculating intermediate gradients. Using the chain rule, we have:

$$\frac{\partial L}{\partial F_k} = \sum_i \frac{\partial L}{\partial n_i} \frac{\partial n_i}{\partial F_k} \tag{11}$$

$$= \sum_i g_i \left( \frac{2A_i}{B_i C_i} \frac{\partial A_i}{\partial F_k} - \frac{A_i}{B_i^2 C_i} \frac{\partial B_i}{\partial F_k} \right) \tag{12}$$

$$\tag{13}$$

which can be simplified to:

$$\frac{\partial \mu_i^F}{\partial F_k} = \frac{\partial \mu_i^M}{\partial M_k} = w_{ik} \tag{14}$$

$$\frac{\partial A_i}{\partial F_k} = \frac{\partial \left( \sum_k w_{ik} F_k M_k - \mu_i^F \mu_i^M \right)}{\partial F_k} = w_{ik} \left( M_k - \mu_i^M \right) \tag{15}$$

and

$$\frac{\partial B_i}{\partial F_k} = \frac{\partial \left( \sum_k w_{ik} F_k^2 - (\mu_i^F)^2 \right)}{\partial F_k} = 2w_{ik}(F_k - \mu_i^F) \tag{16}$$

Substituting these results to Eq. (12) we have:

$$= \sum_i g_i \left( \frac{2A_i}{B_i C_i} \left( w_{ik} \left( M_k - \mu_i^M \right) \right) - \frac{A_i^2}{B_i^2 C_i} 2w_{ik}(F_k - \mu_i^F) \right) \tag{17}$$

$$= \sum_i \frac{2g_i A_i}{B_i C_i} w_{ik} \left[ M_k - \frac{F_k A_i}{B_i} + \mu_i^F \frac{A_i}{B_i} - \mu_i^M \right] \tag{18}$$

Using the property $w_{ik} = w_{ki}$, and letting $\gamma_i = \frac{2g_i A_i}{B_i C_i}$, we rewrite the previous equation as:

$$= M_k \cdot \left( \sum_i w_{ki} \gamma_i \right) - F_k \cdot \left( \sum_i w_{ki} \frac{\gamma_i A_i}{B_i} \right) + \sum_i w_{ki} \gamma_i \left( \frac{\mu_i^F A_i}{B_i} - \mu_i^M \right) \tag{19}$$

$$= M_k \cdot (w * \gamma)_k - F_k \cdot (w * \gamma_{AB})_k + (w * \gamma_{FM})_k \tag{20}$$

where $\gamma_{AB} = \gamma_i \frac{\mu_i^F A_i}{B_i}$, $\gamma_{FM} = \gamma_i \cdot \left( \frac{\mu_i^F A_i}{B_i} - \mu_i^M \right)$ - and $*$ is the convolution operation. Similarly, the gradient with respect to the moving image $M_k$ is:

$$\frac{\partial L}{\partial M_k} = F_k \left( \sum_i w_{ki} \gamma_i \right) - M_k \left( \sum_i w_{ki} \frac{\gamma_i A_i}{C_i} \right) + \sum_i w_{ki} \gamma_i \left( \frac{\mu_i^M A_i}{C_i} - \mu_i^F \right) \tag{21}$$

$$= F_k \cdot (w * \gamma)_k - M_k \cdot (w * \gamma_{AC})_k + (w * \gamma_{MF})_k \tag{22}$$

where $\gamma_{AC} = \gamma_i \frac{\mu_i^M A_i}{C_i}$, $\gamma_{MF} = \gamma_i \cdot \left( \frac{\mu_i^M A_i}{C_i} - \mu_i^F \right)$. To compute the gradients with respect to $F$ and $M$, we need to compute five tensors of the $\gamma$ family, namely $\gamma, \gamma_{AB}, \gamma_{AC}, \gamma_{FM}$, and $\gamma_{MF}$. This is followed by performing a convolution with all the tensors, and computing elementwise operations given by Eq. (20) and Eq. (22). The $\gamma$ family of tensors are simple elementwise operations on the `state` variable, and therefore can be computed by modifying the `state` variable *inplace* to avoid initializing additional HBM memory.

---

**Algorithm 3** Fused LNCC Implementation

---

**Require:** $F$ (fixed image), $M$ (moving image), $w$ (window size), $\epsilon$ (smoothing term)
1: **function** FORWARD($F$, $M$, $w$, $\epsilon$)
2:     Define convolution filter $W$ of size $w \times w \times w$ with $\sum W[i] = 1$
3:     `state` $\leftarrow$ `fused_create_interm`$(F, M)$   ▷ Single HBM read: $(F, M, F^2, M^2, FM)$
4:     `state` $\leftarrow W * $ `state`                      ▷ Convolution on all channels
5:     LNCC $\leftarrow$ `fusedcc_kernel`(`state`, $\epsilon$) ▷ Computes Eqs. (8) to (10) followed by Eq. (3)
6:     **return** LNCC
7: **end function**
8:
9: **function** BACKWARD($g = \frac{\partial \mathcal{L}}{\partial n}$, `state`, $F$, $M$, $W$, `use_ants_approximation`)
10:     `state` $\leftarrow$ `fused_compute_gamma`$(g,$ `state`$)$ ▷ Computes $\gamma$ family of tensors *inplace*
11:     **if** `use_ants_approximation` **then**
12:         `no-op`                             ▷ ANTs approximation: skip convolutions
13:     **else**
14:         `state` $\leftarrow W * $ `state`                  ▷ Convolution on all intermediates
15:     **end if**
16:     $\frac{\partial L}{\partial F} \leftarrow M \odot \gamma - F \odot \gamma_{AB} + \gamma_{FM}$        ▷ Eq. (20) computed in fused kernel
17:     $\frac{\partial L}{\partial M} \leftarrow F \odot \gamma - M \odot \gamma_{AC} + \gamma_{MF}$        ▷ Eq. (22) computed in fused kernel
18:     **return** $\frac{\partial L}{\partial F}$, $\frac{\partial L}{\partial M}$
19: **end function**

---

Table 3: Speedup and memory usage of different LNCC backends

| N | Method | Forward Time (s) | Forward Speedup | Backward Time (s) | Backward Speedup | Memory (MB) | Memory Reduction (%) |
|---|---|---|---|---|---|---|---|
| | Fast LNCC | 0.001 | 2.95 | 0.003 | 4.86 | 21 | 61.9 |
| | FireANTs | 0.003 | 7.18 | 0.002 | 3.07 | 25 | 68 |
| 64 | VoxelMorph | 0.06 | 158.76 | 0.016 | 24.10 | 17 | 52.9 |
| | `torch.compile` | 0.003 | 6.83 | 0.002 | 2.30 | 24 | 66.7 |
| | Ours | < 0.001 | 1.00 | 0.001 | 1.00 | 8 | 0 |
| | Fast LNCC | 0.008 | 5.88 | 0.026 | 34.09 | 168 | 61.9 |
| | FireANTs | 0.013 | 9.04 | 0.008 | 10.73 | 200 | 68 |
| 128 | VoxelMorph | 0.482 | 341.65 | 0.126 | 168.33 | 136 | 52.9 |
| | `torch.compile` | 0.012 | 8.67 | 0.007 | 8.95 | 192 | 66.7 |
| | Ours | 0.001 | 1.00 | 0.001 | 1.00 | 64 | 0 |
| | Fast LNCC | 0.069 | 6.19 | 0.204 | 82.52 | 1344 | 61.9 |
| | FireANTs | 0.103 | 9.25 | 0.294 | 118.80 | 2176 | 76.5 |
| 256 | VoxelMorph | 3.905 | 351.54 | 3.903 | 1577.37 | 1536.2 | 66.7 |
| | `torch.compile` | 0.1 | 9.02 | 0.284 | 114.74 | 2176 | 76.5 |
| | Ours | 0.011 | 1.00x | 0.002 | 1.00x | 512 | 0 |
| | Fast LNCC | 0.627 | 6.56 | 1.657 | 98.75 | 10752 | 61.9 |
| | FireANTs | 0.856 | 8.95 | 2.396 | 142.77 | 17408 | 76.5 |
| 512 | VoxelMorph | 31.335 | 327.71 | 31.665 | 1887.14 | 12288.2 | 66.7 |
| | `torch.compile` | 0.829 | 8.67 | 2.312 | 137.80 | 17408 | 76.5 |
| | Ours | 0.096 | 1.00 | 0.017 | 1.00 | 4096 | 0 |

**ANTs gradient approximation.** In the ANTs implementation, the gradient computation skips performing the convolution of the $\gamma$ family of tensors. We implement this as an additional flag that the user can toggle as an option for faster backward passes. All our experiments use this approximation.

## G.2 PERFORMANCE

We compare the performance of our fused implementation to various backend implementations. Fig. 7 shows the speedup and memory usage over different image sizes; we tabulate the results here. For this experiment, we initialize two random images of size $N_v \times N_v \times N_v$ and compute the runtime and memory usage for the forward and backward passes. Results are in Table 3. Our implementation consistently achieves upto $6\times$ forward time speedup and $\sim 98\times$ backward time speedup compared to (Jia et al., 2025) and consumes upto 76% less memory than a compiled PyTorch implementation and 61.9% less than a groupwise convolution implementation (Jia et al., 2025).

# H  A HIGHLY EFFICIENT MUTUAL INFORMATION IMPLEMENTATION

Mutual Information (MI) is one of the most commonly used loss functions for *multimodal* image matching (Chen et al., 2022; Avants et al., 2009; Mattes et al., 2001). Beyond multimodal image matching, MI is a cornerstone operation in computer vision (Isola et al., 2014; Zhao et al., 2019c), contrastive learning (Quan et al., 2024), remote sensing (Liang et al., 2013), graph learning (Peng et al., 2023), ecological and social community interactions (Luo et al., 2021; Corso et al., 2020), and cosmological dynamics (Sarkar & Pandey, 2020). In biomedical imaging and life sciences, MI is used for multimodal image alignment using the assumption that pixels in multimodal images codify some nonlinear function of the underlying tissue type.

**Vanilla MI implementation**   Given images $I$ and $J$, Mattes MI considers the intensities from the images as samples from probability distributions $p_I$ and $p_J$ that encode some imaging physics. The intensity pairs $(I_k, J_k)$ are considered to be samples from the joint distribution $p_{IJ}$. If the images are aligned, then $I_k$ and $J_k$ are highly 'predictable' from each other, implying a low conditional entropy $H(I|J)$, or equivalently a large distance from the distribution $p_I p_J$ which models the joint distribution if samples from $I$ and $J$ were independent. This is precisely the mutual information criteria. Since the samples $I_k, J_k$ follow some unknown distributions, we use a kernel density estimator using kernel $\kappa$ to estimate the empirical distributions of the joint and marginal distributions. To compute empirical MI, the continuous kernel density estimates are discretized into a probability mass function (PMF) with a finite number of bins. The number of bins $B$ is a hyperparameter that is used to define bin centers $b_i \in [0, 1]$ for $i = \{1, \ldots, B\}$, assuming that the intensities are scaled to the range $[0, 1]$.

To compute the discrete PMF with autodifferentiation, we compute a Parzen Block $\Psi_I \in \mathbb{R}^{B \times N}$, s.t. $\Psi_I(i, k) = \kappa(b_i - I_k)$. This forms the memory bottleneck in computing the Mattes MI similarity criteria. In the following, we provide a fused implementation that avoids the $O(NB)$ cost of the Parzen Block, making our implementation only $O(1)$ additional HBM overhead.

## H.1  IMPLICIT MI IMPLEMENTATION

We implement custom forward and backward passes to compute the joint and marginal histograms $p_{IJ}, p_I, p_J$ from $I$ and $J$ directly, avoiding the $O(NB)$ cost of the Parzen Block. We derive the backward pass first, followed by the forward pass followed by an efficient approximate estimator of the histograms leading to a faster forward pass.

### H.1.1  BACKWARD PASS

We are interested in computing the gradients $\frac{\partial L}{\partial I}, \frac{\partial L}{\partial J}$ given $\frac{\partial L}{\partial p_{IJ}}, \frac{\partial L}{\partial p_I}, \frac{\partial L}{\partial p_J}$. We denote $\omega(b_i - I_k) = \frac{\partial \kappa(b_i - I_k)}{\partial I_k}$.

$$\frac{\partial L}{\partial I_k} = \sum_{m,n} \frac{\partial L}{\partial p_{IJ}[m,n]} \frac{\partial p_{IJ}[m,n]}{\partial I_k} + \sum_{n} \frac{\partial L}{\partial p_I[n]} \frac{\partial p_I[n]}{\partial I_k} \tag{23}$$

$$= \sum_{m,n} g_{IJ}[m,n] \left( \omega(b_m - I_k) \kappa(b_n - J_k) \right) + \sum_{n} g_I[n] \left( \omega(b_n - I_k) \right) \tag{24}$$

$$= \sum_{n} \left[ g_I[n] \omega(b_n - I_k) + \sum_{m} g_{IJ}[m,n] \omega(b_m - I_k) \right] = \sum_{n} \zeta_1[n] + \zeta_2[n] \tag{25}$$

where $\zeta_1[n] = g_I[n] \omega(b_n - I_k)$ and $\zeta_2[n] = \sum_m g_{IJ}[m,n] \omega(b_m - I_k)$.

To compute this backward pass efficiently, we launch $\lceil N/B \rceil$ threadblocks and partition each threadblock in groups of $B$ threads, and compute the partial gradients $\zeta_1[n], \zeta_2[n]$ on each thread. Each group loads the values of $I_k, J_k$ into register memory.  we first compute the quantities $\kappa(b_n - I_k), \kappa(b_n - J_k), \omega(b_n - I_k), \omega(b_n - J_k)$ on thread $n$ and use four shared memory arrays to store them.  On thread $n$, we compute the partial gradient $\zeta_1[n] = g_I[n] \omega(b_n - I_k)$ and $\zeta_2[n] = \sum_m g_{IJ}[m,n] \omega(b_m - I_k)$ using a for-loop over the index $m \in \{1, \ldots, B\}$. Finally, on each thread we store the value $\zeta_1[n] + \zeta_2[n]$ on shared memory indexed at $n$, followed by a $O(\log(n))$

parallel sum over partitioned threads to compute the gradient $\frac{\partial L}{\partial I_k} = \sum_n \zeta_1[n] + \zeta_2[n]$. A similar argument is used to compute the gradient over $\frac{\partial L}{\partial J_k}$. This leads to a faster backward pass than the vanilla PyTorch implementation using no additional HBM overhead Fig. 7(b).

**Generalization to novel kernels**    Note that unlike the vanilla implementation, where some choices of $\kappa$ are more memory intensive than others (for example, the BSpline kernel has $k_P = 14$ versus $k_P = 4$ for the Gaussian kernel), the memory overhead of our implementation does not depend on the analytical form of $\kappa$. To generalize the Implicit MI implementation to novel kernels, the user can specify the form of $\kappa$ and its derivative $\omega$ in the forward and backward passes without any additional considerations.

### H.1.2    Forward Pass

The forward pass is computed similarly. Note that the individual contributions from $I_k, J_k$ to the joint histogram $p_{IJ}[m, n]$ are $p_{IJ}[m, n] = \kappa(b_m - I_k)\kappa(b_n - J_k)$ for all $m, n \in \{1, \ldots, B\}$. The marginal histograms $p_I[n], p_J[n]$ are computed as $p_I[n] = \kappa(b_n - I_k)$ and $p_J[n] = \kappa(b_n - J_k)$ for all $n \in \{1, \ldots, B\}$. Similar to the backward pass, we launch $\lceil N/B \rceil$ threadblocks and partition the threadblock in groups of $B$ threads. Each group of $B$ threads loads the values of $I_k, J_k$ into register memory. On thread $n$, we compute the quantities $\kappa(b_n - I_k), \kappa(b_n - J_k)$ and store them in shared memory. Thread $n$ can add these quantities into the HBM for histogram entries $p_I[n], p_J[n]$ directly. For computing the joint histogram $p_{IJ}[m, n]$, thread $n$ loops over $m \in \{1, \ldots, B\}$ and adds the quantities $\kappa(b_m - I_k)\kappa(b_n - J_k)$ into the HBM for histogram entries $p_{IJ}[m, n]$. Since all values of $\kappa(b_m - I_k), \kappa(b_n - J_k)$ are stored in shared memory, this operation is not bottlenecked by slow HBM reads. To avoid HBM write contentions, we write these values into intermediate histogram buffers of sizes $C \times B \times B, C \times B$ (where $C$ is a constant of choice), and sum along the $C$ dimension. However, this is still a relatively slow operation due to computation of $\kappa(b_m - I_k), \kappa(b_n - J_k)$ and making $NB^2$ HBM writes. We propose an efficient approximate forward pass that launches only $N$ instead of $NB$ threads, and makes only $3N$ HBM writes.

**An approximate histogram estimator**    Given a kernel $\kappa$, we can write $\kappa(b_m - I_k) = \int_t \delta(b_m - I_k - t)\kappa(t)dt = \delta(b_m - I_k) * \kappa$, where $\delta$ is the Dirac delta function with the property $\int_{x=-\infty}^{\infty} \delta(x)f(x)dx = f(0)$ for any function $f$. Using the principle of superposition, we can write $p_I[m] = \frac{1}{N}\sum_k \kappa(b_m - I_k) = \frac{1}{N}\sum_k \kappa * \delta(b_m - I_k) = \kappa * \left(\frac{1}{N}\sum_k \delta(b_m - I_k)\right)$.

In the continuous case, $p_I$ can be obtained *exactly* by calculating the Dirac delta distribution $p_I^\delta(b) = \frac{1}{N}\sum_k \delta(b - I_k)$ and convolving it with the kernel $\kappa$. However, in the discrete case, this value is inexact. To see this, consider a value $I_k$ that is in bin $m$, i.e. $\|I_k - b_m\| < \frac{1}{2B}$. The exact value of the PMF due to this sample is $\kappa(b_m - I_k)$. However, the approximate value of the PMF is $\kappa(0)$ since $\delta(b_m - I_k) = 1$ for all $I_k : \|I_k - b_m\| < \frac{1}{2B}$ due to binning, and convolving with $\kappa$ returns $\kappa(0)$. Since $\|I_k - b_m\| < \frac{1}{2B}$, we can assume that $\|\kappa(0) - \kappa(b_m - I_k)\|$ is small.

To implement this histogram computation efficiently, we launch $N$ threads and in each thread $k$, compute the bin indices $m^* = \lfloor I_k B \rfloor, n^* = \lfloor J_k B \rfloor$ for each thread, avoiding computation of *soft entries* $\kappa(b_m - I_k), \kappa(b_n - J_k)$ altogether. We simply add 1 to the histogram entries $p_{IJ}[m^*, n^*], p_I[m^*], p_J[n^*]$ in the aggregated histogram buffers, avoiding writing into HBM entries for all $(m, n) \in \{1, \ldots, B\}^2$. This reduces the number of HBM writes from $NB^2 + 2NB$ to $3N$. For $B = 32$, this represents $362\times$ less HBM writes. After performing the average, we convolve this histogram with the kernel $\kappa$ to get the approximate PMF. Since the convolution is done on a $B$ and $B \times B$ sized histograms, this operation is cheap. This implementation leads to faster runtime, consistent performance for both TransMorph and FireANTs (see Table 1).

## I    Composite Implicit Grid Sampler

## J    Ring Sampler for scalable distributed interpolation

The random-access nature of deformable interpolation making scaling a difficult challenge for arbitrarily large problem sizes. Given a configuration of sharded images and warp fields across $H$

---

**Algorithm 4** Grid Sampler Implementation

---

**Require:** $J_h$ (moving image shard), $[\mathbf{u}]_j$ (warp field shard), $A_h$ (rescaled affine), $t_h$ (rescaled translation), $S_h$ (diag. scale)

 1: **function** FORWARD($J_h$, $A_h$, $t_h$, $S_h$, $[\mathbf{u}]_j$)
 2:     `out` $\leftarrow$ `zeros_like`($[\mathbf{u}]_j[0]$)
 3:     **for all** target voxels $(z, y, x)$ **in parallel (one thread per voxel) do**
 4:         $X \leftarrow (x, y, z)$
 5:         $X_{\text{aff}} \leftarrow A_h X + t_h$                    ▷ *affine transform only*
 6:         $X_{\text{disp}} \leftarrow S_h [\mathbf{u}]_j[:, z, y, x]$           ▷ *add scaled displacement*
 7:         $X_{\text{src}} \leftarrow X_{\text{aff}} + X_{\text{disp}}$
 8:         `out`$[z, y, x] \leftarrow$ `trilinear_interpolate`($J_h$, $X_{\text{src}}$)     *zero padding at bounds*
 9:     **end for**
10:     **return** `out`
11: **end function**
12:
13: **function** BACKWARD($g = \frac{\partial \mathcal{L}}{\partial \text{out}}$, $J_h$, $A_h$, $t_h$, $S_h$, $[\mathbf{u}]_j$)
14:     Initialize $g_{J_h} = 0$, $g_{[\mathbf{u}]_j} = 0$, $g_{A_h} = 0$, $g_{t_h} = 0$
15:     **for all** target voxels $(z, y, x)$ **in parallel (one thread per voxel) do**
16:         Recompute $X$, $X_{\text{aff}}$, $X_{\text{disp}}$, $X_{\text{src}}$
17:         Compute tri-linear weights $w_{b_x b_y b_z}$ and $\frac{\partial v}{\partial X_{\text{src}}}$
18:         Accumulate $g_{J_h}$ into 8 neighbors using $w_{***} \cdot g[z, y, x]$ *(bounds-checked, zero-padded)*
19:         $g_{[\mathbf{u}]_j}[:, z, y, x] \mathrel{+}= S_h \frac{\partial v}{\partial X_{\text{src}}} g[z, y, x]$
20:         $g_{A_h} \mathrel{+}= \left( \frac{\partial v}{\partial X_{\text{src}}} g[z, y, x] \right) X^\top$
21:         $g_{t_h} \mathrel{+}= \frac{\partial v}{\partial X_{\text{src}}} g[z, y, x]$
22:     **end for**
23:     **return** $g_{J_h}$, $g_{[\mathbf{u}]_j}$, $g_{A_h}$, $g_{t_h}$
24: **end function**

---

hosts, neighboring voxels in the sharded warp field can point to pixels in arbitrary regions in the image, illustrated in Fig. 4(a). Moreover, the control points for interpolation can be irregularly distributed across different hosts, illustrated in Fig. 4(b). This makes computation of the interpolated image challenging for displacements that point to pixels between boundaries of different hosts. One approach to avoid this problem is to store the entire moving image on each GPU to compute the interpolated image. However, this approach is impractical once the image size exceeds the memory per GPU. To achieve weak scaling, the HBM overhead per GPU must be proportional to $N/H$. To alleviate this problem, we propose a ring sampler that avoids the need to store the entire moving image on each GPU by decomposing linear interpolation into partial sums. This produces mathematically correct interpolated images regardless of the nature of the warp field, without storing the entire moving image on each GPU.

## J.1 DERIVATION

Consider a $d$-linear interpolation of an image $I$ defined on $\Omega$ using warp coordinates $[\mathbf{u}]_\Omega$ defined on $\Omega$.

$$I = \sum_{b \in \{0,1\}^n} \left( \prod_{k=1}^n (1 - \alpha_k)^{1 - b_k} \alpha_k^{b_k} \right) I\big[i_1 + b_1,\ i_2 + b_2,\ \ldots,\ i_n + b_n\big] \tag{26}$$

where $i_k = \lfloor \varphi(x)_k \rfloor$, $\quad \alpha_k = \varphi(x)_k - i_k$, $\quad$ for $k = 1, \ldots, d$. Let the individual pixels $I\big[i_1 + b_1,\ i_2 + b_2,\ \ldots,\ i_n + b_n\big]$ be partitioned across $H$ hosts. Since each pixel belongs to exactly one host, we can write $\sum_{h=1}^H \mathbb{I}(\mathbf{i} + \mathbf{b} \in [x]_h) = 1$ and multiply with $I[\mathbf{i} + \mathbf{b}]$ to get:

$$I = \sum_{b \in \{0,1\}^n} \left( \prod_{k=1}^n (1 - \alpha_k)^{1-b_k} \alpha_k^{b_k} \right) \left( I[\mathbf{i} + \mathbf{b}] * \left( \sum_{h=1}^H \mathbb{I}(\mathbf{i} + \mathbf{b} \in [x]_h) \right) \right) \tag{27}$$

$$= \sum_{h=1}^H \sum_{b \in \{0,1\}^n} \left( \prod_{k=1}^n (1 - \alpha_k)^{1-b_k} \alpha_k^{b_k} \right) \left( I[\mathbf{i} + \mathbf{b}] * \mathbb{I}(\mathbf{i} + \mathbf{b} \in [x]_h) \right) \tag{28}$$

$$= \sum_{h=1}^H I_h \tag{29}$$

where

$$I_h = \sum_{b \in \{0,1\}^n} \left( \prod_{k=1}^n (1 - \alpha_k)^{1-b_k} \alpha_k^{b_k} \right) I[\mathbf{i} + \mathbf{b}] * \mathbb{I}(\mathbf{i} + \mathbf{b} \in [x]_h) \tag{30}$$

$$= \sum_{b \in \{0,1\}^n} \left( \prod_{k=1}^n (1 - \alpha_k)^{1-b_k} \alpha_k^{b_k} \right) J_h[\mathbf{i} + \mathbf{b}] \tag{31}$$

where $J_h[\mathbf{x}] = I[\mathbf{x}]$ if $\mathbf{x} \in [\mathbf{x}]_h$ else 0. Image $J_h$ is therefore *identical* to the sharded image $I$ on host $h$. Eq. (31) refers to performing trilinear interpolation on the shard $I_h$ (with zero padding) since the sum is only over coordinates that reside in $[x]_h$. This means the warped image in Eq. (26) can be obtained by performing interpolation over the shards individually and adding the warped images together. This is illustrated in Fig. 4(c). Coordinates residing between multiple shards will accumulate partial sums from each sharded image, and no additional consideration is needed for boundary conditions. The communication protocol in this algorithm is similar to Ring Attention (Liu et al., 2024b), where image shards are passed across hosts, and partial results are accumulated into the final result. Our algorithm requires a memory overhead of only $N/H$ to store the sharded image from host $j \neq i$. Our pseudocode is provided in Algorithm 5.

## J.2 Implementation Considerations

**Rescaling the warp function to sample sharded images**   Interpolating from sharded images requires one additional consideration. The grid sampler interpolates an image $I$ defined on $\Omega$ using warp coordinates $[\mathbf{u}]_\Omega$ defined on $\Omega$. However, the sharded image $J_h$ is defined on the domain $\Omega_h$, and therefore any warped coordinate $\varphi(x) \in \Omega$ must be rescaled to the corresponding coordinates in $\varphi_h(x) \in \Omega_h$. From the implementation standpoint, the leftmost coordinate of $J_h$ is $x_{\min}^h$ when the entire image $I$ is passed to `grid_sampler`. However, when $J_h$ is provided as input to `grid_sampler`, the leftmost pixel of $J_h$ is located at $[-1, -1, \ldots, -1]$ according to PyTorch convention. Since our optimization variables $A, t, [\mathbf{u}]$ refer to locations on $\Omega$, and not $\Omega_h$, we need to rescale these variables appropriately when sampling from $J_h$.

The rescaling corresponds to a diagonal scaling matrix $S_h$ and translation $t_h$ such that $S_h x_{\min}^h + t_h = x_{\min}^\Omega$ and $S_h x_{\max}^h + t_h = x_{\max}^\Omega$. The resampled warp function to sample from $J_h$ becomes $\varphi_h(x) = S_h(Ax + t + u(x)) + t_h = (A_h' x + t_h') + S_h u(x)$. where $A_h', t_h' = S_h A, (S_h t + t_h)$. Therefore, we must sample $J_h$ using the transform $A_h'[\mathbf{x}]_{\Omega_h} + t_h' + S_h[\mathbf{u}]_{\Omega_h}$. In the vanilla grid sampler implementation, the intermediate grid $S_h[\mathbf{u}]_{\Omega_h}$ and its gradient consume another $6N/H$ memory. Combined with the $N/H$ overhead for storing the received image shard, we add a total of $7N/H$ memory overhead, which is less than $N$ for $H \geq 8$, making the algorithm impractical for fewer GPUs (say $H = 4$).

To prevent this $6N/H$ additional overhead, we extend the generalized grid sampler as mentioned Appendix 3.1 to sample from a transform of the form $A[x] + t + S[u]$ directly. This computes the value $Su(x)$ directly inside the CUDA kernel, and the backward pass also computes and accumulates the gradient w.r.t. $u(x)$ directly, avoiding the $6N/H$ overhead.

**Interleaved communication**   An important implementation detail is the interleaving of communication and computation in the ring sampler. While we compute the partial moved image aggregate, the next image shard can be fetched asynchronously in the background. This is illustrated in Fig. 14.

---

**Algorithm 5** Ring Sampler Implementation

---

**Require:** $M_j$ (moving image shard), $[\mathbf{u}]_j$ (warp field shard), $(A, t)$ (affine transform)
1: **function** FORWARD($M_j, [\mathbf{u}]_j, (A, t)$)
2:     Define $\texttt{moved}_j = 0$
3:     **for** $h = 1$ **to** $H$ **do**
4:         $J_h \leftarrow \texttt{send\_and\_recv}(M_j, h)$       ▷ Send and receive the image shard from offset $h$
5:         Compute diagonal $S_h, t_h$ such that $S_h x^h_{\min} + t_h = x^\Omega_{\min}$ and $S_h x^h_{\max} + t_h = x^\Omega_{\max}$
6:         Rescale affine transform $A_h \leftarrow S_h A, t_h \leftarrow S_h t + t_h$
7:         $\texttt{moved}_j \leftarrow \texttt{moved}_j + \texttt{grid\_sampler}(J_h; A_h, t_h, S_h, [\mathbf{u}]_j)$     ▷ Avoid computing $S_h [\mathbf{u}]_j$ explicitly
8:     **end for**
9:     **return** $\texttt{moved}_j$
10: **end function**
11:
12: **function** BACKWARD($g = \frac{\partial \mathcal{L}}{\partial \texttt{moved}_j}, \texttt{moved}_j, M_j, [\mathbf{u}]_j, (A, t)$)
13:     Define $g_{[\mathbf{u}]_j} = 0, g_A = 0, g_t = 0, g_{M_j} = 0$
14:     **for** $h = 1$ **to** $H$ **do**
15:         $J_h \leftarrow \texttt{send\_and\_recv}(M_j, h)$       ▷ Send and receive the image shard from offset $h$
16:         Compute diagonal $S_h, t_h$ such that $S_h x^h_{\min} + t_h = x^\Omega_{\min}$ and $S_h x^h_{\max} + t_h = x^\Omega_{\max}$
17:         Rescale affine transform $A_h \leftarrow S_h A, t_h \leftarrow S_h t + t_h$
18:         **if** $\texttt{requires\_grad}(M_j)$ **then**
19:             $g_{inp} \leftarrow \texttt{zeros\_like}(M_j)$
20:         **else**
21:             $g_{inp} \leftarrow \texttt{None}$
22:         **end if**
23:         Compute $\texttt{backward\_grid\_sampler}(g, J_h, A_h, t_h, S_h, [\mathbf{u}]_j, g_{[\mathbf{u}]_j}, g_A, g_t, g_{inp})$
24:         **if** $\texttt{requires\_grad}(M_j)$ **then**
25:             $g'_{M_j} = \texttt{send\_and\_recv}(g_{inp}, -h)$
26:             $g_{M_j} \leftarrow g_{M_j} + g'_{M_j}$
27:         **end if**
28:     **end for**
29:     **return** $g_{[\mathbf{u}]_j}, g_A, g_t, g_{M_j}$
30: **end function**

---

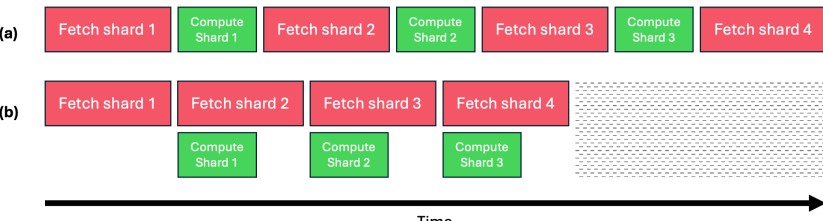

**Figure 14:** Interleaved communication (red) and computation (green) in the ring sampler. gray denotes time saved by interleaving communication and computation.

## J.3 ALTERNATIVE DESIGNS

A naive approach can be to route the coordinate $\varphi(x_i) \in [\mathbf{x}]_j$ to GPU $j$ and retrieve the image coordinate, similar to routing tokens using expert parallelism (EP) used for Mixture-of-Experts (MoEs) (Shazeer et al., 2017; Jordan & Jacobs, 1994). However, this approach has two major drawbacks in our setting. First, due to the deformable nature of $\varphi$, the partitioning of coordinates across hosts is generally uneven. In the worst case, a single GPU can receive all $3N$ coordinates leading to an indirect `allgather` operation resulting in OOMs or uneven GPU utilization across hosts. Second, coordinates that point to regions between two multiple image boundaries need to be sent to variable number of hosts, which is non-trivial to implement. These two factors make both the forward and

backward pass implementations cumbersome. Inspired by (Liu et al., 2024b), we propose a distributed ring sampler that decomposes the computation into partial sums, leading to a simple implementation without degraded scaling performance Fig. 8.

# K    CORRECTNESS OF IMPLEMENTATION

All code is checked for numerical correctness by comparing the results with PyTorch implementations using unit and integration tests. Code and generated data will be made available to the community.

## K.1    ABLATION ON FIREANTS SPEEDUP

We run FireANTs with different backends for LNCC and MI loss functions on the OASIS validation set (Marcus et al., 2007; Hering et al., 2022). We measure the end-to-end runtime, peak memory usage (except the fixed and moving images), and Dice score. We ablate on both Greedy and SyN algorithms; in the case of SyN, additional gradients may be required. Results in Table 4 show that our implementation achieves a significant speedup over the baseline implementations. Although the `torch.compile` version of LNCC is faster than other variants, it leads to brittle performance.

Table 4: **Extended Results on accelerated registration on FireANTs**: Accelerating FireANTs registration with various computation backends and registration algorithms (Greedy and SyN). Our implementations maintain accuracy while substantially reducing runtime and peak memory usage. (Green)/ (Yellow) = best/second; Speedup and memory reduction are computed with respect to our kernels. Our fused kernels maintain accuracy while substantially reducing runtime and peak memory usage.

| Algorithm | Method | Backend | Dice Score ↑ | Runtime (s) ↓ | Memory (MB) ↓ | Speedup ↑ | Mem. Reduction (%) ↑ |
|---|---|---|---|---|---|---|---|
| Greedy | LNCC | VXM/TM | $76.96 \pm 3.60$ | $57.08 \pm 2.45$ | $1418.5 \pm 0.0$ | 113.47 | 59.29 |
| | LNCC | FastLNCC | $76.96 \pm 3.60$ | $3.76 \pm 0.16$ | $1026.3 \pm 0.0$ | 7.48 | 43.73 |
| | LNCC | FireANTs | $72.81 \pm 3.87$ | $1.44 \pm 0.08$ | $1044.5 \pm 0.0$ | 2.87 | 44.71 |
| | LNCC | `torch.compile` | $69.35 \pm 4.09$ | $0.82 \pm 0.04$ | $860.7 \pm 0.0$ | 1.63 | 32.90 |
| | LNCC | Ours | $78.67 \pm 3.04$ | $0.50 \pm 0.01$ | $577.5 \pm 0.0$ | 1.00 | 0.00 |
| Greedy | MI | PyTorch | $75.88 \pm 3.45$ | $7.51 \pm 0.37$ | $12206.3 \pm 0.0$ | 2.59 | 95.27 |
| | MI | `torch.compile` | $75.88 \pm 3.45$ | $1.05 \pm 0.05$ | $3865.5 \pm 0.0$ | 0.36 | 85.06 |
| | MI | Ours | $75.87 \pm 3.44$ | $2.90 \pm 0.16$ | $577.5 \pm 0.0$ | 1.00 | 0.00 |
| | MI | Ours + `torch.compile` | $75.93 \pm 3.47$ | $2.95 \pm 0.16$ | $657.3 \pm 0.0$ | 1.02 | 12.13 |
| SyN | LNCC | VXM/TM | $76.69 \pm 2.88$ | $63.57 \pm 0.58$ | $1892.0 \pm 0.0$ | 65.92 | 50.05 |
| | LNCC | FastLNCC | $76.70 \pm 2.88$ | $4.27 \pm 0.05$ | $1486.7 \pm 0.0$ | 4.43 | 36.43 |
| | LNCC | FireANTs | $74.70 \pm 2.93$ | $2.55 \pm 0.10$ | $1616.4 \pm 0.0$ | 2.65 | 41.54 |
| | LNCC | `torch.compile` | $71.65 \pm 3.41$ | $1.46 \pm 0.04$ | $1472.0 \pm 0.0$ | 1.51 | 35.80 |
| | LNCC | Ours | $78.79 \pm 2.82$ | $0.96 \pm 0.08$ | $945.0 \pm 0.0$ | 1.00 | 0.00 |
| SyN | MI | PyTorch | $76.74 \pm 2.58$ | $12.84 \pm 0.66$ | $17720.8 \pm 0.0$ | 2.96 | 94.67 |
| | MI | `torch.compile` | $76.76 \pm 2.58$ | $2.40 \pm 0.13$ | $7758.9 \pm 0.0$ | 0.55 | 87.82 |
| | MI | Ours | $76.86 \pm 2.59$ | $4.34 \pm 0.28$ | $945.0 \pm 0.0$ | 1.00 | 0.00 |
| | MI | Ours + `torch.compile` | $77.00 \pm 2.57$ | $4.56 \pm 0.24$ | $1104.5 \pm 0.0$ | 1.05 | 14.44 |

**Table 5: Extended Efficiency Results on faux-OASIS-dataset**: Comparison of registration methods across multiple resolutions. Reported metrics include average Dice similarity coefficient (higher is better), wall-clock runtime, GPU cost (measured in GB-hours), relative speedup, and GPU cost reduction with respect to FireANTs + FFDP(Ours). GPU usage (e.g., single GPU, multi-GPU, or CPU) is annotated alongside the cost values.

| Resolution | Method | Avg Dice Score ↑ | Wall Clock ↓ ($10^{-2}$ Hours) | GPU Cost ↓ ($10^{-2}$ GB-Hours) | Speedup | GPU Cost Reduction (%) |
|---|---|---|---|---|---|---|
| 1 mm | TransMorph | $0.851 \pm 0.016$ | 0.015 | $0.262^{1}$ | $0.56\times$ | 87.81 |
| | VFA | $0.851 \pm 0.023$ | 0.017 | $0.216^{1}$ | $0.63\times$ | 85.18 |
| | Ours | $0.838 \pm 0.028$ | 0.027 | $0.032^{1}$ | $1.00\times$ | 0.00 |
| | UniGradICON-IO | $0.826 \pm 0.022$ | 5.167 | $58.498^{1}$ | $194.07\times$ | 99.95 |
| | FireANTs | $0.822 \pm 0.032$ | 0.060 | $0.141^{1}$ | $2.25\times$ | 48.83 |
| | UniGradICON-noIO | $0.815 \pm 0.026$ | 0.067 | $0.238^{1}$ | $2.50\times$ | 86.55 |
| | SynthMorph | $0.801 \pm 0.022$ | 2.155 | $99.061^{1}$ | $80.93\times$ | 99.97 |
| | Anatomix | $0.796 \pm 0.035$ | 0.379 | $2.656^{1}$ | $14.24\times$ | 98.80 |
| | CLAIRE | $0.776 \pm 0.044$ | 0.518 | $1.389^{1}$ | $19.47\times$ | 97.70 |
| | ITK-dreg | $0.662 \pm 0.055$ | 1.527 | $1.017^{CPU}$ | $57.37\times$ | – |
| 500 µm | Ours | $0.872 \pm 0.028$ | 0.109 | $0.862^{1}$ | $1.00\times$ | 0.00 |
| | FireANTs | $0.841 \pm 0.033$ | 0.270 | $4.136^{1}$ | $2.48\times$ | 48.22 |
| | VFA | $0.805 \pm 0.044$ | 0.302 | $3.896^{1}$ | $2.78\times$ | 77.87 |
| | CLAIRE | $0.779 \pm 0.051$ | 25.903 | $396.169^{1}$ | $238.04\times$ | 99.78 |
| | SynthMorph | $0.771 \pm 0.035$ | 4.068 | $187.049^{1}$ | $37.39\times$ | 99.54 |
| | TransMorph | $0.759 \pm 0.028$ | 0.198 | $3.501^{1}$ | $1.82\times$ | 75.38 |
| | Anatomix | $0.758 \pm 0.040$ | 8.837 | $310.818^{1}$ | $81.21\times$ | 99.72 |
| | ITK-dreg | $0.699 \pm 0.056$ | 41.259 | $207.466^{CPU}$ | $379.17\times$ | – |
| | UniGradICON-IO | $0.615 \pm 0.047$ | 84.538 | $1072.657^{1}$ | $776.89\times$ | 99.92 |
| | UniGradICON | $0.610 \pm 0.044$ | 0.842 | $3.545^{1}$ | $7.73\times$ | 75.69 |
| 250 µm | Ours | $0.895 \pm 0.029$ | 1.065 | $47.059^{1}$ | $1.00\times$ | 0.00 |
| | CLAIRE | $0.809 \pm 0.054$ | 1207.536 | $159\,046.981^{4}$ | $1133.84\times$ | 99.97 |
| | FireANTs | $0.777 \pm 0.064$ | 13.588 | $253.295^{1}$ | $11.73\times$ | 81.42 |
| | VFA | $0.714 \pm 0.066$ | 3.872 | $49.939^{1}$ | $3.64\times$ | 5.77 |
| | SynthMorph | $0.690 \pm 0.052$ | 32.808 | $1507.133^{1}$ | $30.80\times$ | 96.88 |
| | TransMorph | $0.689 \pm 0.044$ | 2.597 | $45.965^{1}$ | $2.44\times$ | −2.38 |
| | Anatomix | $0.620 \pm 0.031$ | 88.480 | $3112.015^{1}$ | $83.07\times$ | 98.49 |
| | UniGradICON-IO | $0.398 \pm 0.062$ | 163.812 | $2539.721^{1}$ | $153.80\times$ | 98.15 |
| | UniGradICON | $0.359 \pm 0.044$ | 7.811 | $55.057^{1}$ | $7.33\times$ | 14.53 |
| | ITK-dreg | $0.758 \pm 0.046$ | 1363.868 | $33\,065.677^{CPU}$ | $1280.63\times$ | – |

## L  ADDITIONAL DETAILS ON THE SIMULATED EX-VIVO BRAIN MRI DATASET

In this section, we provide additional details on the synthetic data generation pipeline for the faux-OASIS dataset, followed by baseline configurations, and finally compare performance-efficiency tradeoffs and show qualitative results.

### L.1  SYNTHETIC DATA GENERATION PIPELINE

To emulate high resolution ($250\mu m$ isotropic) T1 weighted images, we use the standard OASIS validation dataset to generate synthetic images. Our method is inspired by Billot et al. (2023); Dey et al. (2025) to use the labelmaps as a starting point and synthesize images that are faithful to the labelmaps. The synthetic data generation pipeline is illustrated in Fig. 15. Specifically, our pipeline has three stages:

1. **Compute per-label intensity statistics**: For each label, we consider all the intensities in the voxels belonging to the label. We store mean and standard deviation of the intensities for each label computed over the entire OASIS validation set.

2. **Geometry-preserving upsampling of labels**: We use the labelmaps at 1mm isotropic and perform surface-based upsampling to resample the labelmaps with subvoxel accuracy (Sullivan & Kaszynski, 2019).

3. **Intensity painting**: We use the per-label intensity statistics and the voxelized labelmaps at $250\mu m$ isotropic to synthesize the images.

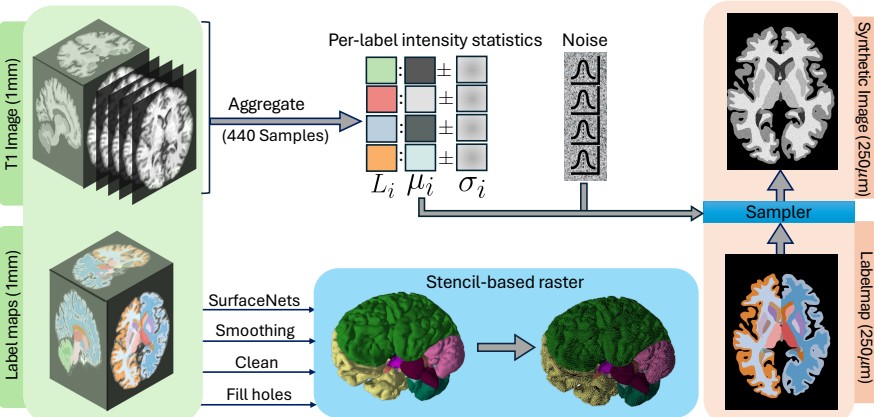

**Figure 15: Synthetic data generation pipeline for faux-OASIS**. Coarse anatomical labels undergo geometry-preserving upsampling via surface reconstruction, followed by statistical intensity painting to produce high-resolution MR images at $0.25\,\mathrm{mm}$.

Following the generation of $250\mu m$ images, we downsample the images to $500\mu m$ and $1mm$ isotropic to show the effect on performance with downsampled images.

We describe the pipeline in detail below.

**Per-label intensity statistics.** Contrary to other synthetic data generation pipelines Dey et al. (2025); Billot et al. (2023), we do not want to generate randomized intensities for each image and want to simulate the T1-weighted images. Towards this end, we compute the per-label intensity statistics for all images in the OASIS validation set.

**Geometry-preserving upsampling of labelmaps.** Given an image volume and labelmap pair $(I, L)$, we upsample the labelmap to $250\mu m$ isotropic $L_\uparrow$. However, naively upsampling the label voxel grid and thresholding typically causes blocky artifacts (Frisken, 2022; Lorensen & Cline, 1998; Schroeder & Tsalikis, 2023), which has led to many sophisticated subvoxel-accurate surface reconstruction algorithms. We use PyVista's SurfaceNets algorithm (Frisken, 2022) to extract surface contours from 3D image label maps. Specifically, an `ImageData` object with labels is converted into *cell data* using `contour_labels` (VTK SurfaceNets) to obtain per-label surfaces $S_\ell$ that respect voxel geometry and avoid block artifacts with voxel based interpolation. The generated surface is smoothed using a constrained Taubin/Windowed-Sinc smoothing with conservative iterations (typically 16-30, relaxation $\approx 0.5$), then use `clean` and `fill_holes` to remove slivers and pinholes while preserving anatomical shape fidelity. The surface $S_\ell$ is voxelized to obtain a binary mask $M_\ell$, then the labelmap $L_\uparrow$ is assembled as

$$L_\uparrow(\mathbf{p}) = \begin{cases} \ell & \text{if } M_\ell(\mathbf{p}) = 1 \text{ for some } \ell \geq 1, \\ 0 & \text{otherwise.} \end{cases}$$

Finally, image-stencil-based rasterization (`voxelize_binary_mask`) is performed into the target `ImageData` at $t = 0.25\,\mathrm{mm}$. When surfaces overlap, later labels in the loop take precedence; we process labels in anatomical priority order to ensure critical structures are preserved. All steps for labelmap upsampling are implemented with PyVista/VTK for robustness and reproducibility.

**Synthesizing the image.** For each label, we fill the voxels with intensities sampled from a normal distribution with the mean and standard deviation of the intensities corresponding to the label.

Given $\{(\mu_\ell, \sigma_\ell)\}$, we synthesize the image by i.i.d. draws within each region:

$$I_{\text{syn}}(\mathbf{p}) \sim \mathcal{N}\big(\mu_{L_\uparrow(\mathbf{p})}, \sigma^2_{L_\uparrow(\mathbf{p})}\big), \qquad \mathbf{p} \in \Omega_t.$$

We follow this step with a Gaussian smoothing with $\sigma = 0.75$ voxels to impart local coherence without washing out label edges. Background ($L_\uparrow{=}0$) is set to zero.

---

**Algorithm 6** High-Resolution MR Synthesis Pipeline

---

**Require:** $L$, spacings $\mathbf{s}$, affine $A$, stats $\{(\mu_\ell, \sigma_\ell)\}_{\ell=1}^K$, target spacing $t$
1: Compute $\tilde{N}_x, \tilde{N}_y, \tilde{N}_z$ and $\tilde{A}$ as above
2: $L_\uparrow \leftarrow 0$ on $\Omega_t$
3: **for** $\ell = 1$ **to** $K$ **do**
4:     $S_\ell \leftarrow \text{SurfaceNets}(L{=}\ell)$; smooth & fill holes
5:     $M_\ell \leftarrow \text{Voxelize}(S_\ell, \Omega_t)$
6:     $L_\uparrow[\text{where } M_\ell{=}1] \leftarrow \ell$                      ▷ Assign label to voxelized region
7: **end for**
8: $I_{\text{syn}} \leftarrow 0$
9: **for** $\ell = 1$ **to** $K$ **do**
10:     $U \leftarrow \{\mathbf{p} \mid L_\uparrow(\mathbf{p}) = \ell\}$
11:     $I_{\text{syn}}[U] \leftarrow \text{Normal}(\mu_\ell, \sigma_\ell^2)$                     ▷ IID draws
12: **end for**
13: $I_{\text{syn}} \leftarrow \text{GaussianBlur}(I_{\text{syn}}, \sigma{=}0.75)$
14: **return** $(I_{\text{syn}}, L_\uparrow, \tilde{A})$

---

**Randomization and metadata.** All stochastic draws are seeded per subject (seed = `base_seed` + `subject_id`) for exact reproducibility.[2] All outputs are written as NIfTI files with same origin and directions as the original images, but with a voxel spacing of $t = 0.25\,\text{mm}$.

### L.2 BASELINES

We augment FireANTs (Jena et al., 2024a) with FFDP to enable scalable image registration at high resolutions. The methods and their hyperparameter settings are described below:

- **CLAIRE**(Mang et al., 2019): CLAIRE is a velocity-based diffeomorphic registration framework optimized for distributed GPU/CPU execution via MPI. We use the official repository inside a custom multi-GPU Docker image that adds CUDA-aware Open MPI (v4.0.3; CUDA 11), since the official container supports only single-GPU runs. We launch one MPI rank per GPU and bind each rank to a distinct device via a lightweight wrapper that maps `OMPI_COMM_WORLD_RANK` to `CUDA_VISIBLE_DEVICES`, enabling data-parallel execution across $N$ GPUs. We keep default solver settings, request deformation maps (`-defmap`), and set the continuation parameter `-betacont 7.75e-04` following the official examples; all other hyperparameters use documented defaults, including the iteration cap (`-maxit 50`). Full-resolution runs use 4 GPUs (4 ranks), while half/quarter resolutions use a single GPU (1 rank).

- **ITK-DReg**(itk): ITK-DReg is a CPU-based, distributed, out-of-memory registration framework built on ITK and `dask.distributed`, formulating registration as block-wise map–reduce. We use the `itk_dreg` pipeline with Elastix in deformable-only B-spline mode: the metric is AdvancedNormalizedCorrelation with three pyramid levels (`NumberOfResolutions=3`, `GridSpacingSchedule=[4,2,1]`), optimized via AdaptiveStochasticGradientDescent with `MaximumNumberOfIterations=500`. We use random sampling with `NumberOfSpatialSamples=5000` (refreshed each iteration). Registration operates in voxel units with `FinalGridSpacingInVoxels=20` and `BSplineTransformSplineOrder=3`. To scale to high resolutions, the fixed image is tiled into $256^3$-voxel chunks with 25% overlap per axis; per-block results are reduced to a global displacement field defined on a grid subsampled by a factor of 4. ITK threading is set via `SetGlobalDefaultNumberOfThreads=24` (reported `GetGlobalMaximumNumberOfThreads=128`).

- **FireANTs + FFDP (Ours)**(Jena et al., 2024a): We use the official repository and scripts, except for our proposed modules (grid sampler, LNCC, and Mutual Information). We perform registration using the multi-scale of $4mm, 2mm, 1mm, 500\mu m$, and $250\mu m$ for $200, 200, 200, 100, 25$ iterations. We also truncate the optimization at $1mm$ and $500\mu m$ resolutions to verify the performance of the method at downsampled resolutions. We use

---

[2]We use `base_seed` = 2025 in our experiments.

our Fused LNCC implementation with a window size of 7, and a learning rate of 0.5. The smoothing kernels are chosen with a $\sigma_{warp} = 0.5$ pixels, and $\sigma_{grad} = 1.0$ pixels.

We also evaluate against state-of-the-art deep learning methods:

- **SynthMorph**(Hoffmann et al., 2021): SynthMorph uses an acquisition-free synthetic data generation pipeline to train a registration network. We use the default `mri_synthmorph` script provided by the vendor. Since all images are affine-aligned, we use the deformable registration mode `-m deform` with a regularization weight of `-r 0.25`.

- **Vector-Field Attention** (Liu et al., 2024c): Vector-Field Attention (VFA) is a weakly-supervised learning-based method utilizing a novel attention module to retrieve per-pixel correspondence based on feature similarity. We evaluate using the pretrained model (trained on OASIS data with weak label supervision) provided in the official repository.

- **UnigradICON** (Tian et al., 2024): UnigradICON is a foundational registration model by training on a composite dataset consisting of lung CT, knee MRI, Abdomen CT, brain MRI, totalling more than 3 million image pairs, of which 4000 image pairs are sampled per epoch to mitigate data imbalance. The model is trained with a bidirectional similarity loss and an inverse consistency loss. UnigradICON also provides an instance optimization based postprocessing step to improve the registration performance. We use the pretrained model and scripts provided in the official repository, and compare performance with and without the instance optimization step.

- **TransMorph** (Chen et al., 2022): TransMorph is one of the first successful application of transformer-based architectures for image registration, marking a departure from traditional convolutional architectures. Compared to other convolutional architectures, TransMorph demonstrates higher performance under domain shift Jian et al. (2024); Jena et al. (2025; 2024b) among the deep learning methods. We use the pretrained model (`TransMorph-Large` trained on the OASIS dataset) that is provided in the official repository.

- **Anatomix + ConvexAdam** (Dey et al., 2025): Anatomix is a feature extractor that is trained to anticipate strong domain shift at training time and uses contrastive learning to extract domain-agnostic features that mitigate the effect of nuisance factors. Anatomix shows strong results on zero-shot registration on abdomen and myocardium. We use the pretrained model and scripts provided in the official repository.

We also acknowledge Quicksilver (Yang et al., 2017) as a relevant baseline that performs patch-based registration. However, despite our best efforts with containerizing the environment (the dependencies are no longer available or supported on modern hardware), we were unable to run this baseline on our system.

All deep learning methods are tested on $1mm$, $500\mu m$, and $250\mu m$ resolutions. On $500\mu m$ and $250\mu m$ resolutions, all methods run out of memory on a single NVIDIA A6000 GPU, and the methods do not provide infrastructure to run on multiple GPUs. We adopt the patch-based registration strategy adopted by the literature on high-resolution registration methods for histology (Wodzinski et al., 2024; Lotz et al., 2015; Liang et al., 2021) as additional baselines with the above deep learning models as registration backends. We choose (Hoffmann et al., 2021; Dey et al., 2025; Tian et al., 2024) as general-purpose deep learning methods to mitigate the effect of domain shift due to patch-based registration at higher resolutions, and (Liu et al., 2024c; Chen et al., 2022) as methods that are trained with weak label supervision on the OASIS dataset to verify performance at 1mm resolution and observe the performance at higher resolutions.

**Robust HD90 (Cumulative)**   Hausdorff distance is a widely adopted boundary-based metric in medical image registration. The conventional definition of HD90 (the 90th percentile Hausdorff distance) simply reports the 90th percentile, but does not provide an average performance for all surface boundaries. In contrast, we employ a modified formulation, which we denote as *cumulative HD90*, designed to provide a more stable and comprehensive estimate. Specifically, rather than selecting the single distance value at the 90th percentile, we compute the mean of all surface distances

up to the 90th percentile. Formally, given sorted distances $\{d_i\}_{i=1}^N$, we compute

$$\text{HD}_{90}^{\text{cu}} = \frac{1}{k} \sum_{i=1}^{k} d_i, \quad k = \lfloor 0.9\,N \rfloor.$$

Distances are computed bidirectionally between ground-truth and predicted surfaces using isotropic voxel spacing, and the final HD90 is defined as the maximum of the two directional estimates.

### L.3 ADDITIONAL RESULTS AND DISCUSSION

Table 5 shows the performance comparison for all methods at different resolutions. All methods using full-context (CLAIRE, ITK-DReg, FireANTs) show improvement in performance with resolution, while all deep learning methods degrade in performance due to (a) progressive domain shift at higher resolutions, even for models trained on multiple or synthetic data, and (b) unlike image registration for histology slides, volumetric datasets like these require large deformations, and patch-based methods do not provide the context to perform well at higher resolutions. In terms of efficiency, our method is substantially more efficient, both on terms of wall clock time, and the total GPU-hours consumed.

Although CLAIRE proposes a distributed GPU framework, the usage of scaling-and-squaring (which requires performing an integral and its adjoint computation every iteration) and other line search subroutines consume a considerable amount of resources. On the faux-OASIS dataset at full resolution, CLAIRE runs out of memory with 1 and 2 GPUs, and does not work on 3 GPUs due to indivisibility of the image size by 3. So the minimum number of GPUs required to run CLAIRE is 4. Our method runs on a single GPU, but does not require the image sizes to be divisible by the number of GPUs, or any other qualitative constraints, allowing researchers to simply plug in their inputs and run their workflows. For large-scale volumetric image registration problems, our method achieves three orders of magnitude of speedup over CLAIRE while enabling multimodal support and arbitrarily loss functions of choice.

### L.4 QUALITATIVE RESULTS

Qualitative results are shown in Figs. 16 to 18. With the exception of CLAIRE, ITK-DReg, and Ours, all methods get progressively worse as the resolution increases.

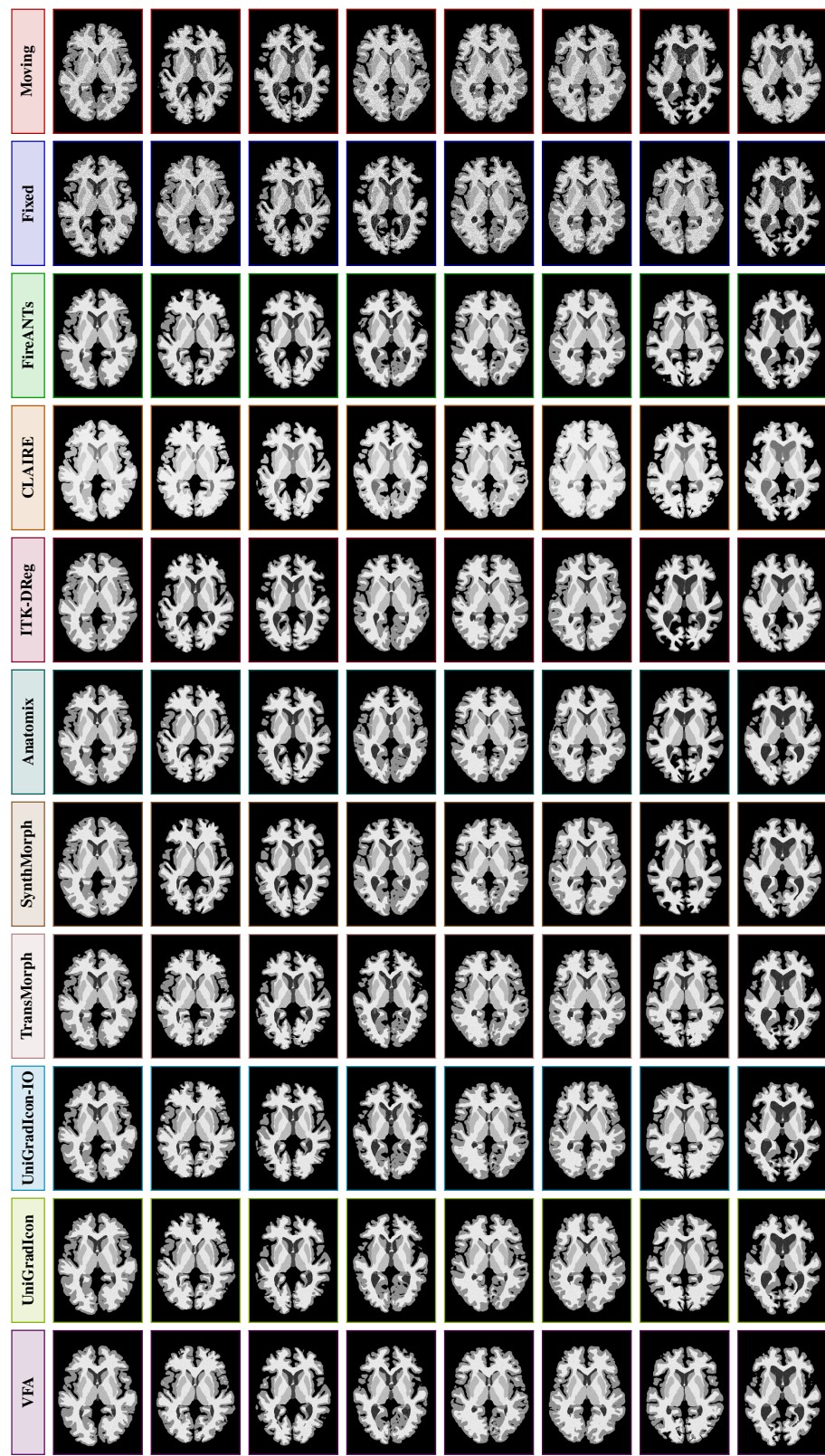

**Figure 16:** Qualitative comparison of registration results at **1 mm**. Each row corresponds to the moving image, fixed image, or one of the registration methods, with 8 representative slices per row. The comparisons illustrate visual alignment quality and anatomical consistency across methods.

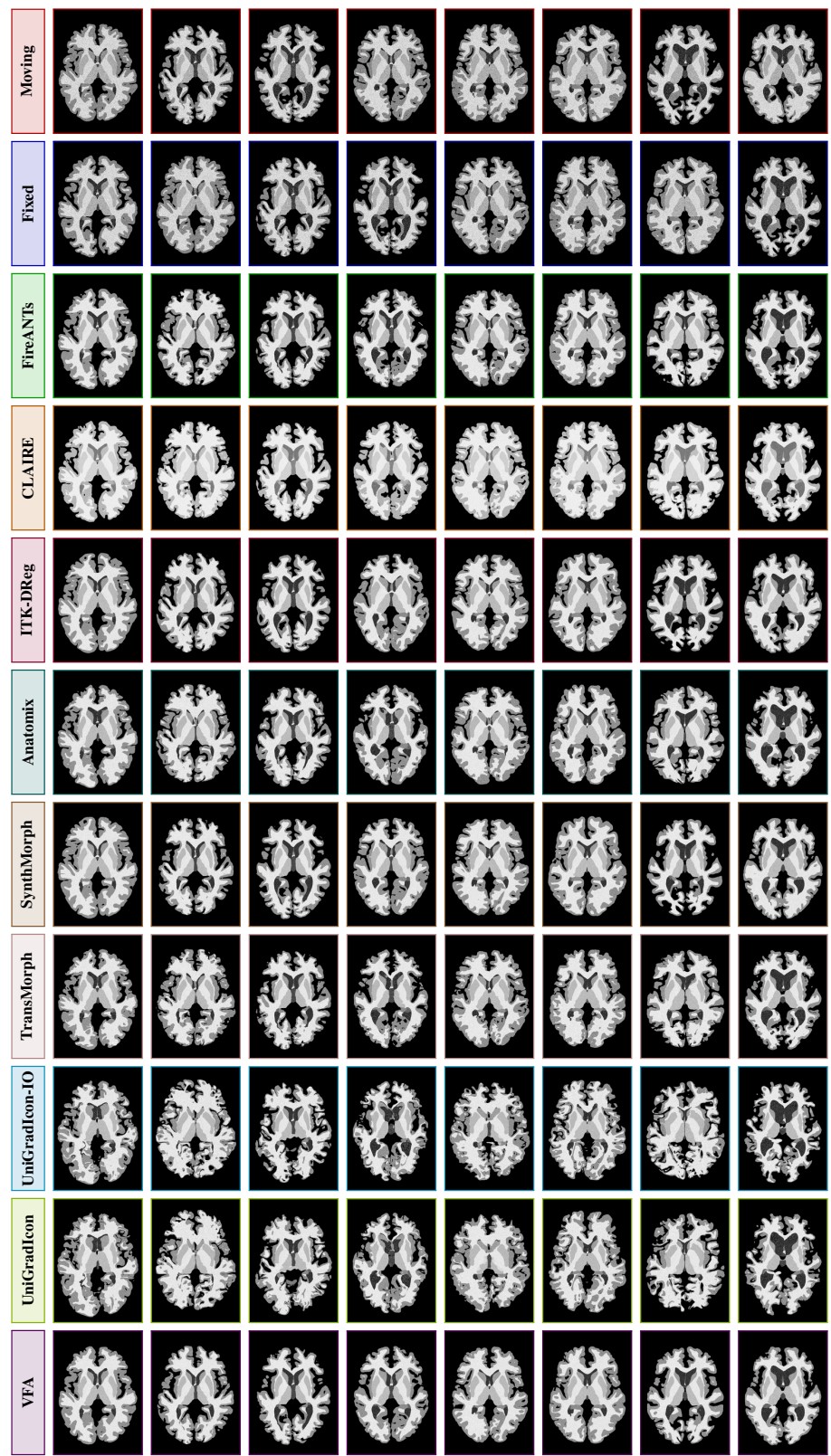

**Figure 17:** Qualitative comparison of registration results at **500um**. Each row corresponds to the moving image, fixed image, or one of the registration methods, with 8 representative slices per row. The comparisons illustrate visual alignment quality and anatomical consistency across methods.

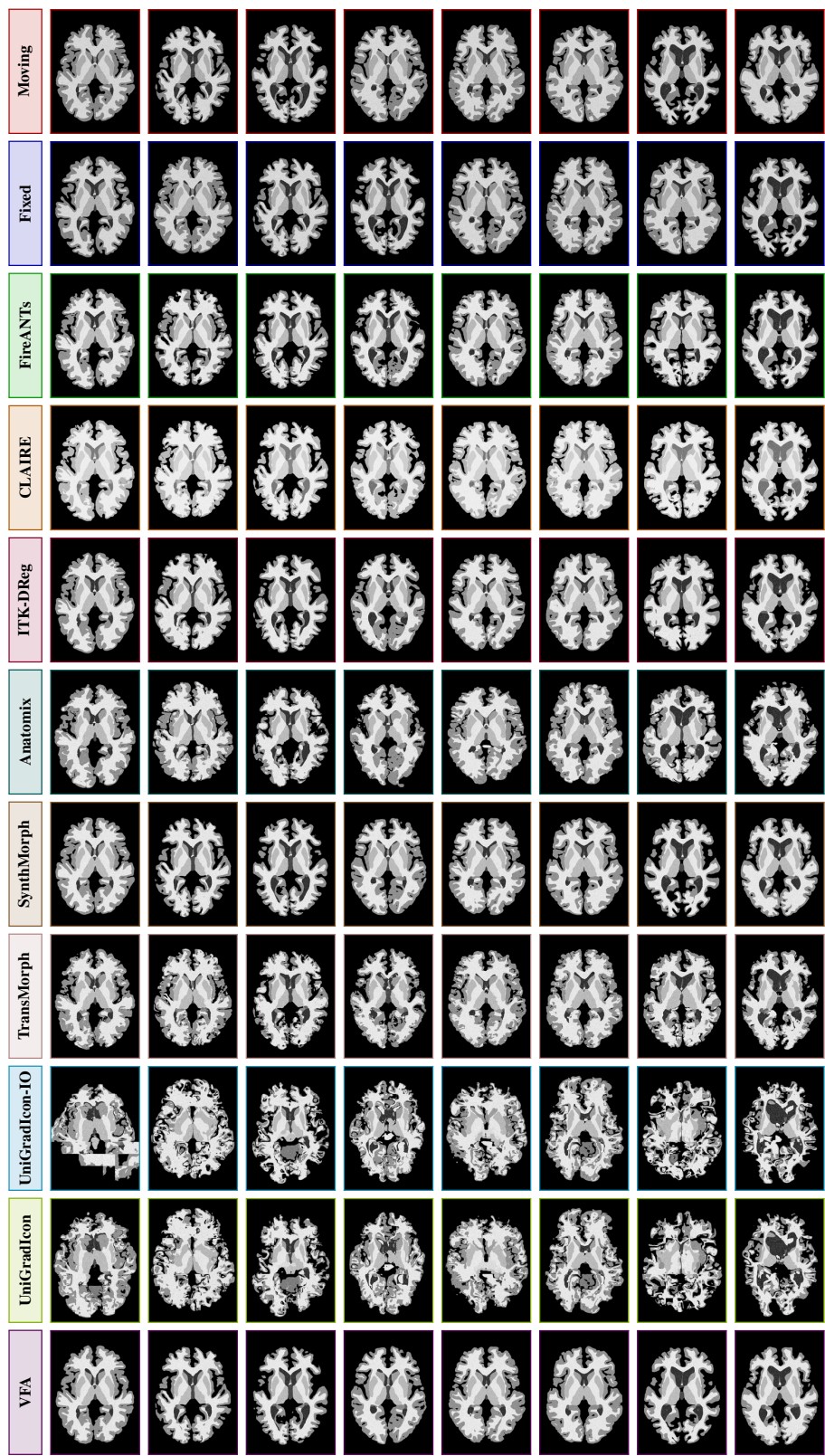

**Figure 18:** Qualitative comparison of registration results at **250um**. Each row corresponds to the moving image, fixed image, or one of the registration methods, with 8 representative slices per row. The comparisons illustrate visual alignment quality and anatomical consistency across methods.

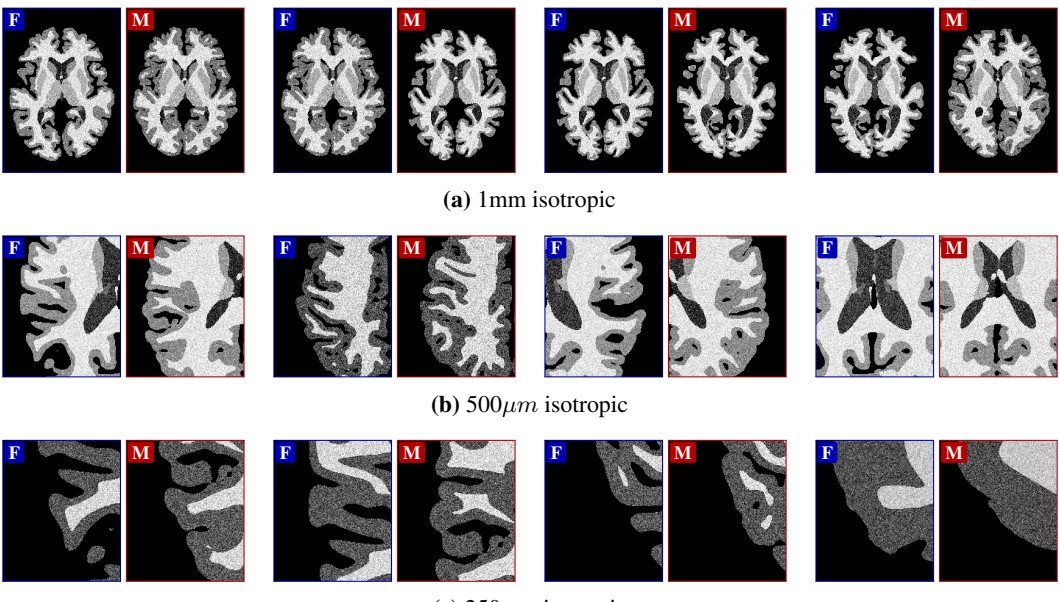

**(a)** 1mm isotropic

**(b)** 500$\mu m$ isotropic

**(c)** 250$\mu m$ isotropic

**Figure 19: Patch pairs seen during registration for patch-based methods**: For the 1mm isotropic images, there is only a single patch, i.e. the entire image. Deep learning methods utilize the global spatial context to perform accurate registration. At 500$\mu m$ isotropic, the patches still have large spatial context, but the images are out-of-distribution, leading to *degraded* performance Fig. 5a. At 250$\mu m$ isotropic, there is no meaningful spatial context and the patches are completely out-of-distribution, leading to poor performance for all patch-based methods.

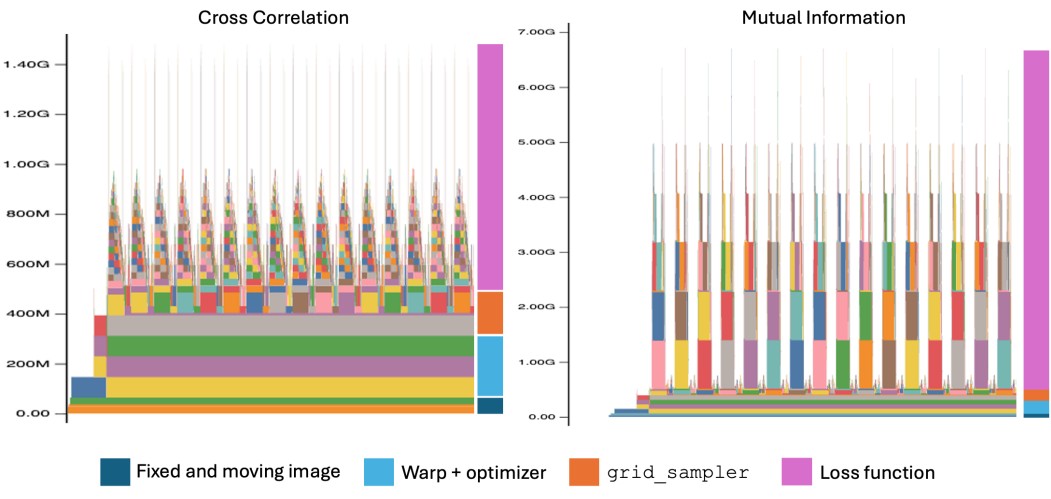

**Figure 20:** Flamegraph of FireANTs for Cross Correlation (left) and Mutual Information (right) losses on the OASIS dataset. The flamegraph is annotated on the right with colored blocks denoting the memory overheads for the fixed and moving images, the warp field and its optimizer state, the `grid_sampler` operation, and the loss function. Most of the computational overhead is due to the loss function, followed by the `grid_sampler` operation. This motivates the use of fused kernels to eliminate intermediate memory overheads.