# OpenReview forum: "A Scalable Distributed Framework for Multimodal GigaVoxel Image Registration"
_ICLR.cc/2026/Conference — ICLR 2026 Oral_

### Official Review · Reviewer_hXkU · 2025-10-25

**Soundness:** 2
**Presentation:** 1
**Contribution:** 2
**Rating:** 2
**Confidence:** 3

**Summary:**

This paper introduces FFDP (Flash Fused Distributed Primitives) as a scalable and distributed framework for large-scale image registration. The main idea is to fuse non-GEMM operations using IO-aware kernels and distributed tensor sharding, enabling registration of multi-billion-voxel images. The proposed framework includes fused kernels for single-GPU efficiency and GridParallel plus RingSampler modules for distributed multi-GPU scalability without full-image allgather operations. Experimental results show that FFDP significantly accelerates registration and reduces memory consumption for both iterative optimization-based and deep learning-based methods.

**Strengths:**

1. System contribution:
FFDP elegantly adapts concepts like ring topology and sharded synchronization to a domain with boundary dependencies.

2. Significant scalability results on iterative methods:
Demonstrates unprecedented registration (11 B + parameters) within realistic compute budgets.

**Weaknesses:**

1. The introduction part does not demonstrate the motivation for full-resolution registration:
The introduction does not sufficiently justify the necessity of performing registration at full resolution. While image acquisition capabilities have improved, in many biomedical and clinical applications registration on downsampled images is adequate. The authors should provide stronger evidence or references showing that downsampling significantly degrades registration accuracy or downstream analysis.

2. Limited benefit for deep learning pipelines:
The proposed framework provides modest gains (16.5% and 24.7% memory reduction) for deep learning networks comparing with it in iterative methods. It would be helpful if the authors could analyze whether further optimizations could narrow this gap.

3. Combined Introduction and Related Work:
The paper merges the introduction and related work into a single section, which reduces readability and makes it harder to follow the motivation versus prior context.

4. Poorly organized experiments:
The experimental section is difficult to follow and lacks clear structure.

**Questions:**

1. Could you provide evidence or references that shows the necessity of full-resolution registration?

2. In Table 1, what do the three “Baseline” rows represent, what do “Top” and “Bottom” refer to, and which dataset was used to generate Table 1? Besides, Table 1 is not referenced in the main paper.

3. Regarding the faux-OASIS dataset experiment, you register the images at a downsampled resolution and perform patchwise registration followed by mosaicing of the final deformation for deep learning methods. This setup may adversely affect their performance, as many hyperparameters proposed in their paper(such as the weighting of the smoothness term) are designed for whole-image registration rather than patchwise processing. Moreover, you did not report the performance of these methods on the original OASIS dataset. If the deep learning methods achieve better performance on the original dataset, it would suggest that your experimental setup is flawed.

4. In Figure 5, you denote “fire-ants with the proposed method” as Ours and compare it with other methods. However, this comparison appears unfair since you did not include the performance of fire-ants itself. It is highly likely that fire-ants alone could outperform many of the other baselines. You should include the performance of the original fire-ants for a fair comparison. If memory limitations are a concern, you may apply the same two modifications used for the deep learning–based methods.

---

> ### Author Response · Authors · 2025-11-17
> **Authors' Response (1/n)**
>
> We appreciate the Reviewer for their insightful and constructive feedback. We are glad that the Reviewer finds our systems contributions elegant and relevant to the problem, and significant scalability results shown for iterative methods on modest compute resources. We reiterate that the Ring Sampler simply does not borrow the ring topology from Ring Attention, but also deconstructs the bilinear interpolation operation into compact subproblems that allows interleaving communication and computation. Note that the “optimal subproblem construction” is novel for bilinear interpolation, and is fairly non-trivial for other interpolations (e.g.spline-based).
>
> We address the remaining concerns below.
>
> **The introduction part does not demonstrate the motivation for full-resolution registration: The introduction does not sufficiently justify the necessity of performing registration at full resolution.**
>
> The introduction is kept concise while highlighting the main bottlenecks of existing technical methods to adhere to the page limit. Inspired by the organization of sections in other works [1,2,3,4,5,6,7], we initially performed an extended discussion of related work in the Appendix.  Following the Reviewer’s suggestion, we have added a Related Work section and extended discussion in the appendix that provides cited evidence of the requirements of large-scale registration and the existing gap between acquisition and registration capabilities.
>
> **The authors should provide stronger evidence or references showing that downsampling significantly degrades registration accuracy or downstream analysis.**
>
> Our evaluation on the synthetic dataset (Figure 5) provides strong evidence that registering at the original 250μm resolution provides improved anatomical overlap compared to significantly downsampled resolution (1mm). This performance gap is significant for smaller structures, shown by more than a 10 point increase measured by InvDice score (0.494 to 0.597). A 10 point improvement in Dice scores of smaller structures simply by increasing resolution (while keeping the algorithm constant) is strong evidence of the fact that high resolution registration is critical for registering small-scale structures. Since the goal of most high resolution human studies is to reliably align small structures such as cortical layers, hippocampal subfields, cortical lesions [9,10, 13, 15], registering at native resolution is crucial to ensure accurate quantitative downstream analysis and characterization of neuropathology.
>
> **Limited benefit for deep learning pipelines: The proposed framework provides modest gains (16.5% and 24.7% memory reduction) for deep learning networks comparing with it in iterative methods. It would be helpful if the authors could analyze whether further optimizations could narrow this gap.**
>
> The statement is partially correct but paints an incomplete picture. Our primary result for deep learning shows that optimizing the LNCC and interpolator kernels alone improves training time by up to 6.14x - reducing training time from a week to a day. This is a very strong result that can dramatically accelerate registration network training, experimentation and ablation studies. This speedup does not come at the cost of accuracy or memory - we show a memory reduction of up to 24.7% and a slight improvement in Dice score.
>
> For deep learning methods for registration, the primary memory bottleneck is the activation map overhead, which is not amenable to further optimization without tradeoffs. To reduce memory usage, standard deep learning tricks like activation checkpointing and CPU offloading can be used, at the cost of increased training time. The savings in runtime from our kernels can now be repurposed to save memory depending on the tradeoff desired by a practitioner.  Existing deep learning methods are not well positioned to tackle tasks with high resolution due to their large activation memory overhead, but our results show that optimizing a single (bottleneck) layer reduces memory consumption by up to 24% and achieves more than 6x speedup.

---

> > ### Author Response · Authors · 2025-11-17
> > **Authors' Response (2/n)**
> >
> > **"Combined Introduction and Related Work: The paper merges the introduction and related work into a single section, which reduces readability and makes it harder to follow the motivation versus prior context. "**
> >
> > We provided an extended discussion of Related Works in the Appendix, inspired by other papers [1-8] to provide extensive coverage of the related work while adhering to the page limit. Since the post-review discussion allows a total of 10 pages, we have added key Related Work after the Introduction and refer to an extended discussion in the Appendix.
> >
> > **Poorly organized experiments: The experimental section is difficult to follow and lacks clear structure.**
> >
> > Lines 342-348 (now Lines 395-401) provide a summarized organization of the experimental section, enumeration of the baselines used in the study, followed by subsections highlighting improvement in performance, a high-resolution multimodal registration demo, and comparative analysis on a simulated dataset to show the benefits of large-scale registration and the suitability of existing methods to tackle the task, followed by ablation studies.
> >
> > We find it hard to address the Reviewer’s concern without more details. We therefore invite the Reviewer to further clarify the sections which are harder to follow so we can adequately address it in the revised paper.
> >
> > **In Table 1, what do the three “Baseline” rows represent, what do “Top” and “Bottom” refer to, and which dataset was used to generate Table 1? Besides, Table 1 is not referenced in the main paper.**
> >
> > “Baseline” represents the baseline (and official) implementation of TransMorph. “Ours” represents the same implementation but using our fused kernels for grid sampling, LNCC, and MI. We have added the clarification in the paper.  Table 1 is referred to in the main paper in Section 4.1, Line 372  (Line 416 in updated paper). We have added another reference in Line 413.
> >
> > **Regarding the faux-OASIS dataset experiment, you register the images at a downsampled resolution and perform patchwise registration followed by mosaicing of the final deformation for deep learning methods. This setup may adversely affect their performance, as many hyperparameters proposed in their paper (such as the weighting of the smoothness term) are designed for whole-image registration rather than patchwise processing.**
> >
> > Our choice of deep learning methods includes methods that claim foundational or domain-agnostic performance (VFA, UniGradICON, Synthmorph, Anatomix). No methods have any specific hyperparameter tuning done for patches - so the methods are compared fairly under identical conditions. Moreover, we compare with SynthMorph and Anatomix that are trained on synthetically generated shapes and do not have a bias towards whole or patched brain images. The degradation in these methods with increased resolution can therefore be attributed to lack of context encountered in patchwise registration instead of train-test domain shift. This bolsters our argument that standard tricks used in histology registration (i.e. mosaicing) cannot be readily applied for large-scale volumetric registration, necessitating large scale image registration.
> >
> > **Moreover, you did not report the performance of these methods on the original OASIS dataset. If the deep learning methods achieve better performance on the original dataset, it would suggest that your experimental setup is flawed.**
> >
> > Our synthetic dataset closely emulates the intensity statistics of the OASIS dataset. Similar pipelines are also used in training foundational models [17, 18] to generate contrast-agnostic representations. On the 1mm resolution of our synthetic dataset,  the deep learning methods (TransMorph, VFA, SynthMorph) indeed achieve the performance at par with the OASIS dataset. This validates both the performance of the deep learning methods on the 1mm resolution, and the fidelity of the synthetic dataset. However, as the resolution increases, the full-context, native-resolution registration of our method leads to improved dice scores compared to the best possible result at 1mm resolution, especially for smaller structures (with up to a 10 point increase in inverse-weighted Dice Score).

---

> ### Author Response · Authors · 2025-11-17
> **Authors' Response (3/n)**
>
> **In Figure 5, you denote “fire-ants with the proposed method” as Ours and compare it with other methods. However, this comparison appears unfair since you did not include the performance of fire-ants itself. It is highly likely that fire-ants alone could outperform many of the other baselines. You should include the performance of the original fire-ants for a fair comparison. If memory limitations are a concern, you may apply the same two modifications used for the deep learning–based methods.**
>
> This is an oversight on our part. We have now added the performance of the original FireANTs to Figure 5. We use a patch size of 256 for FireANTs to ensure direct comparability with the other baselines that also operate at similar patch sizes, although a higher patch size is possible due to the memory efficiency of the baseline FireANTs implementation. We observe that FireANTs performance is better than most deep methods, and at 250μm it only underperforms CLAIRE and Ours which are full-context methods - consistent with our hypothesis that patch based methods are asymptotically weaker.
>
> **While image acquisition capabilities have improved, downsampled registration is often sufficient in biomedical and clinical contexts.**
>
> We respectfully disagree with this statement. We provide literature and prior work that provides compelling evidence for the necessity of performing registration on large resolution images and the limitations of current methods.
>
> **Neuroimaging in humans**. Although in-vivo MRI–MRI registration occurs at millimeter resolution, many neuroanatomical studies rely on high-resolution ex-vivo MRI, blockface, and histology to integrate structural, molecular, and cytoarchitectural information [8, 9]. In-vivo MRI is limited by scan time, so ex-vivo scans ($\sim$100–500 µm) and histology ($\sim$10 µm) serve as ground truth. Accurate alignment of cortical layers, axonal projections, and small nuclei is essential for understanding pathology; e.g., AD-related atrophy in hippocampus and entorhinal cortex requires ≤0.5 mm MRI [9, 10], PD-related degeneration of the  subthalamic nucleus requires ≤0.7 mm or ex-vivo imaging [13, 28], and MS cortical lesions often require ex-vivo MRI and histopathology [14, 15]. High-resolution multimodal projects (Allen Brain Atlas [16],  Seattle Alzheimer's Disease Brain Cell Atlas (SEA-AD) and Human Mouse Brain Atlas (HMBA) consortia) depend critically on fine-scale registration. The 100 µm ex-vivo and 250 µm in-vivo images used in our paper [11, 12] are intended as high-res templates, but current methods struggle computationally; our approach democratizes their use.
>
> **Pulmonary imaging.** Lung CT routinely achieves sub-millimeter resolution [29], yet deep methods often downsample due to memory limits [30]. In EMPIRE10 (0.7 mm/voxel resolution), landmark error reaches 0.649 mm using iterative solvers; but in the Learn2Reg Lung CT challenge (1.25–1.75 mm, downsampled), top methods reach only 1.83 mm error. This demonstrates that rich native-resolution features enable subvoxel accuracy, while downsampling degrades performance.
>
> **Model organisms – rodents.** Large initiatives (e.g., BICCN) acquire multimodal data at extreme resolutions (MRI, STPT <1 µm [19], microscopy, stains, transcriptomics). However, templates like the Allen CCFv3 (10 µm) are created by upsampling registrations run at 25 µm due to compute limits; reproducing these steps would require ~7.26 CPU-years per template iteration [22].  Resource constraints similarly force ANTsX mouse pipelines to use 50 µm instead of 10 µm [20], and developmental atlases to downsample [21, 23]. Our method completes analogous registrations in minutes on 8 GPUs, enabling whole-brain integration at native scale.
>
> **Model organisms – zebrafish.** Although AZBA data are acquired at 4 µm, they are resampled to 8 µm due to system constraints [25], and the required tools [26] primarily support distributed clusters. Brain-wide cellular imaging of transgenic lines [27] likewise depends on large compute clusters, limiting accessibility.
>
> **Model organisms – migratory birds.**  STPT data are acquired at 2×2×5 µm but atlases are produced at 20 µm [24], highlighting the gap between acquisition and computational feasibility. Across human and model-organism studies, high-resolution registration is crucial for accurate anatomical alignment and downstream analyses, yet currently limited by compute. Our scalable framework enables native-resolution multimodal registration with modest resources.
>
> This discussion is kept brief in our response; we provide a substantially more in-depth discussion of the necessity of image registration for large images in the Related Work and Extended Related Work (Appendix) in the updated paper.
>
> We hope that we have addressed all the Reviewer’s concerns, and we encourage the Reviewer to revise their score. We also look forward to also engaging in followup discussions and any additional clarifications we can provide.

---

> > ### Author Response · Authors · 2025-11-17
> > **Authors' Response (References)**
> >
> > [1] https://openreview.net/pdf?id=6ve2CkeQe5S
> >
> > [2] https://openreview.net/forum?id=H4DqfPSibmx
> >
> > [3] https://openreview.net/pdf?id=fCeUoDr9Tq
> >
> > [4] https://openreview.net/pdf?id=mEpqHvbD2h
> >
> > [5] https://openreview.net/pdf?id=HJei-2RcK7
> >
> > [6] https://openreview.net/pdf?id=qssVptHTPN
> >
> > [7] https://openreview.net/pdf?id=4Zz5UELkIt
> >
> > [8] NextBrain: A probabilistic histological atlas of the human brain for MRI segmentation  https://www.nature.com/articles/s41586-025-09708-2
> >
> > [9] Postmortem imaging reveals patterns of medial temporal lobe vulnerability to tau pathology in Alzheimer's disease https://pubmed.ncbi.nlm.nih.gov/38839876/
> >
> > [10] Atrophy in the parahippocampal gyrus as an early biomarker of Alzheimer’s disease https://pmc.ncbi.nlm.nih.gov/articles/PMC3041901/
> >
> > [11] 7 Tesla MRI of the ex vivo human brain at 100 micron resolution  et.al. https://www.nature.com/articles/s41597-019-0254-8
> >
> > [12] T1-weighted in vivo human whole brain MRI dataset with an ultrahigh isotropic resolution of 250 μm  https://pubmed.ncbi.nlm.nih.gov/28291265/
> >
> > [13] Dopamine and Parkinson's Disease https://www.ncbi.nlm.nih.gov/books/NBK6271/
> >
> > [14] Imaging cortical multiple sclerosis lesions with ultra-high field MRI https://pmc.ncbi.nlm.nih.gov/articles/PMC8517925/
> >
> > [15] Improved Visualization of Cortical Lesions in Multiple Sclerosis Using 7T MP2RAGE https://pubmed.ncbi.nlm.nih.gov/29439120/
> >
> > [16] https://atlas.brain-map.org/
> >
> > [17] Brain-ID: Learning Contrast-Agnostic Anatomical Representations for Brain Imaging https://link.springer.com/content/pdf/10.1007/978-3-031-73254-6_19.pdf
> >
> > [18] PEPSI: Pathology-Enhanced Pulse-Sequence-Invariant Representations for Brain MRI (MICCAI) https://arxiv.org/pdf/2403.06227
> >
> > [19] Protocol for using serial two-photon tomography to map cell types and cerebrovasculature at single-cell resolution in the whole adult mouse brain https://pmc.ncbi.nlm.nih.gov/articles/PMC10037193/
> >
> > [20] The ANTsX Ecosystem for Mapping the Mouse Brain https://pubmed.ncbi.nlm.nih.gov/38746199/
> >
> > [21] Developmental mouse brain common coordinate framework  https://www.nature.com/articles/s41467-024-53254-w
> >
> > [22] Dense image registration through MRFs and efficient linear programming  https://www.sciencedirect.com/science/article/pii/S1361841508000297?via=ihub
> >
> > [23] epDevAtlas: mapping GABAergic cells and microglia in the early postnatal mouse brain  https://www.nature.com/articles/s41467-025-64549-x
> >
> > [24] Mapping the magnetoreceptive brain: A 3D digital atlas of the migratory bird Eurasian blackcap (Sylvia atricapilla)  https://www.biorxiv.org/content/10.1101/2025.03.04.641293v1.full.pdf
> >
> > [25] A 3D adult zebrafish brain atlas (AZBA) for the digital age https://elifesciences.org/articles/69988
> >
> > [26] Pydpiper https://github.com/Mouse-Imaging-Centre/pydpiper
> >
> > [27] Brain-wide cellular resolution imaging of Cre transgenic zebrafish lines for functional circuit-mapping https://elifesciences.org/articles/42687
> >
> > [28] Ultra-high-field 7T MRI in Parkinson's disease: ready for clinical use?-a narrative review https://pubmed.ncbi.nlm.nih.gov/37969629/
> >
> > [29] Evaluation of registration methods on thoracic CT: the EMPIRE10 challenge  https://pubmed.ncbi.nlm.nih.gov/21632295/
> >
> > [30] Learn2Reg Challenge: CT Lung Registration - Training Data  https://zenodo.org/records/3835682

---

> ### Comment · Reviewer_hXkU · 2025-11-28
> **Response to Authors’ First-Round Rebuttal**
>
> Thanks for your detailed replies and most of my concerns have been addressed. I have one remaining question: could you further clarify why your method achieves higher DICE scores even when operating at the same spatial resolution as other methods? From the current description, the method appears primarily focused on acceleration and memory efficiency. If you can clearly demonstrate how the proposed approach leads to accuracy improvements (specifically in DICE), I will be able to increase my score.

---

> > ### Author Response · Authors · 2025-11-30
> > **Clarification on performance**
> >
> > Our hypothesis is that accurate large-scale registration requires full context (i.e. FOV of the image patches), i.e. patchification won't work.
> >
> > At 500µm, there is probably enough context within patches that FireANTs (with LNCC) performs good registration. In comparison, in deep learning methods, the domain shift induced by the patches degrades accuracy significantly.
> >
> > At 250µm, full-context methods like CLAIRE outperform FireANTs because the effect of context is significant enough at a sufficiently large resolution.
> >
> > Our method (which is memory-efficient CUDA kernels applied on FireANTs) allows applying a robust method (FireANTs) at full resolution on a single GPU, which was not possible _operationally_ before. The primary improvement in performance occurs by performing registration at full scale, which is what our fused kernels allow a practitioner to do. Hopefully this clarifies the concern. We thank the Reviewer for proactively engaging in discussion and improving the quality of the paper.

---

### Official Review · Reviewer_bohM · 2025-10-29

**Soundness:** 3
**Presentation:** 3
**Contribution:** 4
**Rating:** 10
**Confidence:** 4

**Summary:**

The paper proposes a set of IO-aware fused kernels and a distributed framework enabling deformable image registration at extremely large scale. It significantly reduces the memory usage for interpolation, and 2 very common loss functions. Experimental results show significant acceleration for both classical iterative optimization and deep learning–based registration pipelines, and demonstrate the first native-resolution multimodal registration at extremely high spatial resolutions.

**Strengths:**

1. Enables giga-voxel registration in its native resolution that previously was infeasible.
2. Identifies and proposes solutions for major memory bottlenecks speeding up both iterative and learning based registration.
3. Demonstrates performance on multimodal registration of 250μm to 100μm images completing in 1 minute running on 8 gpus.
4. Ablates each components contribution.
5. Benchmarks the performance of registration algorithms on clinical scale and very high resolution brain images.
6. The language of the paper is clear.
7. I really enjoyed reading the paper.

**Weaknesses:**

**Weaknesses and questions**

1. I was wondering why the affine transformations is denoted as φ(x)=Ax+t. Isn’t it the case normally that the translation is incorporated in A along with rotation and scaling?
2. Although the scalability demonstrated on ex-vivo MRI is impressive, it is unclear why the method was not evaluated on multi-gigapixel histopathology datasets, which seem like an ideal target application given their fine-scale structural alignment requirements. Such experiments would directly validate the paper’s stated motivation and better highlight FFDP’s advantages over existing patch-based histology workflows.
3. While the paper emphasizes improvements for multimodal registration, LNCC is widely used for monomodal scenarios too, so the proposed optimizations appear broadly applicable. I am therefore unsure why the authors chose to position their contribution primarily around multimodal registration and not a general purpose registration with more speedup in the multi-modal case.
4. Not sure if I missed it, but are all methods (learning-based and iterative optimization ones) trained with the same objectives?
5. Multi-resolution strategies are commonly used in high-resolution registration to manage memory and computation, and while they suffer from similar limitations as those discussed in the paper, acknowledging them would improve completeness and situate the work more clearly within existing practice.
6. Regarding the learning-based baselines, their encoders are not designed for extremely high-resolution inputs, and while patchification is clearly suboptimal, it seems that alternative architectural designs (e.g., improved downsampling kernels in the feature space, larger capacity models, more spatially aware context handling) might yield stronger performance even if at higher memory cost. It would be helpful for the authors to comment on whether such adaptations were considered and how FFDP compares in those scenarios. I would be interested in the authors’ perspective on whether the limitations observed are inherent to deep approaches at scale or simply due to the baselines being used outside their intended operating regime.
7. No codebase or link to code is available.

**Questions:**

Please see the section above.

---

> ### Author Response · Authors · 2025-11-17
> **Authors' Response (1/3)**
>
> We thank the Reviewer for their thoughtful and encouraging feedback, which we believe has improved the quality of the manuscript. We are pleased that the Reviewer found the paper clear, enjoyable to read, and recognized the significance of enabling native-resolution giga-voxel registration, addressing key memory bottlenecks, and accelerating both iterative and learning-based pipelines. We appreciate the Reviewer’s acknowledgement of our multimodal high-resolution results, component-wise ablations, and comprehensive benchmarking. We address the Reviewer’s remaining concerns below.
>
> **I was wondering why the affine transformations is denoted as φ(x)=Ax+t. Isn’t it the case normally that the translation is incorporated in A along with rotation and scaling?**
>
> The translation can be incorporated in A if the coordinates x are homogenous coordinates (as is often the case in projective transform in computer vision and robotics applications). For 3D coordinates, A can be a 4x4 matrix or 3x4 matrix depending on whether the output of the transform itself is a homogenous coordinate or not. For sake of clarity, we assumed x is not a homogenous coordinate, but assuming otherwise would not change the subsequent analysis.
>
> **Although the scalability demonstrated on ex-vivo MRI is impressive, it is unclear why the method was not evaluated on multi-gigapixel histopathology datasets, which seem like an ideal target application given their fine-scale structural alignment requirements. Such experiments would directly validate the paper’s stated motivation and better highlight FFDP’s advantages over existing patch-based histology workflows.**
>
> This is an excellent question! A typical MRI to histology workflow would involve multiple registrations, i.e. premortem MRI ←→  postmortem MRI ←→ blockface imaging ←→ histology. Registering histology directly to premortem or postmortem is hard due to sectioning artifacts, uneven staining of the histology slides, usage of multiple different stains, tissue distortions from slicing and mounting, and missing or damaged sections. This requires a histology preprocessing and reconstruction step using blockface imaging as an intermediate reference point. Moreover, the lack of deep learning baselines for MRI to histology might make comparative analysis harder and more prone to errors or bias. Furthermore, we are not aware of public datasets or benchmarks for MRI to histology. Focusing on these aspects could possibly overshadow the technical contribution of this work.
>
> In this paper, we focus on the first part of the pipeline, i.e. in-vivo to ex-vivo MRI registration. The 7 Tesla ex vivo brain at 100μm resolution [1] is a representative dataset with multi-flip FLASH data highlighting GM-WM boundaries, excellent delineation of cortical layers, strong contrast of myelinated fiber bundles making it an excellent candidate for integration with microstructure imaging like histology. T1w scans are usually acquired for clinical and research studies on AD/FTD/other diseases, and we chose [2] as a high resolution scan to qualitatively evaluate the resolution of registered structures. Focusing on in-vivo to ex-vivo registration as an example (in Section 4.1) also naturally follows up to the synthetic 250μm in-vivo whole brain registration (Section 4.3). This natural progression also allows us to (a) fairly compare against established deep learning methods on a familiar domain (T1 weighted MRI at 1mm isotropic), (b) analyse whether post-hoc methods like mosaicing and downsampling improve performance of existing deep methods for larger resolutions, and (c) examine whether registration at native scale improves labelmap overlap, especially for smaller structures (since the goal of collecting data at extremely high resolutions is to better delineate smaller intricate structures).
>
> We believe this framing allows us to focus the discussion on the technical aspects of the work, while showcasing practical utility on a highly relevant target application. Completing the MRI to histology pipeline with our proposed components, and high-resolution template building and registration for life science applications like rodent, avian and zebrafish brain datasets forms the foundation for application-focused future work.

---

> ### Author Response · Authors · 2025-11-17
> **Authors' Response (2/3)**
>
> **Regarding the learning-based baselines, their encoders are not designed for extremely high-resolution inputs, and while patchification is clearly suboptimal, it seems that alternative architectural designs (e.g., improved downsampling kernels in the feature space, larger capacity models, more spatially aware context handling) might yield stronger performance even if at higher memory cost.**
>
> We agree that the standard learning-based architectures we benchmark (SynthMorph, TransMorph, VFA) were originally designed for relatively low-resolution volumes and do not natively support gigavoxel inputs due to memory bottlenecks. We intentionally included these methods without architectural redesign to assess the usability of off-the-shelf baselines on these large datasets under the current state of infrastructure. Notably, SynthMorph and Anatomix are trained on synthetic shapes and do not have a bias towards brain images or patches - so the degradation in the results are a limitation of patchification and not the method itself. Our new results (FireANTs with patch-based registration) also show similar trends.
>
> Currently, there are a few impediments to applying existing deep learning methods for large problems.
>
> First, the key memory bottleneck in applying deep learning to large problems is the activation overhead of the intermediates in the forward pass. Squashing this memory overhead requires smart architectural designs (improved downsampling kernels in the feature space, more spatially aware context handling) to reduce this memory overhead while preserving registration fidelity. However, we are not aware of any such design in the registration literature that explicitly addresses this issue for a significantly large image dataset. Alternatively, larger capacity models (i.e. a larger variant of an existing network) can achieve good performance with enough data but can be significantly expensive to train on large datasets. For instance, a “250μm OASIS datum” would require anywhere from 48 to 64 GPUs to train an existing SOTA deep learning network to accommodate the spatial extent of the activation maps. Moreover, most existing model parallel techniques are natively supported for transformers so many UNet-based models cannot scale even during training without a model parallel primitive like GridParallel (which we introduce in this paper).
>
> Second, applications like registration and atlas generation of MRI to histology, model organisms like rodent, zebrafish and avian brains are “data poor” compared to a clinical setting like in-vivo neuroimaging datasets (OASIS, LUMIR). Data collection, preprocessing, and dissemination efforts are significantly more challenging as well, as evident in the lack of large-scale established benchmarks or challenges in these domains. Therefore, training a deep learning method for such modalities is harder from a dataset perspective as well, even with access to large amounts of compute.
>
> **It would be helpful for the authors to comment on whether such adaptations were considered and how FFDP compares in those scenarios.**
>
> With larger compute budgets and a distributed deep learning framework, FFDP can play a key role in completing a distributed training pipeline by computing the moved image using ring sampler and computing the loss function in a distributed mode (the non deep learning bottlenecks in registration), in synchrony with model parallel techniques to shard the actual DL model across all available GPUs. However, FFDP is not designed to squash memory bottlenecks within the model architecture, thus requiring complementary innovations in reducing activation bottlenecks for deep learning methods.
>
> We did consider architectures like implicit neural representations as promising directions that can potentially substitute an explicit voxel grid with an implicit parameter space of a network, thereby bypassing materialization of large activation maps. We did find architectures that perform instance optimization with a neural representation as a prior [4, 5], but none that perform statistical learning from a dataset.

---

> ### Author Response · Authors · 2025-11-17
> **Authors' Response (3/3)**
>
> **I would be interested in the authors’ perspective on whether the limitations observed are inherent to deep approaches at scale or simply due to the baselines being used outside their intended operating regime.**
>
> Based on the previous discussion, we believe that the limitations observed are inherent to deep approaches at scale. In summary, deep learning for large datasets suffer from two main limitations, (a) excessively large activation maps that scale with image size, (b) relatively “data poor” and non-standard nature of data in these application areas. These scaling problems are fundamental to deep learning approaches rather than the baselines being used outside their intended operating regime. We did consider downsampling to 1mm as a “fix”, but it leaves more performance on the table that post-hoc techniques like mosaicing cannot recover. A promising alternative is using Implicit Neural Representations to avoid computing voxel maps altogether, but how to use it in a learning setting is unclear to the authors.
>
> **While the paper emphasizes improvements for multimodal registration, LNCC is widely used for monomodal scenarios too, so the proposed optimizations appear broadly applicable. I am therefore unsure why the authors chose to position their contribution primarily around multimodal registration and not a general purpose registration with more speedup in the multi-modal case.**
>
> We assume that the definition of multimodal registration subsumes unimodal registration (“multi” meaning any two modalities - even if the modalities are the same). We specifically highlight the multimodal support because (a) modern imaging workflows (MRI to histology studies, template building and registration for rodents, zebrafish, avian brains) are natively multimodal by integrating MRI with microscopy and sectioning tomography, and (b) there exists work that performs distributed optimization but only supports MSE loss [3], limiting its applicability for the aforementioned domain areas. We have clarified our scope of “multimodal” in the paper.
>
> **Not sure if I missed it, but are all methods (learning-based and iterative optimization ones) trained with the same objectives?**
>
> For all learned methods, we use their pretrained models. TransMorph, VFA, UnigradICON are trained on LNCC. SynthMorph is trained on Dice score of synthetically generated volumes, and Anatomix+ConvexAdam is run with MSE loss (using discrete optimization) on pretrained features.  CLAIRE only supports MSE, and ITK-DReg uses the AdvancedNormalizedCorrelation loss packaged in the Elastix package. We have added these details in Section K.2.
>
> **Multi-resolution strategies are commonly used in high-resolution registration to manage memory and computation, and while they suffer from similar limitations as those discussed in the paper, acknowledging them would improve completeness and situate the work more clearly within existing practice.**
>
> Multi-resolution strategies do not lower the peak memory consumption, since the registration at the final level (downsampling scale of 1x) determines the peak memory usage regardless of the number of iterations for which it is run. Running a multi-resolution scale at a maximum resolution lower than the native resolution of the image is equivalent to running registration on a downsampled image (with the downsampling factor equal to the max resolution of the multi-scale  optimization).
>
> To our knowledge, the primary benefit of multi-resolution based methods are typically proposed to avoid local minima associated with gradient-based methods by using a scale-space approach for registration. However, we are open to the Reviewer pointing us to related literature where multi-resolution is used to reduce peak memory and computation. Just to reiterate, FFDP applied to FireANTs is also a multi-scale algorithm.
>
> **No codebase or link to code is available.**
>
> We could not share the codebase due to licensing issues. We will release the library with user guides, documentation, and examples upon paper acceptance.
>
> We hope that we have addressed all the Reviewer’s concerns, and would be elated if the Reviewer would further champion our paper.
>
> ---
>
> [1] 7 Tesla MRI of the ex vivo human brain at 100 micron resolution  et.al. https://www.nature.com/articles/s41597-019-0254-8
>
> [2] T1-weighted in vivo human whole brain MRI dataset with an ultrahigh isotropic resolution of 250 μm  https://pubmed.ncbi.nlm.nih.gov/28291265/
>
> [3] https://pmc.ncbi.nlm.nih.gov/articles/PMC8513530/pdf/nihms-1631183.pdf
>
> [4] Implicit Neural Representations for Deformable Image Registration  https://proceedings.mlr.press/v172/wolterink22a.html
>
> [5] Robust deformable image registration using cycle-consistent implicit representations https://ieeexplore.ieee.org/stamp/stamp.jsp?tp=&arnumber=10268959

---

### Official Review · Reviewer_TnBN · 2025-10-30

**Soundness:** 3
**Presentation:** 3
**Contribution:** 3
**Rating:** 6
**Confidence:** 4

**Summary:**

In this paper, we propose the FFDP framework, which aims to solve the problem of insufficient algorithmic scalability in ultra-high resolution multimodal image alignment.FFDP designs three I/O-aware non-GEMM fusion kernels (composite implicit grid sampler, implicit Parzen window mutual information estimation, and efficient fusion inter-correlation) as well as distributed architectures (grid parallelism, ring sampler, and distributed loss computation) to realize the fast processing of tasks at the level of hundreds of billions of voxels. Experiments show that FFDP outperforms existing methods on both synthetic and real datasets, increasing the runtime speed by 6-7 times, reducing peak memory by 20-59%, and significantly reducing GPU computational overhead while maintaining higher alignment accuracy (e.g., Dice coefficient of 89.5% at 250µm resolution).

**Strengths:**

- Motivation is clear: It precisely targets the core contradiction between high-resolution acquisition capabilities and the insufficient scalability of existing algorithms in medical imaging, with real scenarios like 100µm ex-vivo human brain MRI clearly illustrating the necessity.
- Experiments are rich and comprehensively prove the effectiveness of its method: It covers multiple types of datasets and various dimensions of experiments, from component-level ablation to system-level comparison with mainstream baselines, fully demonstrating the method's validity.
- Performance improvement is significant

**Weaknesses:**

While this paper targets at improving efficiency, it would be better provide experimental evaluations on accuracy against counterparts.

**Questions:**

Please refer to weaknesses

---

> ### Author Response · Authors · 2025-11-17
> **Authors' Response**
>
> We thank the Reviewer for their positive assessment of FFDP, including the clear recognition of our motivation, the significance of addressing the scalability gap in ultra–high-resolution multimodal alignment, and the strength and breadth of our experimental validation. We appreciate the Reviewer’s acknowledgement that our fused kernels, distributed architecture, and empirical performance gains collectively represent meaningful progress for large-scale registration. We address the remaining concerns and clarifications below.
>
> **While this paper targets at improving efficiency, it would be better to provide experimental evaluations on accuracy against counterparts.**
>
> We do provide experimental evaluations on accuracy against counterparts. Our synthetic inter-subject registration (Section 4.3) emulates a typical high-resolution registration task between in-vivo and ex-vivo brain scans of the same modality (T1 weighted MRI). These high resolution registration tasks are crucial in advancing our understanding of neurodegenerative diseases and the effect of pathology on structural morphology [1-7] which typically collect data anywhere from 100-500μm depending on the study characteristics. However, such studies are dominated by custom iterative optimization-based workflows due to data scarcity, and to our knowledge, deep learning based registration frameworks are not adopted in these workflows.
>
> Our dataset is designed to (a) fairly compare against established deep learning methods on a familiar domain (T1 weighted MRI at 1mm isotropic), (b) establish whether post-hoc methods like mosaicing and downsampling improve performance for larger resolutions, and (c) examine whether registration at native scale improves labelmap overlap, especially for smaller structures (since the goal of collecting data at extremely high resolutions is to better delineate smaller structures including cortical layers, hippocampal subfields, cortical lesions to perform quantitative morphometry and analyse the effect of neurodegenerative diseases on local morphology).
>
> Our results in Figure 5 provide experimental evaluations on both the accuracy of labelmap overlap with increasing resolutions, and the accuracy versus GPU compute cost and accuracy versus wall clock time.
> We believe we have addressed all the Reviewer’s concerns and encourage the Reviewer to revise their score. We also look forward to more discussion or followup.
>
>
> [1] NextBrain: A probabilistic histological atlas of the human brain for MRI segmentation  https://www.nature.com/articles/s41586-025-09708-2
>
> [2] Postmortem imaging reveals patterns of medial temporal lobe vulnerability to tau pathology in Alzheimer's disease https://pubmed.ncbi.nlm.nih.gov/38839876/
>
> [3] Atrophy in the parahippocampal gyrus as an early biomarker of Alzheimer’s disease https://pmc.ncbi.nlm.nih.gov/articles/PMC3041901/
>
> [4] 7 Tesla MRI of the ex vivo human brain at 100 micron resolution  et.al. https://www.nature.com/articles/s41597-019-0254-8
>
> [5] T1-weighted in vivo human whole brain MRI dataset with an ultrahigh isotropic resolution of 250 μm  https://pubmed.ncbi.nlm.nih.gov/28291265/
>
> [6] Dopamine and Parkinson's Disease https://www.ncbi.nlm.nih.gov/books/NBK6271/
>
> [7] Imaging cortical multiple sclerosis lesions with ultra-high field MRI https://pmc.ncbi.nlm.nih.gov/articles/PMC8517925/

---

> > ### Comment · Reviewer_TnBN · 2025-11-28
> >
> > Thank the authors for their responses that resolved my concerns and I will increase my score.

---

### Official Review · Reviewer_oLeD · 2025-11-01

**Soundness:** 4
**Presentation:** 3
**Contribution:** 4
**Rating:** 8
**Confidence:** 3

**Summary:**

This paper makes large 3D medical image registration run fast and fit in GPU memory.
Key idea the reviewer finds: real bottlenecks at giga-voxel scale are not matrix multiplies but ops like grid sampling, LNCC, MI. They write fused, IO-aware CUDA kernels plus a ring sampler so GPUs don’t need full image copies. Result: 6–7× speed, up to ~59% memory saved, and a demo of 11.8B-parameter multimodal brain registration in about a minute on 8×A6000. This work is quire impressive.

The reviewer is familiar with registration but not with optimization work. But this work appears impressive and practical. Therefore, the reviewer will assign a low confidence score but high rating.

**Strengths:**

- Writing: Problem importance is well motivated: when the image is 100–1000× larger, registration algorithms did not keep up. It is an important gap.
- Distributed story is novel for registration.
- Empirical section is broad: classical (ANTs-like), learning (TransMorph, SynthMorph, VFA, UniGradICON), and purpose-built (CLAIRE, ITK-DReg). The 250µm setting, where every other method degrades while FFDP improves, is persuasive.
- The accelerations for existing code (TransMorph training 6× faster, big memory drops, and also for FireANTs) mean this work is really a critical progress.

**Weaknesses:**

- This work enables faster/larger registration, but it lacks a clear section on how to apply this. It does not specify which registration methods are directly supported, how to integrate the kernels, or which methods are not supported and why.
- Only mentioned related work on GPU/system acceleration. The author does not survey other GPU acceleration outside registration.

**Questions:**

See weakness.

---

> ### Author Response · Authors · 2025-11-17
> **Authors' Response**
>
> We appreciate the Reviewer’s careful reading and are encouraged by their positive assessment of our work, including the recognition of the practical impact of our IO-aware fused kernels, the novelty of our distributed strategy and ring sampler, and the scope of our evaluation including the demo. We are grateful that the Reviewer found the problem motivation compelling and the results impressive across classical and learning-based registration frameworks. We address the remaining concerns and clarifications below.
>
> **This work enables faster/larger registration, but it lacks a clear section on how to apply this. It does not specify which registration methods are directly supported, how to integrate the kernels, or which methods are not supported and why.**
>
> We thank the Reviewer for pointing this out. Our work is packaged as a library written in PyTorch and CUDA that has an automatic Parallel State management (with 2D parallelism) similar to Megatron-LM, and nn.Modules for grid sampler, Cross Correlation and Mutual Information, where the internal implementation is distributed-aware. For single-GPU applications, users can simply call the functions like they would in a regular PyTorch script.
>
> ```python
> from ffdp.losses import FusedLNCC, FusedMI
> from ffdp.sampler import composite_grid_sampler
> baseline_loss_fn = Vanilla_LNCC()  # baseline
> ours_loss_fn = FusedLNCC(kernel_size=9, )  # ours
> grid = F.affine_grid(affine, shape=fixed_image.shape)
> moved_image_baseline = F.grid_sample(moving_image, grid + warp, )
> moved_image_ours = composite_grid_sampler(moving_image, affine, warp)  # no grid required
> loss = ours_loss_fn(fixed_image, moved_image_ours)
> loss.backward()
> ```
> For distributed settings, a registration script would mimic a torchrun script with Megatron-style initialization. An illustrative pseudocode is shown below:
> ```
> from ffdp.parallel import parallel_state
> from ffdp.losses import FusedLNCC, FusedMI
> from ffdp.sampler import composite_grid_sampler
> ps = parallel_state.init()  # world size and rank is initialized from torchrun script, sets device id based on process rank
> # initialize image into torch tensors
> device = torch.cuda.get_current_device()  # gets device depending on rank
> fixed_image_shard = ps.shard_image(fixed_image_cpu).to(device)
> moving_image_shard = ps.shard_image(moving_image_cpu).to(device)
> warp_shard = ps.shard_image(torch.zeros(1, H, W, D, 3), warp=True).to(device)
> optim_shard = Adam([warp_shard])  # or use FireANTs for diffeomorphic
> loss_fn = FusedLNCC()   # has access to parallel state internally
> for i in range(100):
>       moved_image = ps.ring_sampler(moving_image_shard, warp_shard)
>       # if we do not want to use ring sampler
>       # moving_image_gather = ps.allgather_across_gp(moving_image_shard)
>       # moved_image = composite_grid_sampler(moving_image_gather, warp_shard)
>       loss = loss_fn(moved_image, fixed_image_shard)
>       loss.backward()
>       optim.step()
> ps.save_warp_distributed(warp_shard)
> ```
> and the script can be called with `torchrun –nproc-per-node=8 <rdvz parameters for multi node> distributed_registration.py`, fully leveraging the distributed API of PyTorch.
> Currently, FFDP is integrated into TransMorph and FireANTs. Technically, any framework written in PyTorch (e.g. several deep learning methods, iterative methods like ConvexAdam) can natively support FFDP and its full functionality by switching out the respective modules. It is harder to add support for tools like CLAIRE and ITK-DReg that use other backends like PETSc or ITK. Our library comes with detailed user guides and documentation, which we will release upon acceptance.
>
> **Only mentioned related work on GPU/system acceleration. The author does not survey other GPU acceleration outside registration.**
>
> We have mentioned a few relevant works in the introduction. Lines 81-90 in the old version (now Lines 77-84 and Lines 104-114) mention a few highlights of GPU/system acceleration, including the widely used FlashAttention family of papers and Distributed FlashAttention, Megatron with 5D parallelism used to train up to trillion-parameter models, long-sequence training, PyTorch FSDP, ThunderKittens for tiled kernels, and RingAttention. We follow up with a more extensive discussion of related works in the Appendix.
>
> However, taking this feedback and the rebuttal-phase page limit into account, we have added a dedicated Related Works section in the main paper and further extended the discussion of Related Works in the Appendix.
>
> We hope we have addressed all the Reviewer’s concerns and encourage the Reviewer to revise their score to further champion our paper. We also look forward to more discussion and an opportunity to further clarify any points in the paper.

---

> > ### Comment · Reviewer_oLeD · 2025-11-25
> > **Reply to rebuttal**
> >
> > Thank you for your response addressing my concern. Based on the review of other reviewers and your reply, I will raise my score, but I will still maintain relatively low confidence.

---

> > > ### Author Response · Authors · 2025-11-27
> > > **Thank you!**
> > >
> > > We thank the Reviewer for engaging in the discussion and are glad that we were able to address any remaining concerns.
> > >
> > > Authors of #5251

---

### Author Response · Authors · 2025-12-02

Dear Area Chair,


Please find a summary of the main concerns of all reviewers and our efforts to address them to make the paper stronger.
Reviewers `oLeD`, `TnBN`, and `bohM` had largely positive feedback, agreeing on the importance of the problem motivation [`oLeD,TnBN`], novelty, practicality, and performance of the proposed framework [`oLeD,TnBN,bohM,hXkU`], rich and comprehensive empirical section [`oLeD,TnBN,bohM`], and good writing and organization of the paper [`bohM`]. Reviewer `hXkU`’s primary concerns were the lack of motivation for full-resolution registration and limited benefit for deep learning pipelines, which we have meticulously addressed in the Authors’ Response.

We summarize the primary concerns of each reviewer and our main changes:

- Reviewer `oLeD` requested additional discussions on how to apply the proposed framework and the scope of tools that can leverage the proposed methods, and to survey additional related work on GPU acceleration outside registration. We engaged in the discussion, demonstrating sample scripts to incorporate the fast and memory efficient kernels and distributed optimization, showing its broad applicability in most frameworks written in PyTorch. We also clarified and pointed out the non-registration GPU acceleration literature that we surveyed. and added a dedicated Related Works section in the main paper and further extended the discussion of Related Works in the Appendix. After considering the Author Response and other reviews, Reviewer `oLeD` increased their rating score to 10 while maintaining their confidence score.

- Reviewer `TnBN` requested experimental evaluations on accuracy against counterparts beyond efficiency and speedup. We clarified that we already provided a convincing experimental setup on realistic synthetic data that reflects the resolution at which data is typically collected for neuropathological studies, and a familiar setup for deep learning methods to perform well at a downsampled clinical resolution. Both accuracy and efficiency were studied under increasing resolutions, and we invited the Reviewer for further discussion or clarification, upon which they responded positively and increased their score. We did not see what they changed their score to.

- Reviewer `bohM` recommended the best possible score (10), and most of the weaknesses and questions were clarifications on choice of notation, experimental setup (demonstrate applicability on histology), acknowledging tradeoffs of multi-resolution registration for accuracy and memory efficiency, and the authors’ perspective on limitations of deep methods for high resolution registration. We believe all the clarifications were very constructive and addressable. We followed up with a detailed point by point response, which we believe adequately clarified their concerns.

- Reviewer `hXkU` initially posited that downsampling is adequate and requested strong motivation for full-resolution registration, and requested more analysis on the modest improvement in memory efficiency for deep networks. They also requested more clarification on experiment setup, intro and related work, and adding FireANTs+patchification as a baseline. We responded by clarifying that the primary improvement for deep networks is a $6-7\times$ speedup, which comes with 25% less memory overhead. We provided extensive evidence for the gap between image acquisition and image registration capabilities in various clinical and life science domains, and added them to the paper. Our revised paper has a clearer Introduction and Related Works section that explicitly motivates the need for full-scale registration, and we clarified the experiments section, and added references as per the Reviewer’s feedback. We also added the FireANTs + patchification baseline for the synthetic data evaluation. The Reviewer mentioned that their concerns were addressed and they would increase their score if we could address a followup question regarding the performance of the FireANTs baseline. We responded with the answer which we believe addressed their concern adequately.

In summary, the reviewer feedback was greatly instrumental in improving the paper’s clarity, strengthening the motivation, and improving coverage of related work. We have made significant changes to the paper and our Author’s responses have addressed the reviewers’ primary concerns. Overall, 3 out of 4 reviewers had a positive sentiment about the paper pre-rebuttal, and after the rebuttal all reviewers were satisfied with the Author Response and either increased or showed willingness to improve their rating.

---

### Meta-Review · Area_Chair_o6WG · 2026-01-07

**Summary:**

This paper received three positive recommendations and one negative recommendations in the initial stage. The authors have provided detailed responses and the reviewer with negative recommendation acknowledged that the the major concerns have been well addressed.

**Reviewer Concerns:**

All the major concerns have been well addressed.

**Reviewer Scores:**

Reviewer #hXkU may change the rating to positive.

---

### Decision · Program_Chairs · 2026-01-26

Accept (Oral)